

# Higher-point integrands in $\mathcal{N}=4$ super Yang-Mills theory

**Till Bargheer[1][*], Thiago Fleury[2][†] and Vasco Gonçalves[3][‡]**

**1** Deutsches Elektronen-Synchrotron DESY, Notkestr. 85, 22607 Hamburg, Germany
**2** International Institute of Physics, Federal University of Rio Grande do Norte,
Campus Universitário, Lagoa Nova, Natal, RN 59078-970, Brazil
**3** Centro de Fisica do Porto e Departamento de Fisica e Astronomia,
Faculdade de Ciencias da Universidade do Porto, Porto 4169-007, Portugal

[*] till.bargheer@desy.de ,   [†] tsi.fleury@gmail.com ,
[‡] vasco.dfg@gmail.com

## Abstract

We compute the integrands of five-, six-, and seven-point correlation functions of twenty-prime operators with general polarizations at the two-loop order in $\mathcal{N}=4$ super Yang–Mills theory. In addition, we compute the integrand of the five-point function at three-loop order. Using the operator product expansion, we extract the two-loop four-point function of one Konishi operator and three twenty-prime operators. Two methods were used for computing the integrands. The first method is based on constructing an ansatz, and then numerically fitting for the coefficients using the twistor-space reformulation of $\mathcal{N}=4$ super Yang–Mills theory. The second method is based on the OPE decomposition. Only very few correlator integrands for more than four points were known before. Our results can be used to test conjectures, and to make progresses on the integrability-based hexagonalization approach for correlation functions.



# 1 Introduction

Correlation functions of locals operators are among the most interesting physical observables in a conformal gauge theory. In four-dimensional $\mathcal{N} = 4$ supersymmetric Yang–Mills theory (SYM), they are even more relevant, due to a rich structure of dualities between correlation functions, scattering amplitudes, and null polygon Wilson loops [1–6].

A particularly important class of operators in $\mathcal{N} = 4$ SYM are scalar single-trace half-BPS operators. Their scaling dimensions and three-point functions are protected from quantum corrections due to supersymmetry [7]. They appear to be the simplest operators in the theory, yet their higher-point correlation functions encode a wealth of information about more complex operators and observables via the operator product expansion, null limits *etc*. Correlation functions of these operators have served as an essential laboratory for holography and the development of computational methods since the early days of the AdS/CFT duality (see [8] for a recent review). Their all-loop integrands are conjectured to take part in a hidden ten-dimensional symmetry that extends the correlator/amplitude duality to Coulomb-branch amplitudes [9, 10]. Correlation functions of single-trace half-BPS operators are also most amenable to the integrability-based hexagonalization approach [11–15], especially at higher points and higher genus [16–20].

Despite their essential role, correlators of half-BPS operators beyond one-loop order have only been computed for four points, with few exceptions [19–22]. For the further exploration of new techniques, symmetries, and dualities, more perturbative data is highly needed. The goal of this paper is to produce exactly such higher-point and higher-loop data. For half-BPS operators of lowest charge (called $\mathbf{20'}$ operators), it is possible to write a compact formula for correlation functions for any number of operator insertions at tree and one-loop level. These results are reviewed in Appendix A. The main result of this paper is the computation of the

integrand for the five-, six-, and seven-point function of $\mathbf{20'}$ operators at two loops, as well as the five-point function at three loops. In all cases, we are assuming general polarizations for the external operators.

Perturbative computations are, in general, notoriously hard in quantum field theory. The number of diagrams that have to be dealt with using standard methods grows factorially with the loop order. Another issue is that symmetries of a given theory might not be manifest within conventional approaches. Fortunately, there is a more efficient alternative in $\mathcal{N} = 4$ SYM that exploits the underlying symmetries of the theory. The paradigmatic example is the computation of astoundingly high loop order integrands for four-point functions of half-BPS operators [23, 24]. This method takes advantage of the so-called Lagrangian insertion procedure to relate the integrand of an $\ell$-loop correlation function of $n$ operators with the tree-level correlator of $n + \ell$ operators, where the additional $\ell$ operators are Lagrangian insertions. The structure of these tree-level correlators is highly constrained; they are given just by rational functions of the positions, which in turn makes it possible to construct an ansatz for this object.[1] Then the problem boils down to fixing the coefficients in the ansatz.

For four-point correlation functions of half-BPS operators, it was possible to use consistency conditions (*i.e.* a bootstrap approach) to completely determine the coefficients in the ansatz up to high loop orders [26]. For five or more operators, this strategy is in general more complicated (however it is simple at one-loop order, as we show below). We attempt such a bootstrap approach, with success for five points at two-loop order, see Section 5. One reason for the increased difficulty at higher points lies in our lack of understanding of the consequences of superconformal Ward identities for more than four points. Notice that these identities can be written in a very compact notation using superconformal invariants as in [27, 28].[2] Solving these superconformal Ward identities would also be very helpful in the strong-coupling limit, as emphasized in [22].

Given this state of affairs, we take a different approach to compute the undetermined coefficients in the ansatz, taking advantage of the reformulation of $\mathcal{N} = 4$ SYM in twistor space. The chiral part of correlation functions of stress-tensor multiplets can be computed using a perturbative prescription based on twistor methods [29].[3] This prescription allows a numerical treatment firstly applied in [34] to compute the four-loop non-planar integrand of four $\mathbf{20'}$ operators. Notice that the integrand was known before (by using symmetries and dynamical constraints) only up to four unknown coefficients multiplying four polynomials [23]. The same numerical method is used in this work for computing several new correlators. In fact, it can be used in many more cases, and the main difficulty at the moment is the construction of an ansatz (basis of integrals) with not too many undetermined coefficients.

For computing correlation functions using the twistor methods of [29], it is necessary to introduce an auxiliary twistor that breaks some of the symmetries at intermediate stages of the computation. Nevertheless, the resulting correlators are independent of the auxiliary twistor, and this happens only after summing all Feynman diagrams. A different method of computation that maintains all the symmetries manifests at all steps was proposed in [27] (again, see also the review [8]). Namely, the chiral correlators transform covariantly under the $\mathcal{N} = 4$ superconformal transformations, see Section 3. Thus, it is possible to construct superconformal invariants and multiply them by all possible polynomials, imposing $S_n$ permutation symmetry. The result is a basis for the integrand, and many of the coefficients can be fixed by imposing

---

[1]In [2, 25], the Lagrangian insertion procedure together with super-Feynman rules was used to obtain the integrand for five- and six-point functions of $\mathbf{20'}$ operators at two loops, in the limit where consecutive points are light-like separated and in a special polarization.

[2]A complete analysis of $G_{6,1}$, *i.e.* six-points at Grassmann degree 4, was performed in [27].

[3]It is not known in general how to reconstruct the full super-correlator from the knowledge of its chiral part alone. For four-point functions, the procedure is described in [30]. For half-BPS operators of charge $k$, one can use the composite operators of [31–33].

physical constraints. For example, the six-point tree-level correlator could be computed by this procedure [27]. In this particular case, the correlator could be completely fixed by Ward identities and the light-like limit. In this work, we only write explicitly results which would correspond to particular fermionic components in such a basis of invariants. Nevertheless, it would be great to understand the interplay between our results and the invariants further and to look for simplifications, for example from Ward identities.[4]

In our computation, we restrict ourselves to the parity-even part of the integrand. The action of $\mathcal{N} = 4$ SYM theory will also generate parity-odd terms. However, these parity-odd terms will integrate to zero, since the only parity-odd term in the Lagrangian is the topological term $iF\tilde{F}$, which is a total derivative.[5] For this reason, we exclude parity-odd terms from the very beginning, by not including them in our ansatz. This is easily done, since all parity-odd terms will involve the four-dimensional totally symmetric epsilon tensor, which we simply disregard. The twistor computation produces the parity-even as well as the parity-odd terms. Since we work in Lorentzian signature, the parity-odd terms numerically appear as imaginary parts and are easily dropped. That being said, it would in principle be possible to reconstruct also the parity-odd part by matching the imaginary terms against a suitable ansatz. We refrain from doing so in this paper.

The paper is organized as follows. Section 2 contains the ansatz for the basis of integrals used for the numerical fitting. In Section 3, the twistor reformulation of $\mathcal{N} = 4$ SYM is reviewed, in particular the perturbative prescription and the twistor-space Feynman rules. Our results for the five-, six- and seven-points two-loop correlators as well as the five-point three-loop correlator are given in Section 4. In Section 5, we describe a bootstrap approach, and we fix the two-loop five-point function using it. We also analyze the OPE of the integrated correlator, and extract the two-loop four-point function of one Konishi operator and three **20'** operators from our data. We end the paper with a discussion in Section 6. Additional details, some definitions and a review of some results in the literature appear in several Appendices.

## 2 Ansatz

**Correlation Functions.** We are interested in correlation functions

$$\mathcal{G}_n = \langle \mathcal{O}_1 \mathcal{O}_2 \dots \mathcal{O}_n \rangle = \sum_{\ell=0}^{\infty} g^{2\ell} \mathcal{G}_n^{(\ell)}, \qquad \text{with} \qquad g^2 = \frac{g_{\text{YM}}^2 N_c}{4\pi^2}, \tag{1}$$

of local single-trace scalar half-BPS operators

$$\mathcal{O}_i = \text{tr}[(Y_i \cdot \Phi(x_i))^{k_i}], \qquad \Phi = (\phi_1, \dots, \phi_6), \qquad Y_i \cdot Y_i = 0, \tag{2}$$

where $N_c$ is the rank of the gauge group $\text{SU}(N_c)$,[6] $g_{\text{YM}}$ is the Yang–Mills coupling constant, $\phi_i$ are the six scalar fields of $\mathcal{N} = 4$ SYM, and $Y_i$ are six-dimensional null polarization vectors. Let us comment on the dependence on the number of colors $N_c$: By absorbing a factor of $N_c$ in the coupling $g$, we are anticipating the 't Hooft planar large-$N_c$ limit, in which all dependence on $N_c$ is contained in $g$. In the full finite-$N_c$ theory, the coefficients $\mathcal{G}_n^{(\ell)}$ still have a non-homogeneous dependence on $N_c$, with subleading terms in $1/N_c$ signifying non-planar corrections. In all our computations, we make no assumption on planarity. However, we still

---

[4]A conjecture about the role of the 10d symmetry of [10] for higher-point functions written in terms of invariants was put forward in [8].

[5]For a discussion of parity and the fate of parity-odd terms in the $\mathcal{N} = 2$ superspace approach, see Appendix A of [2].

[6]Whether the gauge group is $\text{U}(N_c)$ or $\text{SU}(N_c)$ does not make a difference for the correlators we consider in this work. This is generally true when all external operators are half-BPS operators of charge two (**20'** operators) [34].

find that all terms $\mathcal{G}_n^{(\ell)}$ computed in this paper are homogeneous in $N_c$, *i.e.* are free of non-planar corrections and only consist of the planar contribution. At one- and two-loop order, this is understood, since we focus on correlators of lowest-charge operators ($\mathbf{20'}$ operators) whose dependence on $N_c$ is particularly simple, and because there are no non-planar Feynman integrals at one- and two-loop order. At three loops, non-planar terms may start contributing, but we find that they are absent in our results (see also Section 4.5 below).

One way to compute loop corrections to correlation functions in perturbation theory is via the Lagrangian insertion method. Here, the $\ell$-loop contribution is given by an integral over the spacetime positions of $\ell$ insertions of the Lagrangian operator [35]

$$g^{2\ell}\mathcal{G}_n^{(\ell)} = \int\left(\prod_{i=1}^{\ell} d^4 x_{n+i}\right)G_n^{(\ell)},$$

$$G_n^{(\ell)} = \langle \mathcal{O}_1\ldots\mathcal{O}_n\mathcal{L}_{n+1}\ldots\mathcal{L}_{n+\ell}\rangle_{\text{tree}}, \qquad \mathcal{L}_i = \mathcal{L}(x_i), \qquad (3)$$

where $G_n^{(\ell)}$ is the $(n+\ell)$-point correlator with $\ell$ insertions of the chiral on-shell Lagrangian operator $\mathcal{L}$, evaluated at leading order in perturbation theory.[7] For supersymmetric theories, the Lagrangian insertion method was first introduced in [7]. It has played a key role in the study of $\mathcal{N}=4$ SYM correlation functions, especially in constructing the four-point integrand to high loop orders [3, 23, 24, 26, 29, 34, 35, 37–41].

The correlators $\mathcal{G}_n^{(\ell)}$ are functions of both the operator positions $x_i$ and the polarization vectors $Y_i$. Due to the internal SO(6) invariance, $\mathcal{G}_n^{(\ell)}$ (as well as $G_n^{(\ell)}$) are polynomials in the basic invariants $Y_i \cdot Y_j$. From the operator definition (2), it follows that the correlators are homogeneous in each $Y_i$ with weight $k_i$. Writing the invariants $Y_i \cdot Y_j$ in terms of propagator factors

$$d_{ij} = \frac{Y_i \cdot Y_j}{x_{ij}^2}, \qquad x_{ij}^2 = (x_i - x_j)^2, \qquad (4)$$

the tree-level $(n+\ell)$-point function can be unambiguously decomposed into a finite sum[8]

$$G_n^{(\ell)} = C_{k_1\ldots k_n}g^{2\ell}\sum_{\boldsymbol{a}}\left(\prod_{1\le i<j\le n}d_{ij}^{a_{ij}}\right)f_{\boldsymbol{a}}^{(\ell)}\left(x_{ij}^2\right), \qquad \boldsymbol{a} = \{a_{ij} \mid 1\le i<j\le n\}. \qquad (5)$$

Here, we have pulled out an overall constant prefactor $C_{k_1\ldots k_n}$ that depends on $N_c$ and the charges $k_i$. The explicit factor $g^{2\ell}$ arises from the Lagrangian insertions and is required for consistency. The sum over $\boldsymbol{a}$ is a finite sum over polarization structures $\prod_{ij}d_{ij}^{a_{ij}}$ that absorbs all dependence on the polarization vectors $Y_i$, such that the coefficient functions $f_{\boldsymbol{a}}^{(\ell)}$ only depend on the coordinates $x_i$.[9] The polarization structures that can occur in the sum over $\boldsymbol{a}$ are constrained by the charges of the operators:

$$\text{For all } i = 1,\ldots,n: \quad k_i = \sum_{i\neq j=1}^{n}a_{ij} \qquad (a_{ij}\equiv a_{ji}). \qquad (6)$$

---

[7]This formula naively follows from differentiating the path integral expression for $\mathcal{G}_n$ with respect to the coupling constant. The effect of the differentiation on the coupling-dependent operators is canceled by contact terms. The chiral on-shell Lagrangian is obtained from the full Lagrangian by applying equations of motion. See *e.g.* [36] for a careful treatment.

[8]A similar decomposition for the loop correlators $\mathcal{G}_n^{(\ell)}$ directly follows.

[9]The functions $f_{\boldsymbol{a}}^{(\ell)}$ in general may also carry a non-trivial dependence on $N_c$. However, all examples computed in this paper are free of subleading terms in $1/N_c$, and therefore all dependence on $N_c$ can be absorbed in the overall factor $C_{k_1\ldots k_n}$.

**Coefficient Functions.** Each polarization structure is multiplied by a rational function $f_a^{(\ell)}$ of the $n+\ell$ positions $x_i$. Due to Lorentz invariance, the functions $f_a^{(\ell)}$ only depend on squared distances $x_{ij}^2$.[10] We can further constrain their form by considering their singularity structure, which is constrained by the operator product expansions (OPEs) among the external half-BPS operators $\mathcal{O}_i$ as well as the internal Lagrangian operators $\mathcal{L}$. Since the functions $f_a^{(\ell)}$ constitute components of a tree-level correlation function (5), we only need to consider the tree-level OPEs. Let us first consider the case of two external operators $\mathcal{O}_i$ and $\mathcal{O}_j$. In their tree-level OPE, all inverse powers of $x_{ij}^2$ originate in Wick contractions of the constituent fields $Y_i \cdot \Phi(x_i)$, and hence only occur in the combination $d_{ij} = Y_i \cdot Y_j / x_{ij}^2$. The OPE therefore takes the form[11]

$$\mathcal{O}_i \times \mathcal{O}_j \sim \sum_{0 \le k} d_{ij}^k \times (\text{regular}), \tag{7}$$

where (regular) stands for terms of order $\mathcal{O}\left(x_{ij}^0\right)$. This shows that all inverse powers of $x_{ij}$ in the OPE are accompanied by numerators $Y_i \cdot Y_j$. But all dependence of the correlation function $G_n$ on the polarizations $Y_i$ is absorbed in the propagator products $\prod d_{ij}^{a_{ij}}$, hence the coefficient functions $f_a^{(\ell)}$ must be regular:

$$f_a^{(\ell)} \xrightarrow{x_i \to x_j} \mathcal{O}\left(x_{ij}^0\right), \qquad 1 \le i, j \le n. \tag{8}$$

Next, consider the OPE between one external operator $\mathcal{O}_i$ and one internal Lagrangian operator $\mathcal{L}_j$. A basic analysis shows that it has the form[12]

$$\mathcal{O}_i \times \mathcal{L}_j \sim \frac{C_{\mathcal{O}\mathcal{L}\mathcal{O}}}{x_{ij}^4} \mathcal{O}_i + \mathcal{O}\left(x_{ij}^{-2}\right). \tag{9}$$

However, the coefficient of the leading singularity is proportional to the three-point function $C_{\mathcal{O}\mathcal{L}\mathcal{O}} \sim \langle \mathcal{O}_i \mathcal{L} \bar{\mathcal{O}}_i \rangle$, which vanishes since the two-point function $\langle \mathcal{O}_i \bar{\mathcal{O}}_i \rangle$ is protected. Hence only the subleading term contributes: $\mathcal{O}_i \times \mathcal{L}_j \sim \mathcal{O}\left(x_{ij}^{-2}\right)$. Finally the OPE of two Lagrangian operators reads[13]

$$\mathcal{L}_i \times \mathcal{L}_j \sim \delta^4(x_{ij})(...) + \mathcal{O}\left(x_{ij}^{-2}\right). \tag{10}$$

The dominant contributions are contact terms proportional to $\delta^4(x_{ij})$. We only consider the tree-level correlation function (5) for Lagrangians at distinct points, hence we can ignore these terms and are only left with the subleading $\mathcal{O}\left(x_{ij}^{-2}\right)$ term.[14] The two relations (9) and (10) imply that

$$f_a^{(\ell)} \xrightarrow{x_i \to x_j} \mathcal{O}\left(x_{ij}^{-2}\right), \qquad 1 \le i \le n + \ell, \quad n < j \le n + \ell. \tag{11}$$

---

[10]The positions could also appear in invariant contractions $\varepsilon_{\mu\nu\sigma\tau} x_i^\mu x_j^\nu x_k^\sigma x_\ell^\tau$ with the totally antisymmetric tensor $\varepsilon_{\mu\nu\sigma\tau}$. However, because of parity symmetry, our results do not contain such terms. Nevertheless, the twistor action is chiral, and therefore generates such terms at the integrand level, but they are always total derivatives because they follow from a topological term in the action. See Section 3 for more details.

[11]See [41] for a more detailed discussion of this OPE.

[12]If there was a lower-dimension operator in the OPE, it would involve three or four contractions between $\mathcal{O}_i$ and $\mathcal{L}$. The relevant term in the Lagrangian is $\text{tr}([\phi_n, \phi_m][\phi^n, \phi^m])$, hence such contractions evaluate to zero.

[13]A simple explanation for the form of this OPE is as follows: The chiral Lagrangian $\mathcal{L} \simeq \mathfrak{Q}^4 \mathcal{O}^{(2)}$ is a superdescendant of the chiral primary $\mathcal{O}^{(2)}$. The fact that $\mathfrak{Q}\mathcal{L} = 0$ without total derivatives implies that the contact term proportional to $\delta^4(x_{ij})$ is also proportional to the chiral Lagrangian [36]. Moreover the three-point functions $\langle \mathcal{L}\mathcal{L}\bar{\mathcal{O}} \rangle$ are invariant under a U(1)$_Y$ "bonus symmetry" [7, 42], under which $\mathfrak{Q}$ has charge $+1$. Thus $\bar{\mathcal{O}}$ must be of the form $\bar{\mathfrak{Q}}^8 \mathcal{P}$ for a chiral primary $\mathcal{P}$, that is it must be part of a long multiplet. The lowest-dimension long multiplet is the Konishi multiplet, in which case $\bar{\mathcal{O}}$ has dimension $2 + 8/2 = 6$. Therefore all operators in the regular part of the $\mathcal{L} \times \mathcal{L}$ OPE have dimension at least six, which means the divergence is at most $1/x_{ij}^2$. We thank Paul Heslop for clarifying this point.

[14]These contact terms however play an important role for the consistency of the Lagrangian insertion method [36, 37, 39].

Combining the two constraints (8) and (11), the rational functions $f_a$, when written as single fractions, take the form

$$f_a^{(\ell)} = \frac{P_a^{(\ell)}}{\prod_{(i,j)\in I} x_{ij}^2}, \qquad I = \{(i,j) \mid 1 \le i \le n+\ell, \, n < j \le n+\ell, \, i < j\}, \qquad (12)$$

where the numerators $P_a^{(\ell)}$ are *polynomials* in the squared distances $x_{ij}^2$. The degree of these polynomials in the various $x_i$ is fixed by the conformal weights of the respective operators: The external BPS operators $\mathcal{O}_i$ have conformal weights $k_i$, and the Lagrangian $\mathcal{L}$ has conformal weight 4. Hence also the correlator $G_n$ has these weights in the respective points $x_i$. The squared distances $x_{ij}^2$ have conformal weights $-1$ at points $x_i$ and $x_j$, the products $\prod d_{ij}^{a_{ij}}$ have weights $k_i$ at all external points, and the denominator factor in (12) has weights $\ell$ at external points, and weights $n+\ell-1$ at internal points. It follows that $P_a^{(\ell)}$ must have conformal weights $-\ell$ at all external points, and conformal weights $5-n-\ell$ at all internal points. Hence for all external points $i$, the total degree in all $x_{ij}^2$, $j \ne i$ must be $\ell$, whereas for all internal points $i$, the total degree in all $x_{ij}^2$, $j \ne i$ must be $(n+\ell-5)$.

**Correlators of 20′ Operators.** For **20′** operators, the charge is $k = 2$. In this case, there are only few possible terms ($Y$-structures) in the sum over $a$ in equation (5). Up to permutations of the external points, the only possibilities for the five-point function are

$$a_{12} = 2, \quad a_{34} = a_{45} = a_{15} = 1, \quad \text{all other } a_{ij} = 0, \qquad (13)$$

which we call the **2 × 3** component, and

$$a_{12} = a_{23} = a_{34} = a_{45} = a_{15} = 1, \quad \text{all other } a_{ij} = 0, \qquad (14)$$

which we call the **5** component. We will label the functions $f_a$ multiplying the respective propagator products as $f_{2\times 3} \equiv f_{23}$ and $f_5$. All other possible configurations $a$ are obtained from (13, 14) by permutations of the five external points. By invariance of $G_n^{(\ell)}$ under such permutations, the component functions $f_{23}$ and $f_5$ uniquely determine all other component functions $f_a$. In other words, the five-point correlator of **20′** operators can be written as

$$G_5^{(\ell)} = C_{22222} \, g^{2\ell} \Big\langle d_{12}^2 d_{34} d_{45} d_{53} f_{23}^{(\ell)}(x_{ij}^2) + d_{12} d_{23} d_{34} d_{45} d_{51} f_5^{(\ell)}(x_{ij}^2) \Big\rangle_{S_5}, \qquad (15)$$

where $\langle \cdot \rangle_{S_5}$ denotes averaging over permutations of the five external points. Similarly, the six-point and seven-point functions can be written as

$$G_6^{(\ell)} = C_{2\dots 2} \, g^{2\ell} \Big\langle d_{12}^2 d_{34}^2 d_{56}^2 f_{222}^{(\ell)}(x_{ij}^2) + d_{12}^2 d_{34} d_{45} d_{56} d_{63} f_{24}^{(\ell)}(x_{ij}^2) \qquad (16)$$
$$+ d_{12} d_{23} d_{31} d_{45} d_{56} d_{64} f_{33}^{(\ell)}(x_{ij}^2) + d_{12} d_{23} d_{34} d_{45} d_{56} d_{61} f_6^{(\ell)}(x_{ij}^2) \Big\rangle_{S_6},$$

$$G_7^{(\ell)} = C_{2\dots 2} \, g^{2\ell} \Big\langle d_{12}^2 d_{34}^2 d_{56} d_{67} d_{75} f_{223}^{(\ell)}(x_{ij}^2) + d_{12}^2 d_{34} d_{45} d_{56} d_{67} d_{73} f_{25}^{(\ell)}(x_{ij}^2)$$
$$+ d_{12} d_{23} d_{31} d_{45} d_{56} d_{67} d_{74} f_{34}^{(\ell)}(x_{ij}^2) + d_{12} d_{23} d_{34} d_{45} d_{56} d_{67} d_{71} f_7^{(\ell)}(x_{ij}^2) \Big\rangle_{S_7}.$$

For the correlators computed in this paper, we find

$$C_{2\dots 2} = \frac{N_c^2 - 1}{(2\pi)^{2n+2\ell}}. \qquad (17)$$

Some words on the notation: For a correlator of a general number $n$ of **20′** operators, all possible polarization structures $\prod d_{ij}^{a_{ij}}$ take the form of a set of polygons, where each vertex

Table 1: Numbers of independent coefficients in various ansatz polynomials $P_a^{(\ell)}$ that enter the correlator (5) via (12). (The double occurrence of 2435 is not a typo.) See also Table 5 for further reduced ansatz sizes.

| | $\ell = 2$ | $\ell = 3$ |
|---|---|---|
| $P_{23}^{(\ell)}$ | 64 | 3286 |
| $P_5^{(\ell)}$ | 66 | 3576 |
| $P_{222}^{(\ell)}$ | 235 | 46873 |
| $P_{24}^{(\ell)}$ | 572 | 137596 |
| $P_{33}^{(\ell)}$ | 173 | 32701 |
| $P_6^{(\ell)}$ | 657 | 174074 |

| | $\ell = 2$ |
|---|---|
| $P_{223}^{(\ell)}$ | 2435 |
| $P_{25}^{(\ell)}$ | 4637 |
| $P_{34}^{(\ell)}$ | 2435 |
| $P_7^{(\ell)}$ | 6143 |

| | $\ell = 2$ |
|---|---|
| $P_{2222}^{(\ell)}$ | 4170 |
| $P_{224}^{(\ell)}$ | 21709 |
| $P_{233}^{(\ell)}$ | 10808 |
| $P_{26}^{(\ell)}$ | 48419 |
| $P_{35}^{(\ell)}$ | 21153 |
| $P_{44}^{(\ell)}$ | 11264 |
| $P_8^{(\ell)}$ | 68453 |

| | $\ell = 2$ |
|---|---|
| $P_{2223}^{(\ell)}$ | 68255 |
| $P_{225}^{(\ell)}$ | 205851 |
| $P_{234}^{(\ell)}$ | 197580 |
| $P_{27}^{(\ell)}$ | 547435 |
| $P_{333}^{(\ell)}$ | 17474 |
| $P_{36}^{(\ell)}$ | 232772 |
| $P_{45}^{(\ell)}$ | 205851 |
| $P_9^{(\ell)}$ | 818180 |

represents one $\mathbf{20'}$ operator, and each edge represents a propagator $d_{ij}$. A polarization is therefore uniquely specified by a monotonically increasing set of integers

$$(r_1, r_2, \ldots, r_s), \qquad r_i \leq r_{i+1}, \qquad \sum_{i=1}^{s} r_i = n, \tag{18}$$

where $s$ is the number of polygons, and $r_i$ is the size of the $i$'th polygon. As can be seen in (16), we label the corresponding coefficient functions $f_a \equiv f_{r_1 \ldots r_s}$ by these sequences. By convention, the first $r_1$ operators populate the first polygon, the next $r_2$ operators populate the second polygon, and so on.

**Graph Counting.** To complete the construction of the ansatz, it remains to find the most general polynomials $P_a^{(\ell)}$ in (12) for the various Y-structures $a$. By mapping each factor $x_{ij}^2$ to an edge between vertices $i$ and $j$, we can identify each monomial in $x_{ij}^2$ with a multi-graph (*i.e.* a graph that admits "parallel" edges between the same vertices $i$ and $j$). Finding the most general polynomials hence amounts to listing all multi-graphs with $n$ external vertices with valency $\ell$, and $\ell$ internal valencies with valency $n + \ell - 5$, and taking a general linear combination of the corresponding monomials.

Here, we can make use of permutation symmetry: Each propagator structure $\prod_{ij} d_{ij}^{a_{ij}}$ typically is invariant under a residual group $K_a \subset S_n$ of permutations of the external points $\{1, \ldots, n\}$. By the total $S_n$ permutation symmetry of the full correlator $G_n^{(\ell)}$, the respective component polynomial $P_a$ must also respect that residual permutation symmetry. Moreover, all polynomials $P_a$ must be fully symmetric under $S_\ell$ permutations of the Lagrangian insertion points $\{n+1, \ldots, n+\ell\}$. We can thus impose these symmetries on the ansatz polynomials from the beginning, which significantly reduces the numbers of undetermined coefficients.

For the correlators of $\mathbf{20'}$ operators, we list the numbers of independent terms for the various polynomials in Table 1. For more details on the construction of these ansatz polynomials, see Appendix B.

**Gram Identities.** An $n$-point correlation function in a four-dimensional Lorentz-invariant theory has $4n - 10$ kinematic degrees of freedom. On the other hand, there are $n(n-1)/2$ different squared distances $x_{ij}^2$. Hence the $x_{ij}^2$ must satisfy some non-trivial relations. One way to construct such relations is as follows.

Table 2: Statistics of Gram determinant relations at various stages. *First row:* Numbers of $7 \times 7$ minors of $\boldsymbol{X}$, up to permutations of integration labels. *Second row:* Numbers of relations that remain after canonicalizing each relation over $K_a$ permutations. *Third row:* Numbers of relations after saturating the weights by multiplying with all possible monomials and canonicalizing *each term* over $K_a$ permutations. *Last row:* Final numbers of linearly independent relations that can be used to reduce the manifestly $K_a$-symmetric ansätze for the polynomials $P_a^{(\ell)}$.

| $P_{23}^{(2)}$ | $P_5^{(2)}$ | $P_{222}^{(2)}$ | $P_{24}^{(2)}$ | $P_{33}^{(2)}$ | $P_6^{(2)}$ | $P_{223}^{(2)}$ | $P_{25}^{(2)}$ | $P_{34}^{(2)}$ | $P_7^{(2)}$ | $P_{23}^{(3)}$ | $P_5^{(3)}$ |
|---|---|---|---|---|---|---|---|---|---|---|---|
| 1 | 1 | 29 | 29 | 29 | 29 | 463 | 463 | 463 | 463 | 22 | 22 |
| 1 | 1 | 6 | 10 | 6 | 7 | 55 | 59 | 55 | 49 | 9 | 6 |
| 1 | 1 | 7 | 13 | 7 | 9 | 199 | 241 | 199 | 259 | 177 | 154 |
| 1 | 1 | 5 | 9 | 5 | 7 | 81 | 102 | 81 | 115 | 103 | 91 |

Points $x^\mu$ in Minkowski space $\mathbb{R}^{1,3}$ can be identified with null rays $X \in \mathbb{R}^{2,4}$, $X^2 = 0$, $tX \cong X$ [43], for example via

$$X = \left( \tfrac{1}{2}(1 + x^2), x^\mu, \tfrac{1}{2}(1 - x^2) \right). \tag{19}$$

The fundamental two-point Lorentz invariants can then be written as $x_{ij}^2 \propto X_i \cdot X_j$. Now consider the matrix

$$\boldsymbol{X} = \left[ X_{ij} \right]_{i,j=1}^{n+\ell}, \qquad X_{ij} = \begin{cases} 0, & i = j, \\ x_{ij}^2, & i \neq j. \end{cases} \tag{20}$$

Then $X_{ij} \propto X_i \cdot X_j$. In six dimensions, at most six vectors $X_i$ can be linearly independent, hence all $7 \times 7$ minors of $\boldsymbol{X}$ must be zero. This introduces non-linear relations among the fundamental invariants $x_{ij}^2$ called *Gram determinant relations*. These relations are obviously polynomial. However, by multiplying each Gram determinant relation with suitable monomials,[15] we obtain non-trivial *linear* relations among the various terms in the ansätze for the polynomials $P_a^{(\ell)}$. These relations can be used to reduce the number of undetermined coefficients in the ansätze for $P_a^{(\ell)}$.

Concretely, we construct the independent Gram relations as follows: First, we list all $7 \times 7$ minors of the matrix $\boldsymbol{X}$. There are $p(p+1)/2$ such minors, where $p = \text{Binomial}(n+\ell, 7)$. Next, we canonicalize each of these minors over $S_\ell$ permutations of $x_{n+1}, \ldots, x_{n+\ell}$, thereby identifying expressions that only differ by permutations of the $\ell$ integration labels. The numbers of minors that remain for $(n, \ell) = (5,2), (6,2), (7,2), (5,3)$ are $(1, 29, 463, 22)$, as shown in the first row of Table 2. Each of these expressions is a non-trivial polynomial in $x_{ij}^2$ that evaluates to zero, by construction. To compare to our ansatz polynomials for $P_a^{(\ell)}$, we still need to symmetrize over the respective permutations $K_a$ of external points, and we need to saturate the weights of the relations. We do this in three steps. First, we canonicalize each relation over the permutation group $K_a$. This reduces the numbers of relations to the second row in Table 2. Next, we saturate the weights to $\ell$ for the external points and to $(n+\ell-5)$ for the integration points by multiplying each relation with all possible monomials that yield the desired weights.[16] Finally, we expand the weight-completed relations and canonicalize *each term* over the permutations $K_a$ of external points. Some relations become manifestly zero, others become identical to each other. The resulting numbers of relations are listed in the third

---

[15]Finding all suitable monomials for a given polynomial relation is another exercise in graph enumeration.

[16]For some relations, this is not possible. For example, one Gram determinant relation at $(n, \ell) = (6, 2)$ has weight deficits $-1$ in $x_7$ and $-3$ in $x_8$. Obviously, there is no monomial in $x_{ij}^2$ with such weights. We drop relations whose weights cannot be saturated by monomials.

row of Table 2. Not all of these relations are linearly independent. To find the ambiguity in our ansatz polynomials, we need to pick a linearly independent set. The sizes of the maximal linearly independent sets are listed in the last row of Table 2.

The sizes of these ambiguities might not seem big compared to the numbers of terms in the ansätze listed in Table 1. However, when matching these ansätze to data by comparison at many numerical points $x_i$, any ambiguity easily leads to very "unnatural" solutions, with arbitrary numerical coefficients. Having a good handle on the ambiguities is essential to resolve, or even better avoid, such "arbitrary" solutions.

**Ancillary File.** We provide the complete ansatz expressions for the polynomials $P_a^{(\ell)}$ as well as lists with all Gram relations for $(n, \ell) \in \{(5,2),(6,2),(7,2),(5,3)\}$ in the attached MATHE-MATICA file `ansatzAndGram.m`.

## 3 Twistors

In this section, we briefly review the reformulation of $\mathcal{N} = 4$ SYM in supertwistor space and the procedure for computing the correlation functions of the chiral stress-tensor supermultiplet using this formalism. In addition, we explain the numerical methods used for the computations. The Feynman rules in twistor space depend on the choice of an auxiliary supertwistor, however the final results for the correlators are independent of the choice of this auxiliary supertwistor and they can be expressed in terms of $\mathcal{N} = 4$ superconformal invariants. These invariants are discussed in [27, 28] and we also briefly review them here.

### 3.1 Computing correlation functions using twistors

The relevant superspace for studying $\mathcal{N} = 4$ SYM has sixteen odd variables $\theta^{a\alpha}$, $\bar{\theta}_a^{\dot{\alpha}}$, with $a = 1, \ldots, 4$ and $\alpha, \dot{\alpha} = 1, 2$. The stress-tensor supermultiplet is a short half-BPS supermultiplet and it depends on only eight of the odd variables, four chiral and four anti-chiral variables [44]. In what follows, we are mostly interested in its chiral part, i.e. all the anti-chiral variables $\bar{\theta}_a^{\dot{\alpha}}$ are going to be set to zero. In order to describe this supermultiplet, it is convenient to introduce the so called harmonic variables

$$u_a^b \equiv (u_a^{+\mathfrak{b}}, u_a^{-\mathfrak{b}'}), \quad \text{parametrizing} \quad \frac{\mathrm{SU}(4)}{\mathrm{SU}(2) \times \mathrm{SU}(2)' \times \mathrm{U}(1)}, \tag{21}$$

with $\mathfrak{b}, \mathfrak{b}' = 1, 2$ being fundamental representation indices of the $\mathrm{SU}(2)$, $\mathrm{SU}(2)'$ respectively and the $\pm$ indicates the charge under the $\mathrm{U}(1)$ factor. The harmonic variables obey several constraints due to the fact that they belong to $\mathrm{SU}(4)$, see [29] for example. Using these variables and their complex conjugates $\bar{u}$, we can decompose the odd variables as follows

$$\theta_\alpha^a = \theta_\alpha^{+\mathfrak{b}} \bar{u}_{+\mathfrak{b}}^a + \theta_\alpha^{-\mathfrak{b}'} \bar{u}_{-\mathfrak{b}'}^a, \tag{22}$$

where

$$\theta_\alpha^{+\mathfrak{b}} = \theta_\alpha^a u_a^{+\mathfrak{b}}, \qquad \theta_\alpha^{-\mathfrak{b}'} = \theta_\alpha^a u_a^{-\mathfrak{b}'}, \tag{23}$$

and similarly for $\bar{\theta}_a^{\dot{\alpha}}$. The odd variables defined above are useful for writing down an expansion of the stress-tensor supermultiplet $\widehat{\mathcal{T}}(x, \theta, \bar{\theta}, u)$ keeping the $\mathrm{SU}(4)$ symmetry manifest. The bottom state of the supermultiplet is the operator $\mathcal{O}_{20'}(x, u)$, which is annihilated by the following eight supersymmetries (and also by all the superconformal generators)

$$Q_{-\mathfrak{a}'}^\alpha \cdot \mathcal{O}_{20'}(x, u) = (\bar{Q}^+)^{\dot{\alpha}\mathfrak{a}} \cdot \mathcal{O}_{20'}(x, u) = 0, \quad \text{with} \quad Q_{-\mathfrak{a}'}^\alpha = \bar{u}_{-\mathfrak{a}'}^a Q_a^\alpha, \quad (\bar{Q}^+)^{\dot{\alpha}\mathfrak{a}} = u_a^{+\mathfrak{a}} \bar{Q}^{\dot{\alpha}a}. \tag{24}$$

Then the supermultiplet is given by

$$\widehat{\mathcal{T}}(x,\theta,\bar{\theta},u) = \exp\left(\theta_\alpha^{+\mathfrak{b}}Q_{+\mathfrak{b}}^\alpha + \bar{\theta}_{-\mathfrak{a}'}^{\dot{\alpha}}\bar{Q}_{\dot{\alpha}}^{-\mathfrak{a}'}\right)\cdot\mathcal{O}_{20'}(x,u). \tag{25}$$

In particular, its chiral part $\mathcal{T}(x,\theta^+,u)$ only depends on the four $\theta_\alpha^{+\mathfrak{b}}$ variables of (23). We have schematically

$$\mathcal{T}(x,\theta^+,u) \equiv \widehat{\mathcal{T}}(x,\theta,0,u) = \mathcal{O}^{++++}(x) + \ldots + (\theta^+)^4\mathcal{L}(x), \tag{26}$$

and in the expansion above only the relevant terms for this work were written down. The top operator $\mathcal{L}(x)$ is the chiral on-shell Lagrangian and the bottom one is the **20'** operator given by

$$\mathcal{O}^{++++}(x) = \mathrm{Tr}(\phi^{++}\phi^{++}), \qquad \text{with} \qquad \phi^{++} = \phi^{ab}u_a^{+\mathfrak{b}}u_b^{+\mathfrak{c}}\epsilon_{\mathfrak{bc}}, \tag{27}$$

with $\epsilon_{\mathfrak{bc}}$ the usual antisymmetric tensor with $\epsilon_{12} = 1$ and $\phi^{ab} = -\phi^{ba}$ are combinations of the six real scalar fields $\Phi^I$ of $\mathcal{N} = 4$ SYM, *i.e.* $\phi^{ab} = (\sigma_I)^{ab}\Phi^I$ with $(\sigma_I)^{ab}$ the SO(6) Pauli matrices. Usually, the length-two half-BPS operator is written in terms of 6d null polarization vectors $Y_I$ in the form (2),

$$\mathcal{O}_i = \mathrm{tr}[(Y_i\cdot\Phi(x_i))^2], \qquad Y_i\cdot Y_i = 0, \tag{28}$$

and the scalar propagator was given in (4). The connection between the two descriptions is made easily by choosing a convenient parametrization for the harmonic variables and their complex conjugates as the one in [29]

$$u_\mathfrak{b}^{+\mathfrak{a}} = (\delta_\mathfrak{b}^\mathfrak{a},y_{\mathfrak{b}'}^\mathfrak{a}), \quad u_\mathfrak{b}^{-\mathfrak{a}'} = (0,\delta_{\mathfrak{b}'}^{\mathfrak{a}'}), \quad \bar{u}_{+\mathfrak{a}}^\mathfrak{b} = (\delta_\mathfrak{a}^\mathfrak{b},0), \quad \bar{u}_{-\mathfrak{a}'}^\mathfrak{b} = (-y_{\mathfrak{a}'}^\mathfrak{b},0). \tag{29}$$

We have for the propagator in the representation (27)

$$\langle\phi_1^{++}\phi_2^{++}\rangle \propto \frac{1}{x_{12}^2}\epsilon^{abcd}(u_1)_a^{+\mathfrak{a}}(u_1)_b^{+\mathfrak{b}}\epsilon_{\mathfrak{ab}}(u_2)_c^{+\mathfrak{c}}(u_2)_d^{+\mathfrak{d}}\epsilon_{\mathfrak{cd}} \propto \frac{1}{x_{12}^2}(y_{12})_{\mathfrak{a}'}^{\mathfrak{b}}(y_{12})_{\mathfrak{b}}^{\mathfrak{a}'} \propto \frac{y_{12}^2}{x_{12}^2}, \tag{30}$$

where we have used

$$y_{12}^2 = -(y_{12})_{\mathfrak{a}'}^{\mathfrak{b}}(y_{12})_{\mathfrak{b}}^{\mathfrak{a}'}/2, \quad (y_{12})_{\mathfrak{a}'}^{\mathfrak{b}} = (y_1)_{\mathfrak{a}'}^{\mathfrak{b}} - (y_2)_{\mathfrak{a}'}^{\mathfrak{b}}, \quad y_{\mathfrak{b}}^{\mathfrak{a}'} = y_{\mathfrak{b}'}^{\mathfrak{a}}\epsilon^{\mathfrak{b}'\mathfrak{a}'}\epsilon_{\mathfrak{ab}}. \tag{31}$$

Notice that in this notation the null condition (28) is automatic. In this work, we are interested in computing both the components $\theta_i^+ = 0$ and $(\theta_j^+)^4$ of the correlation functions

$$\mathfrak{G}_n = \langle\mathcal{T}(1)\ldots\mathcal{T}(n)\rangle, \quad \text{with} \quad \mathcal{T}(i) = \mathcal{T}(x_i,\theta_i^+,u_i), \tag{32}$$

for several values of $n$. Notice that for $\theta_i^+ = 0$ for all $i$, we have

$$\mathfrak{G}_n\Big|_{\theta_i^+=0} = \mathcal{G}_n = \langle\mathcal{O}_1\mathcal{O}_2\ldots\mathcal{O}_n\rangle. \tag{33}$$

Despite the fact that we are interested in correlation functions of twenty prime operators $\mathcal{O}_i$, the components with $(\theta_j^+)^4$ of $\mathfrak{G}_n$ are useful for computing loop corrections via the Lagrangian insertion method (see the previous section for a complete discussion; in this section we have an additional $\theta^+$ integration when compared with similar formulas of (3))

$$\frac{1}{m!}\frac{\partial^m\mathfrak{G}_n}{\partial g_{\mathrm{YM}}^{2m}} = \int\prod_{i=1}^m d^4x_{n+i}\,d^4\theta_{n+i}^+\,\mathfrak{G}_{n+m}. \tag{34}$$

The correlation function $\mathfrak{G}_n$ in (32) admits an expansion in $\theta_i^{\alpha\mathfrak{b}+}$ of the form

$$\mathfrak{G}_n = \mathcal{G}_n + \mathcal{G}_{n;1} + \ldots + \mathcal{G}_{n;n}, \tag{35}$$

where by definition $\mathcal{G}_{n;k}$ has Grassmann degree $4k$ and the other orders in $\theta$ are zero by symmetry. Moreover, the correlators transform covariantly under the $\mathcal{N} = 4$ superconformal transformations. Consider the following combination of sixteen generators

$$(\epsilon \cdot Q) = \epsilon_\alpha^a Q_a^\alpha, \qquad (\bar{\xi} \cdot \bar{S}) = \bar{\xi}^{\dot{\alpha}a} \bar{S}_{\dot{\alpha}a}. \tag{36}$$

Note that the generators above form an anti-commuting subalgebra

$$\{Q, \bar{S}\} = \{Q, Q\} = \{\bar{S}, \bar{S}\} = 0. \tag{37}$$

It is possible to show that the variables transform as [44]

$$\delta x \propto \bar{\theta}, \qquad \delta u^+ \propto \bar{\theta}, \qquad \delta\bar{\theta} \propto \bar{\theta}^2, \tag{38}$$

and

$$\theta_i^{\alpha\mathfrak{b}+} \to \hat{\theta}_i^{\alpha\mathfrak{b}+} = \theta_i^{\alpha\mathfrak{b}+} + (\epsilon^{\alpha a} + x_i^{\alpha\dot{\alpha}}\xi_{\dot{\alpha}}^a)u_{ia}^{+\mathfrak{b}} = e^{(\epsilon \cdot Q)+(\bar{\xi}\cdot\bar{S})}\theta_i^{\alpha\mathfrak{b}+}. \tag{39}$$

Inspecting the transformations above, we see that the value $\bar{\theta} = 0$ is left unchanged and only the $\theta_i^{\alpha\mathfrak{b}+}$ transforms. Therefore the sixteen transformation parameters $\{\epsilon_\alpha^a, \bar{\xi}^{\dot{\alpha}a}\}$ can be used to set for example (for any $\alpha$ and $\mathfrak{b}$)

$$\theta_{n-3}^{\alpha\mathfrak{b}+} = \theta_{n-2}^{\alpha\mathfrak{b}+} = \theta_{n-1}^{\alpha\mathfrak{b}+} = \theta_n^{\alpha\mathfrak{b}+} = 0. \tag{40}$$

Thus, the expansion (35) in fact truncates for this choice:

$$\mathfrak{G}_n = \mathcal{G}_n + \mathcal{G}_{n;1} + \ldots + \mathcal{G}_{n;n-4}. \tag{41}$$

A special case is $n = 4$, which implies that all the $\theta_i^+$'s can be set to zero in this case.

Because the set of fermionic generators considered above commute among themselves and are nilpotent, it is possible to define the following set of invariants $\mathcal{I}_{n;k}$, see [8, 27, 28]

$$\mathcal{I}_{n;k}(x, y, \theta^+) = Q^8 \bar{S}^8 \mathcal{J}_{n,k+4}(x, y, \theta^+) = \int d\epsilon \, d\xi \, \mathcal{J}_{n,k+4}\left(x, y, \hat{\theta}^+\right), \tag{42}$$

with $\mathcal{J}_{n,k}(x, y, \theta^+)$ completely unconstrained, $\hat{\theta}^+$ was defined in (39) and $\epsilon^{\alpha a}, \xi_{\dot{\alpha}}^a$ are the fermionic parameters of the transformations appearing in that formula. As before the second index $k$ indicates that the object has Grassmann degree $4k$ and the fermionic generators remove sixteen $\theta_i^+$'s in all possible ways (not necessary the ones in (40), but that particular case is among the resulting terms). Notice that the fermionic generators annihilates $x$ and $y$ because of (38).

The components of the correlation functions can then be expanded as follows

$$\mathcal{G}_{n,k}(x, y, \theta^+) = \sum_i \left(\mathcal{I}_{n;k}\right)_i (x, y, \theta^+) f_{n;k;i}(x). \tag{43}$$

It is non-trivial that the functions $f_{n;k;i}(x)$ do not depend on the polarizations $y$'s. However, this follows from considering the behavior of the correlators under inversion. All the dependency on the $y$'s will come from the invariants which also depend on the positions $x$.

The sum over $i$ goes through all the independent invariants. It is not easy to count this number precisely apart from extremal cases. In addition under an arbitrary permutation $\sigma$ of the points $(x_i, y_i, \theta_i^+) \to (x_{\sigma \cdot i}, y_{\sigma \cdot i}, \theta_{\sigma \cdot i}^+)$, the functions satisfy[17]

$$f_{n;k;i}(x_1, \ldots, x_n) = f_{n;k;\sigma \cdot i}(x_{\sigma \cdot 1}, \ldots, x_{\sigma \cdot n}). \tag{44}$$

As mentioned in the Introduction, it will be nice to write our results in the form (43). In [29], a procedure for computing $\mathfrak{G}_n$ using supertwistor techniques was described, and we are going to perform the calculations using the numerical version of it firstly used in [34]. The procedure uses the twistor $\mathcal{N} = 4$ SYM action given in [45, 46]. Supertwistors $\mathcal{Z}^A$ live in the complex projective superspace $\mathbb{CP}^{3|4}$, and they are parametrized as ($\chi^a$ are fermionic coordinates)

$$\mathcal{Z}^A = (Z^I, \chi^a), \quad \text{with} \quad Z^I = (\lambda_\alpha, \mu^{\dot\alpha}), \tag{45}$$

and $Z^I$ are bosonic twistors. A spacetime point $x^{\dot\alpha\beta}$ corresponds to a line in bosonic twistor space, more precisely, the relation of these twistor variables with the $\mathcal{N} = 4$ superspace variables when all the $\bar\theta_a^{\dot\alpha}$'s are zero is given by the following incidence relations[18]

$$\mu^{\dot\alpha} = i x^{\dot\alpha\beta} \lambda_\beta, \qquad \chi^a = \theta^{a\alpha} \lambda_\alpha. \tag{46}$$

For computing correlation functions perturbatively, we need both an expression for the superfield $\mathcal{T}(x, \theta^+, u)$ and the Feynman rules in twistor space. We only summarize the formulas here and we refer the reader to [29] for details and derivations. The chiral stress-tensor supermultiplet $\mathcal{T}(x, \theta^+, u)$ in this language is given by

$$\mathcal{T}(x, \theta^+, u) = \int d^4\theta^- L_{\text{int}}(x, \theta), \tag{47}$$

where $L_{\text{int}}(x, \theta)$ is the interaction Lagrangian in twistor space. This follows as a consequence of the Lagrangian insertion procedure. The interaction term $L_{\text{int}}(x, \theta)$ in a particular gauge is a sum of infinitely many terms containing the one-form superfield $\mathcal{A}(\mathcal{Z}_{1,2})$

$$
\begin{aligned}
\mathcal{A}(\mathcal{Z}_{1,2}) = {} & a(Z_{1,2}) + \chi^a \psi_a(Z_{1,2}) + \frac{1}{2} \chi^a \chi^b \phi_{ab}(Z_{1,2}) \\
& + \frac{1}{3!} \epsilon_{abcd} \chi^a \chi^b \chi^c \psi'^d(Z_{1,2}) + \frac{1}{4!} \epsilon_{abcd} \chi^a \chi^b \chi^c \chi^d a'(Z_{1,2}).
\end{aligned} \tag{48}
$$

In the formula above, the fields $a(Z_{1,2})$ and $a'(Z_{1,2})$ are the two helicity gluons, $\psi_a(Z_{1,2})$ and $\psi'^d(Z_{1,2})$ are the gluinos and $\phi_{ab}(Z_{1,2})$ are the six scalars. The bosonic twistors $Z_{1,2} = \{Z_1, Z_2\}$ are two independent twistors parametrizing a line given by the spacetime point where the fields are defined. Since $L_{\text{int}}(x, \theta)$ has terms with arbitrary many superfields $\mathcal{A}(\mathcal{Z}_{1,2})$, a general order perturbative calculation of the correlation functions can have in principle vertices with arbitrary valences. Given a set of operators $\mathcal{T}(i)$ at space-time positions $x_i^\mu$ determining lines in twistor space parametrized by two independent twistors $Z_{i,1}, Z_{i,2}$, the Feynman rules in the so called axial gauge are summarized below. This gauge choice is defined by the vanishing of the superfield $\mathcal{A}(\mathcal{Z}_{1,2})$ of (48) in the direction of an auxiliary twistor $\mathcal{Z}_\diamond$. It is possible to take the fermionic part of $\mathcal{Z}_\diamond$ to zero without losing generality. The bosonic part of it will be denoted by $Z_\diamond$. Of course, any correlation function result is expected to be independent of the $Z_\diamond$ choice. Notice that the individual diagrams depend on the value of $Z_\diamond$, and this breaks the manifest $\mathcal{N} = 4$ superconformal invariance at intermediate steps, however there are cancellations among the graphs, see [29].

---

[17]We are using a very compact notation here. The index $i$ labels the invariants, but can also depend on the operator labels. This is the reason why the index transforms under an arbitrary permutation $\sigma$.

[18]The relations are more complicated when $\bar\theta_a^{\dot\alpha} \neq 0$. They can be found for example in [47, 48].

The Feynman rules are the following. If the lines $i$ and $j$ are connected by a propagator, the graph is multiplied by the factor $(y_{ij}^2/x_{ij}^2)\delta^{a_i a_j}$ where the delta function is a color delta function. The $m$-valence vertex connecting the line $i$ to the lines $j_1, \dots, j_m$ is given by

$$V^i_{j_1,\dots,j_m} = R^i_{j_1,\dots,j_m} \operatorname{Tr}(T^{a_1} \dots T^{a_m}), \tag{49}$$

where $T^a$ are $U(N_c)$ generators[19] and

$$R^i_{j_1 j_2 \dots j_k} = -\int \frac{d^4\theta_i^-}{(2\pi)^2} \frac{\delta^2\left(\langle \sigma_{ij_1}\theta_i^-\rangle + A_{ij_1}\right)\delta^2\left(\langle\sigma_{ij_2}\theta_i^-\rangle + A_{ij_2}\right)\dots\delta^2\left(\langle\sigma_{ij_k}\theta_i^-\rangle + A_{ij_k}\right)}{\langle\sigma_{ij_1}\sigma_{ij_2}\rangle\langle\sigma_{ij_2}\sigma_{ij_3}\rangle\dots\langle\sigma_{ij_k}\sigma_{ij_1}\rangle}, \tag{50}$$

with

$$\sigma^\alpha_{ij} = \epsilon^{\alpha\beta}\frac{\langle Z_{i,\beta}Z_\diamond Z_{j,1}Z_{j,2}\rangle}{\langle Z_{i,1}Z_{i,2}Z_{j,1}Z_{j,2}\rangle}, \qquad \text{and} \qquad \langle\sigma_i\sigma_j\rangle = \epsilon_{\alpha\beta}\sigma_i^\alpha\sigma_j^\beta. \tag{51}$$

Finally, each diagram is multiplied by an explicit factor

$$\left(\frac{g_{\text{YM}}^2}{4\pi^2}\right)^p, \tag{52}$$

where $p = P - V$, with $P$ the total number of propagators and $V$ the total number of vertices of the diagram.[20]

In the formulas above,

$$A^{\mathfrak{a}'}_{ij} = \left[\langle\sigma_{ji}\theta_j^{+\mathfrak{b}}\rangle + \langle\sigma_{ij}\theta_i^{+\mathfrak{b}}\rangle\right]\left(y_{ij}^{-1}\right)^{\mathfrak{a}'}_{\mathfrak{b}}. \tag{53}$$

The matrix $(y_{ij})^{\mathfrak{b}}_{\mathfrak{a}'}$ was defined in (31). Its inverse appearing above satisfies $(y_{ij}^{-1})^{\mathfrak{c}'}_{\mathfrak{b}}(y_{ij})^{\mathfrak{b}}_{\mathfrak{a}'} = \delta^{\mathfrak{c}'}_{\mathfrak{a}'}$ and $(y_{ij})^{\mathfrak{c}}_{\mathfrak{a}'}(y_{ij}^{-1})^{\mathfrak{a}'}_{\mathfrak{b}} = \delta^{\mathfrak{c}}_{\mathfrak{a}}$. The four bracket is defined as

$$\langle Z_1 Z_2 Z_3 Z_4\rangle = \epsilon_{IJKL}Z_1^I Z_2^J Z_3^K Z_4^L. \tag{54}$$

In particular, using the parametrization of a bosonic twistor given in (45), one has

$$\langle Z_{i,1}Z_{i,2}Z_{j,1}Z_{j,2}\rangle = (\epsilon^{\alpha\beta}\lambda_{i,\alpha}\lambda_{i,\beta})(\epsilon^{\gamma\delta}\lambda_{j,\gamma}\lambda_{j,\delta})x_{ij}^2. \tag{55}$$

In this work, we are going to perform the twistor calculations using a Lorentzian signature metric. As long as the external points span the full four-dimensional space, terms of the form $\epsilon_{\mu\nu\rho\sigma}x_i^\mu x_j^\nu x_k^\rho x_l^\sigma$ can be nonzero. In the twistor reformulation of $\mathcal{N} = 4$ SYM, the action has the total derivative term $iF\tilde{F}$ where $F_{\mu\nu}$ is the field strength and $\tilde{F}_{\mu\nu}$ its dual. The term $iF\tilde{F}$ can potentially generate such $\epsilon$ terms to the integrands, however the integrated correlator is insensitive to these contributions because they are produced by a total derivative term. Using a Lorentzian metric, the $\epsilon$ terms appear as imaginary contributions to the correlation functions, and therefore can easily be isolated numerically.[21] It is also possible to numerically bootstrap the $\epsilon$ terms by writing a basis of integrals and performing a numerical fitting, see [34] for an example. In this work, we are going to consider only the real part of the numerical results for the correlation functions obtained by the twistor method.

---

[19]In this paper we are going to consider only planar correlation functions and for this case the $U(N_c)$ and $SU(N_c)$ groups give the same results. This is also true in general when all the half-BPS external operators have length two [34].

[20]From the definition of $R$ (50), one can see that, with this definition of $p$, the diagram is of order $\theta^{4p}$.

[21]These terms also change sign when some of the $(x_i)_\mu \to -(x_i)_\mu$. So even using Euclidean signature it is possible to isolate these kind of contributions by taking special combinations of points.

Table 3: There are in total 76 six-point two-loop skeleton graphs (some of them are disconnected). At this loop order, there must be 8 vertices (length of the sequence of numbers) and 10 propagators (half the sum of the numbers). The command in Sage is `graphs(8, degree_sequence = (...))`, with `(...)` replaced by an entry of the table. The graphs can be transformed to a list of adjacency matrices and saved in a file.

| degree_sequence | # Graphs |
|---|---|
| [2,2,2,2,2,2,2,6] | 1 |
| [2,2,2,2,2,2,3,5] | 7 |
| [2,2,2,2,2,2,4,4] | 9 |
| [2,2,2,2,2,3,3,4] | 31 |
| [2,2,2,2,3,3,3,3] | 28 |

In order to compute loop corrections for the $n$-point functions $\mathcal{G}_n$ of $\mathbf{20'}$ operators, we are going to use the formula (34). At $\ell$ loops, we need to compute $G_{n+\ell}$ at order $4\ell$ on the $\theta_i^{+}$'s for $i > n$. The diagrams contributing for this case contain $n+\ell$ vertices (lines in twistor space) and $n+2\ell$ propagators. This follows from the Feynman rules described above, as the $n$-valence vertex with $n > 2$ contributes with $2(n-2)$ $\theta$'s, see (50). It then follows that the diagram is of order $\theta^{4\ell}$. The correlator $G_{n+\ell}$ of $n$ $\mathbf{20'}$ operators and $\ell$ Lagrangian insertions is extracted by setting $\theta_i^{+} = 0$ for $i \leq n$, which projects to the $\prod_{i=n+1}^{n+\ell}(\theta_i^{+})^4$ component, whose coefficient is the desired correlator. Collecting powers of $2\pi$ and $g_{\text{YM}}$, and doing the color algebra, one finds that every diagram contains an overall factor

$$\frac{N_c^2 - 1}{(2\pi)^{2n+2\ell}} \left( \frac{g_{\text{YM}}^2 N_c}{4\pi^2} \right)^{\ell} = \frac{N_c^2 - 1}{(2\pi)^{2n+2\ell}} g^{2\ell} = C_{2\dots2}\, g^{2\ell}\,. \tag{56}$$

This is exactly the overall prefactor (17).

For the connected part of the correlator, only connected diagrams are important, because it is possible to show that disconnected twistor diagrams contribute only to lower-point correlators. Notice that (50) implies that two operators can at most be connected by a single propagator, otherwise the factor $R^i_{j_1 j_2 \dots j_k}$ vanishes as it is antisymmetric. Moreover, all the vertices must have valence at least two. It is possible to generate all the necessary skeleton graphs very efficiently using SAGEMATH [49]. For example, all the six-point two-loop skeleton graphs can be generated by using the code of Table 3.

Using the skeleton graphs and the Feynman rules, it is possible to generate all the graphs. The total number increases fast with the number of points $n$ and loops $\ell$, and it is hard even for the simplest cases to do any analytical simplification. Thus we have evaluated all the graphs numerically. A great simplification is that even numerically it is possible to select a particular polarization from the beginning, which projects out many permutations and graphs. The vertices $R^i_{j_1 j_2 \dots j_k}$ contain the factors of $(y_{ij}^{-1})^{\mathfrak{a}'}_{\mathfrak{b}}$ inside the $A_{ij}^{\mathfrak{a}'}$, see (53), and in principle these factors can change the factors of $y_{ij}^2$ coming from the propagators. However, the factors $(y_{ij}^{-1})^{\mathfrak{a}'}_{\mathfrak{b}}$ multiply a $\theta_i^{+}$ or a $\theta_j^{+}$ and because we are only interested in the contributions with $\theta_i^{+} = 0$ for $i \leq n$ the $R$-charge coming from the external propagators are never canceled.

For example, in the case of five- and six-point functions, it is possible to select the operator polarizations in such way that only the cyclic contribution $(y_{12}^2 y_{23}^2 \dots y_{i1}^2)$, or any other disconnected contribution, is non vanishing. This is not true for $n > 6$ points. The operators

for the cyclic case are given in [50] and they read for five points

$$\mathcal{O}_1 = \mathrm{Tr}(XX), \quad \mathcal{O}_2 = \mathrm{Tr}(\bar{X}\bar{Y}), \quad \mathcal{O}_3 = \mathrm{Tr}(\bar{Z}Y), \quad \mathcal{O}_4 = \mathrm{Tr}(ZZ), \quad \mathcal{O}_5 = \mathrm{Tr}(\bar{Z}\bar{X}), \tag{57}$$

where the bar means complex conjugation, and the fields $\{X, Y, Z\}$ are defined in terms of the real scalars $\Phi^I$ as

$$X = \frac{1}{\sqrt{2}}\left(\Phi^1 + i\Phi^2\right), \quad Y = \frac{1}{\sqrt{2}}\left(\Phi^3 + i\Phi^4\right), \quad Z = \frac{1}{\sqrt{2}}\left(\Phi^5 + i\Phi^6\right). \tag{58}$$

For the six-point case, one has,

$$\begin{aligned} \mathcal{O}'_1 &= \mathrm{Tr}(XX), \quad \mathcal{O}'_2 = \mathrm{Tr}(\bar{X}\bar{Y}), \quad \mathcal{O}'_3 = \mathrm{Tr}(YY), \\ \mathcal{O}'_4 &= \mathrm{Tr}(\bar{Z}\bar{Y}), \quad \mathcal{O}'_5 = \mathrm{Tr}(ZZ), \quad \mathcal{O}'_6 = \mathrm{Tr}(\bar{Z}\bar{X}). \end{aligned} \tag{59}$$

These polarizations can be reproduced by selecting particular values for the matrices $(y_i)^{\mathfrak{a}}_{\mathfrak{b}'}$ appearing in the definitions of the harmonic variables in (29). We define the function

$$y'(a, b, c, d) = a\,\mathbf{e}_{11} + b\,\mathbf{e}_{12} + c\,\mathbf{e}_{21} + d\,\mathbf{e}_{22}, \tag{60}$$

where $\mathbf{e}_{ij}$ is a $2 \times 2$ matrix with a single nonzero component at position $\{i, j\}$ with value 1. The cyclic six-point polarization, for example, can be taken to be

$$(y_1)^{\mathfrak{a}}_{\mathfrak{b}'} = y'(1, 0, 0, 0), \qquad (y_2)^{\mathfrak{a}}_{\mathfrak{b}'} = y'(0, 0, 1, 1), \qquad (y_3)^{\mathfrak{a}}_{\mathfrak{b}'} = y'(0, 1, 0, 0), \tag{61}$$

$$(y_4)^{\mathfrak{a}}_{\mathfrak{b}'} = y'(0, 0, 1, 0), \qquad (y_5)^{\mathfrak{a}}_{\mathfrak{b}'} = y'(1, 0, -1, 1), \qquad (y_6)^{\mathfrak{a}}_{\mathfrak{b}'} = y'(0, 1, 1, 1/2). \tag{62}$$

The explicit polarizations that we used in our computation are given in Section 4.1 below.

Finally, the numerical results for the correlators obtained with the twistors were fitted against the ansatz of integrals described in Section 2. The positions of the operators were generated randomly, and the number of equations were always greater than the number of unknowns coefficients in the basis.

# 4 Results

## 4.1 Method

**Strategy.** We fix the free coefficients in the ansatz polynomials $P_a^{(\ell)}$ constructed in Section 2 by matching the ansatz correlators (15, 16) against the $(n + \ell)$-point tree correlator computed from twistors (Section 3) on many numerical points $(x_i, y_i)$. The numerical data points provide a linear system for the free coefficients in the ansatz that we solve numerically. The solution is not unique due to non-trivial Gram determinant relations among the various terms in the ansatz. However, at least at two loops, we notice that once we restrict the ansätze for the component functions $f_a^{(2)}$ to a certain set of conformal integrals, all ambiguity is removed, and the solution becomes unique.

**Computational Aspects.** In the computation of the data points and finding the solution for the ansatz parameters, there are two main bottlenecks:

- The symbolic algebra of Grassmann-odd variables $\theta_i$ in the twistor computation, especially when the number of contributing twistor diagrams becomes large,

- Solving large and dense numerical linear systems.

Table 4: The numbers of twistor graphs that contribute to the two- and three-loop correlators for the various choices of polarizations.

|  | $Y_{23}$ | $Y_5$ | $Y_6$ | $Y_{7,1.1}$ | $Y_{7,1.2}$ | $Y_{7,2}$ | $Y_{7,3}$ |
|---|---|---|---|---|---|---|---|
| $\ell = 2$ | 557 | 1790 | 221 910 | 21 154 | 21 154 | 24 502 | 18 688 |
| $\ell = 3$ | 73 380 | 167 430 |  |  |  |  |  |

For the first point, we could boost the performance by representing homogeneous polynomials in a finite number of Grassmann-odd variables as component vectors. Multiplication of two or more homogeneous polynomials can then be implemented by precomputed numerical tensors, such that the computation becomes completely numerical. With this and some other optimizations as well as parallelization, we could compute a few thousand data points per day on a 48-core machine.

The second point is somewhat more essential: Our method inevitably produces large and completely dense linear systems, whose coefficients are either high-precision floats, or large rationals. Solving such systems is a hard computational problem. Using MATHEMATICA's Nsolve, we could solve such systems up to size $\sim 10\,000$, but would run out of memory beyond that, even on a machine with 256 GB memory.

**Polarizations.** In order to contain the sizes of the linear systems, as well as the numbers of graphs that contribute to the twistor computation, we identify a few numerical choices for the polarizations $Y_i$, $1 \le i \le n + \ell$ that set all but a few of the polarization structures $\prod d_{ij}^{a_{ij}}$ to zero. For each polarization choice, we then evaluate the ansatz and compute the twistor correlator for many numerical values of the coordinates $x_i$, $1 \le i \le n + \ell$. At five points, we use the following two polarizations:

$$
\boldsymbol{Y}_{23} = \begin{pmatrix} -1 & 0 & 1 & i & 0 & -i \\ 1 & 0 & 1 & i & 0 & -i \\ 0 & -1 & 1 & 0 & -i & -i \\ 0 & -1 & 1 & 0 & i & -i \\ 0 & -1/3 & 2 & 2i & -i/3 & 0 \end{pmatrix}, \qquad \boldsymbol{Y}_5 = \begin{pmatrix} -1 & 0 & 1 & i & 0 & -i \\ 1 & 0 & 1 & i & 0 & -i \\ 1 & -1 & 3 & 3i & i & i \\ 0 & -1 & 2 & 2i & -i & 0 \\ 2 & -3 & 2 & 4i & -i & 0 \end{pmatrix}, \tag{63}
$$

where we collected the external $Y_i$, $1 \le i \le 5$ into a vector $\boldsymbol{Y}$. In all cases, the polarizations of the integration points $Y_i$, $i > n$ is set to arbitrary fixed values, such that all $d_{ij}$, $i \le n$, $j > n$, are non-zero. For the choice $\boldsymbol{Y}_{23}$, the only non-zero polarization structure is

$$
d_{12}^2 d_{34} d_{45} d_{53} = \frac{128}{3 x_{12}^4 x_{34}^2 x_{45}^2 x_{35}^2}, \tag{64}
$$

hence we can use this polarization to determine the $f_{23}$ component function. For the choice $\boldsymbol{Y}_5$, the only non-zero polarization structure is

$$
d_{12} d_{23} d_{34} d_{45} d_{51} = \frac{-64}{x_{12}^2 x_{23}^2 x_{34}^2 x_{45}^2 x_{15}^2}, \tag{65}
$$

hence we can use this polarization to determine the $f_5$ component function. We list the numbers of twistor graphs that contribute to the two- and three-loop correlators for the various choices of polarizations in Table 4. At six points, we use the relatively random polarization

$$Y_6 = \begin{pmatrix} \frac{622}{7} & -\frac{274}{13} & \frac{1101}{91} & \frac{624}{7}i & -\frac{272}{13}i & \frac{919}{91}i \\ 0 & -2 & 1 & 2i & 0 & -i \\ 2 & -4 & 6 & 6i & 2i & 4i \\ -4 & -\frac{1171}{78} & \frac{333}{26} & 8i & -\frac{1169}{78}i & \frac{281}{26}i \\ -15 & -12 & -30 & 7i & -14i & -32i \\ \frac{111}{2} & 80 & -127 & -\frac{113}{2}i & -76i & -129i \end{pmatrix}. \tag{66}$$

With this polarization, all $d_{ij}$ are non-zero, so all component functions contribute. At two loops, the total number of unknowns in the ansatz then is (see Table 1) $235 + 572 + 173 + 657 = 1637$. A linear system of this size is still easily solvable. At seven points, we use four different polarizations:

$$Y_{7.1.1} = \begin{pmatrix} -1 & 0 & 1 & i & 0 & -i \\ 1 & -1 & 1 & i & i & -i \\ 0 & -1 & 1 & 0 & -i & -i \\ 0 & -1 & 1 & 0 & i & -i \\ 0 & -1 & 2 & 2i & i & 0 \\ 1 & -2 & 0 & i & 0 & -2i \\ 1 & 0 & 1 & i & 0 & -i \end{pmatrix}, \quad Y_{7.1.2} = \begin{pmatrix} -1 & 0 & 1 & i & 0 & -i \\ 1 & -1 & 1 & i & i & -i \\ 0 & -1 & 1 & 0 & -i & -i \\ 0 & -1 & 1 & 0 & i & -i \\ 0 & 2 & 2 & 2i & -2i & 0 \\ -1 & -2 & 0 & -i & 0 & -2i \\ 1 & 0 & 1 & i & 0 & -i \end{pmatrix}, \tag{67}$$

$$Y_{7.2} = \begin{pmatrix} -1 & 0 & 1 & i & 0 & -i \\ 1 & -1 & 1 & i & i & -i \\ 0 & -1 & 1 & 0 & -i & -i \\ 0 & -1 & 1 & 0 & i & -i \\ -\frac{1}{2} & -\frac{3}{2} & \frac{1}{2} & \frac{1}{2}i & -\frac{1}{2}i & -\frac{3}{2}i \\ -1 & -2 & 0 & -i & 0 & -2i \\ 1 & 1 & 3 & 3i & i & i \end{pmatrix}, \quad Y_{7.3} = \begin{pmatrix} -1 & 0 & 1 & i & 0 & -i \\ 1 & -1 & 1 & i & i & -i \\ 0 & -1 & 1 & 0 & -i & -i \\ 1 & -1 & 1 & i & i & -i \\ 3 & 10 & 8 & 11i & -4i & 6i \\ \frac{16}{99} & -\frac{265}{198} & \frac{17}{9} & \frac{160}{99}i & \frac{329}{198}i & -\frac{1}{9}i \\ 12 & 11 & 13 & 12i & 13i & 11i \end{pmatrix}.$$

For the first polarization $Y_{7.1.1}$, the only non-zero polarization structures are

$$d_{23}^2 d_{45}^2 d_{16} d_{67} d_{71} = \frac{-256}{x_{23}^4 x_{45}^4 x_{16}^2 x_{67}^2 x_{71}^2}, \qquad d_{12}^2 d_{67}^2 d_{34} d_{45} d_{53} = \frac{256}{x_{12}^4 x_{67}^4 x_{34}^2 x_{45}^2 x_{53}^2}, \tag{68}$$

hence we can use this polarization to determine the 2435 coefficients of the $f_{223}$ component function. For the next polarization $Y_{7.1.2}$, the only non-zero polarization structures are

$$d_{12}^2 d_{67}^2 d_{34} d_{45} d_{53} = \frac{-128}{x_{12}^4 x_{67}^4 x_{34}^2 x_{45}^2 x_{53}^2}, \quad d_{12} d_{23} d_{34} d_{45} d_{56} d_{67} d_{71} = \frac{128}{x_{12}^2 x_{23}^2 x_{34}^2 x_{45}^2 x_{56}^2 x_{67}^2 x_{71}^2}. \tag{69}$$

Plugging in the known answer for the $f_{223}$ component function, we can therefore use this polarization to determine the 6143 coefficients of the $f_7$ component function. For the polarization $Y_{7.2}$, the only contributing components are one $f_{223}$ function, one $f_7$ function, and two $f_{25}$ functions. With the final polarization $Y_{7.3}$, the contributing components are two $f_{223}$ functions and one $f_{34}$ function. We can thus use these polarizations to independently determine the remaining components $f_{25}$ and $f_{34}$. Recall that the polarizations are expressed using two-dimensional indices in Section 3. It is possible to solve for all the $(y_i)_{\alpha'}^{\flat}$ in (31) allowing complex solutions and imposing that all the $y_{ij}^2 \sim Y_i \cdot Y_j$ obtained here are reproduced.

## 4.2 Two-loop integrals

Up to two loops and seven points, we find that the correlators of $\mathbf{20'}$ operators can be expressed in terms of the following conformally invariant integrals (see Figure 1):

$B_{123}$

$F_1^{1234}$

$F_2^{1234}$

$B_{1,23,45}$

$\Pi_{1,25,34}$

$\Delta_{12,345}$

$B_{123,456}$

$\Pi_{1,23,456}$

$\Delta_{14,23,56}$

$\Pi_{123,4567}$

$\Delta_{1,234,567}$

Figure 1: The complete set of conformal integrals that appear in the correlation functions of **20'** operators for up to seven points and two loops, see (70). *Black:* Propagators $1/x_{ij}^2$. *Red dashed:* Numerator factors $x_{ij}^2$. Further two-loop conformal integrals are shown in Figure 2.

$$F_1^{1243} \equiv x_{13}^2 x_{24}^2 \int \frac{d^4 x_5}{x_{15}^2 x_{25}^2 x_{35}^2 x_{45}^2},$$

$$B_{123} \equiv x_{12}^2 x_{13}^2 x_{23}^2 \int \frac{d^4 x_4 \, d^4 x_5}{(x_{14}^2 x_{24}^2 x_{34}^2) x_{45}^2 (x_{15}^2 x_{25}^2 x_{35}^2)} = B_{123,231},$$

$$F_2^{1243} \equiv x_{13}^2 x_{24}^2 x_{14}^2 \int \frac{d^4 x_5 \, d^4 x_6}{(x_{15}^2 x_{25}^2 x_{45}^2) x_{56}^2 (x_{46}^2 x_{36}^2 x_{16}^2)} = B_{124,341},$$

$$B_{1,23,45} \equiv x_{12}^2 x_{15}^2 x_{34}^2 \int \frac{d^4 x_6 \, d^4 x_7}{(x_{46}^2 x_{56}^2 x_{16}^2) x_{67}^2 (x_{17}^2 x_{27}^2 x_{37}^2)} = B_{132,541},$$

$$\Pi_{1,25,34} \equiv x_{25}^4 x_{34}^2 \int \frac{x_{16}^2 \, d^4 x_6 \, d^4 x_7}{(x_{26}^2 x_{36}^2 x_{46}^2 x_{56}^2) x_{67}^2 (x_{57}^2 x_{17}^2 x_{27}^2)} = \Pi_{125,3245},$$

$$\Delta_{12,345} \equiv x_{34}^2 x_{35}^2 x_{45}^2 \int \frac{x_{17}^2 x_{26}^2 \, d^4 x_6 \, d^4 x_7}{(x_{16}^2 x_{36}^2 x_{46}^2 x_{56}^2) x_{67}^2 (x_{37}^2 x_{47}^2 x_{57}^2 x_{27}^2)} = \Delta_{3,254,145},$$

$$B_{123,456} \equiv x_{13}^2 x_{46}^2 x_{25}^2 \int \frac{d^4 x_7 \, d^4 x_8}{(x_{17}^2 x_{27}^2 x_{37}^2) x_{78}^2 (x_{48}^2 x_{58}^2 x_{68}^2)},$$

$$\Pi_{1,23,456} \equiv x_{46}^2 x_{13}^2 x_{15}^2 \int \frac{x_{28}^2 \, d^4 x_7 \, d^4 x_8}{(x_{27}^2 x_{37}^2 x_{17}^2) x_{78}^2 (x_{18}^2 x_{48}^2 x_{58}^2 x_{68}^2)} = \Pi_{231,4561},$$

$$\Delta_{14,23,56} \equiv x_{13}^2 x_{14}^2 x_{46}^2 \int \frac{x_{28}^2 x_{57}^2 \, d^4 x_7 \, d^4 x_8}{(x_{17}^2 x_{27}^2 x_{37}^2 x_{47}^2) x_{78}^2 (x_{48}^2 x_{58}^2 x_{68}^2 x_{18}^2)} = x_{14}^2 x_{46}^2 \left[ \frac{\Delta_{1,234,567}}{x_{16}^2 x_{47}^2} \right]_{7 \to 4},$$

$$\Pi_{123,4567} \equiv x_{27}^2 x_{35}^2 x_{46}^2 \int \frac{x_{19}^2 \, d^4 x_8 \, d^4 x_9}{(x_{18}^2 x_{28}^2 x_{38}^2) x_{89}^2 (x_{49}^2 x_{59}^2 x_{69}^2 x_{79}^2)},$$

$$\Delta_{1,234,567} \equiv x_{13}^2 x_{16}^2 x_{47}^2 \int \frac{x_{29}^2 x_{58}^2 \, d^4 x_8 \, d^4 x_9}{(x_{18}^2 x_{28}^2 x_{38}^2 x_{48}^2) x_{89}^2 (x_{19}^2 x_{59}^2 x_{69}^2 x_{79}^2)}. \tag{70}$$

Here, $F_1$ and $F_2$ are the one-loop and two-loop ladder integrals, $B$ are double-box integrals, $\Pi$ are penta-box (pentaladder) integrals, and $\Delta$ are double-penta integrals. Additional conformal integrals that do appear in the ansätze, but whose coefficients are set to zero in the actual functions $f_a^{(2)}$ are shown in Figure 2. The integrals that *do* contribute to the functions $f_a^{(2)}$ can be characterized as follows: Either they are products of one-loop box integrals $F_1$, or they have one loop-loop propagator factor $1/x_{n+1,n+2}^2$ and at most one numerator factor $x_{ij}^2$ per integration point $j \in \{n+1, n+2\}$. Conversely, the integrals in Figure 2 that *do not* contribute fall into three classes:

- Two-loop integrals with two or more numerator factors connecting to the same integration point (first row),

- Products of one-loop integrals that include numerator factors (second row),

- Two-loop integrals that include loop-loop numerator factors $x_{n+1,n+2}^2$ (last two rows).

The fact that such integrals do not contribute is an observation for which we do not have a direct derivation at this point. What we can say is that excluding all these integrals completely removes all Gram-relation ambiguity. In other words, there is no linear combination of Gram relations that is free of these excluded integrals. This means that once these integrals are excluded, the ansatz becomes free of redundant parameters, *i.e.* all coefficients can be uniquely fixed by matching to data or any other type of constraints. Moreover, and perhaps even more importantly for future bootstraps, the sizes of the ansatz polynomials greatly reduce by dropping these integrals, especially at higher points, see Table 5.

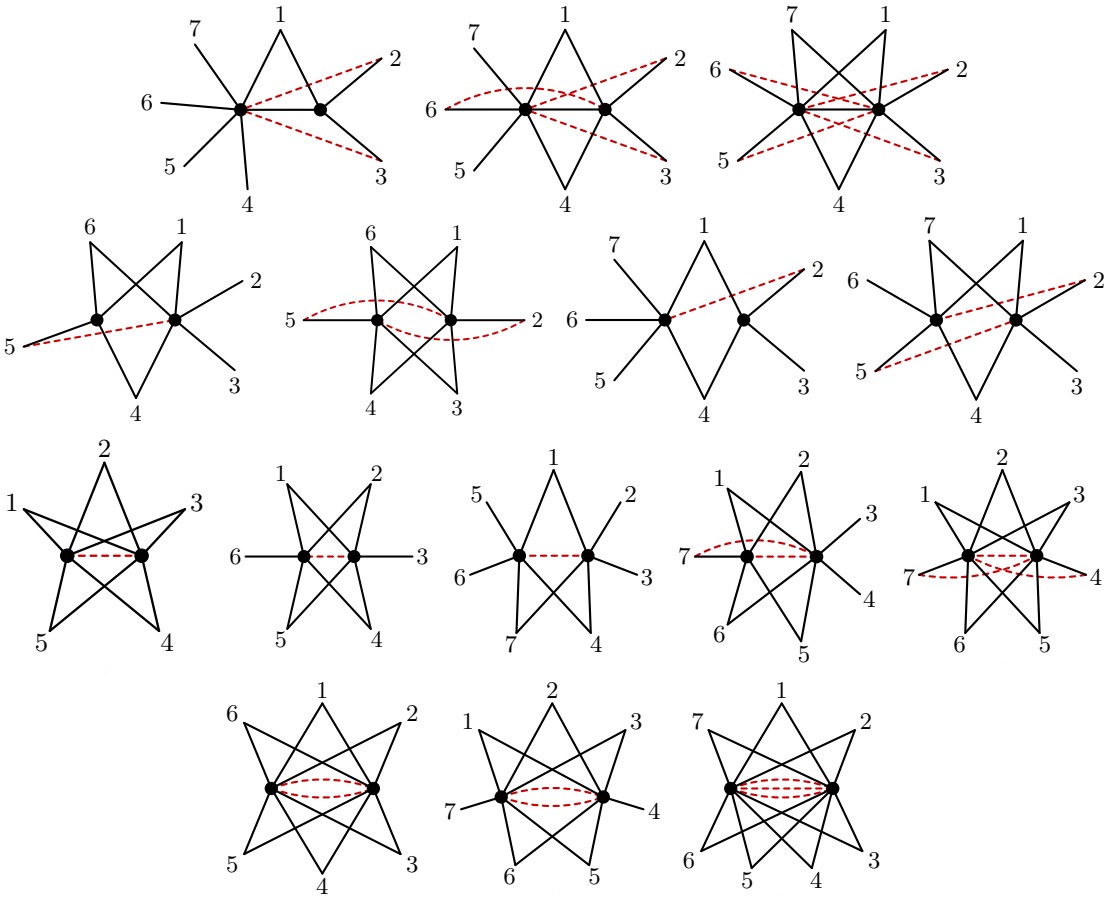

Figure 2: Two-loop conformal integrals with up to seven external points that appear in the ansätze as constructed in Section 2, but do not occur in the final two-loop correlation functions of **20'** operators. Unlike in Figure 1, numerator factors between external points are not drawn.

Table 5: Numbers of independent coefficients in the two-loop ansatz polynomials $P_a^{(2)}$ after excluding classes of integrals that do not appear for $n = 5, 6, 7$. As can be seen by comparing to Table 1, the numbers are greatly reduced, especially for $n = 8, 9$.

| | | | | | | | |
|---|---|---|---|---|---|---|---|
| $P_{23}^{(2)}$ | 59 | $P_{223}^{(2)}$ | 974 | $P_{2222}^{(2)}$ | 724 | $P_{2223}^{(2)}$ | 3948 |
| $P_5^{(2)}$ | 60 | $P_{25}^{(2)}$ | 1850 | $P_{224}^{(2)}$ | 3666 | $P_{225}^{(2)}$ | 10370 |
| | | $P_{34}^{(2)}$ | 974 | $P_{233}^{(2)}$ | 1886 | $P_{234}^{(2)}$ | 11138 |
| $P_{222}^{(2)}$ | 160 | $P_7^{(2)}$ | 2457 | $P_{26}^{(2)}$ | 7781 | $P_{27}^{(2)}$ | 25812 |
| $P_{24}^{(2)}$ | 400 | | | $P_{35}^{(2)}$ | 3500 | $P_{333}^{(2)}$ | 1116 |
| $P_{33}^{(2)}$ | 117 | | | $P_{44}^{(2)}$ | 1919 | $P_{36}^{(2)}$ | 11996 |
| $P_6^{(2)}$ | 465 | | | $P_8^{(2)}$ | 10793 | $P_{45}^{(2)}$ | 10370 |
| | | | | | | $P_9^{(2)}$ | 37083 |

## 4.3 Two-loop results

**Five Points.** We can express the final answers for the two-loop component functions $f_a^{(2)}$ in terms of the conformal integrals (70). Pulling out the overall prefactor (56) from the twistor result, we find

$$f_{23}^{(2)} = 12\Big\langle 2\Big(1 - \frac{u_4}{u_3 u_5}\Big)B_{1,23,45} + 4u_1 u_3\big(B_{3,14,25} - B_{3,12,45}\big) - B_{123}$$
$$+ 2\Pi_{3,45,12} - 2u_1 u_4 \Pi_{3,14,25} + \Big(1 - 2\frac{1}{u_1 u_4}\Big)\Pi_{3,12,45} + \Delta_{34,125}$$
$$+ \frac{u_1(2 + u_1 u_3 - u_2 u_4)}{u_2^2 u_5}F_1^{1234}F_1^{1235} + 2F_2^{1324} + \frac{u_1 u_3}{u_2}\big(6F_2^{1234} - F_2^{3142}\big)\Big\rangle_{23}, \quad (71)$$

where $\langle\cdot\rangle_{23}$ means the average over permutations in the symmetry group $K_{23} \equiv S_2 \times S_3$ of the respective propagator structure. The second ("cyclic") component function is given by

$$f_5^{(2)} = 10\Big\langle \Big(4 + \frac{2u_2 - 4 - 1/u_5}{u_1 u_3}\Big)B_{1,23,45} + \Big(\frac{4 - 2u_4 - 1/u_1}{u_3 u_5} - 2\Big)B_{1,24,35}$$
$$+ 2\Big(\frac{1}{u_1} - u_3 u_5\Big)B_{1,25,34} + 2\Big(1 + \frac{1}{u_2 u_4}\Big)\Pi_{1,23,45} + 2\big(u_3 - 1 - u_2 u_4\big)\Pi_{1,24,35}$$
$$+ \frac{2(1 - u_2 u_4 - u_3)}{u_3}\Pi_{1,25,34} + \frac{1 + u_2 u_4 - u_3}{u_3}\Pi_{1,34,25} - \Delta_{12,345}$$
$$+ \frac{1}{u_2}\Big(\frac{u_1 u_4 - 2 + 1/u_2}{u_5} - 2u_2 - 2u_1 u_3 + 2\Big)F_1^{1234}F_1^{1235} + \frac{4(1 - u_2)}{u_2}F_2^{1234}$$
$$+ (u_2 + u_1 u_3 - 1)F_2^{1243} - 2(u_1 + 2u_2 u_5)F_2^{1253} - 2u_1 u_4 F_2^{1254} + B_{123}\Big\rangle_5, \quad (72)$$

where $\langle\cdot\rangle_5$ means averaging over the symmetry group $K_5 \equiv D_5$ of dihedral permutations. In both expressions, we have used the five-point cross ratios defined in (C.2).

The above expressions are not unique due to the existence of Gram relations among the terms in the ansätze for the component functions $f_a$. At five points and two loops, there is one Gram determinant relation (see Table 2). After reducing over permutations of the integration points $\{x_6, x_7\}$, the relation has 442 terms. Further reducing each term over the permutation symmetry group $K_{23}$ ($K_5$), the number of terms in the relation reduces to 64 (66). However, the relation unavoidably includes a conformal integral that is not in the list (70), namely

$$\int \frac{dx_6\, dx_7\, x_{67}^2}{x_{16}^2 x_{26}^2 x_{36}^2 x_{46}^2 x_{56}^2\, x_{17}^2 x_{27}^2 x_{37}^2 x_{47}^2 x_{57}^2}, \quad (73)$$

which is the first integral in the third row of Figure 2. Excluding this integral makes the expressions above unique.

We find that the two component functions $f_{23}^{(2)}$ and $f_5^{(2)}$ can alternatively be written in terms of the following four monomials:

$$q_{1234567}^1 = x_{16}^2 x_{17}^2 x_{26}^2 x_{27}^2 x_{34}^2 x_{35}^2 x_{45}^2, \qquad q_{1234567}^2 = x_{16}^2 x_{17}^2 x_{25}^2 x_{27}^2 x_{34}^2 x_{36}^2 x_{45}^2,$$
$$q_{1234567}^3 = x_{15}^2 x_{16}^2 x_{27}^2 x_{34}^2 x_{36}^2 x_{45}^2, \qquad q_{1234567}^4 = x_{17}^4 x_{26}^2 x_{34}^2 x_{35}^2 x_{45}^2. \quad (74)$$

This compares with four different polynomials that appear in the three-loop four point function of $\mathbf{20'}$ operators [23]. Multiplying with the common denominator as in (12),

$$P_a^{(2)} = x_{16}^2 x_{17}^2 x_{26}^2 x_{27}^2 x_{36}^2 x_{37}^2 x_{46}^2 x_{47}^2 x_{56}^2 x_{57}^2 x_{67}^2 f_a^{(2)}, \quad (75)$$

we find that the coefficient of the $2 \times 3$ polarization $d_{12}^2 d_{34} d_{35} d_{45}$ is given by

$$
\begin{aligned}
P_{23} = \big[ & 2q^4_{1345672} - q^3_{1345267} - 2q^1_{1345627} + q^2_{1346725} + 3q^2_{1354627} + q^2_{1356427} - 2q^3_{1364257} \\
& + 4q^2_{1364527} + 2q^2_{1647325} + 2q^3_{3164572} - 4q^1_{3412567} + 2q^4_{3412567} - 8q^1_{3412756} \\
& + 4q^4_{3412756} - 2q^1_{3612547} - 2q^3_{1436275} \big] + \text{perm} ,
\end{aligned}
\tag{76}
$$

while the coefficient of the cyclic polarization $d_{12} d_{23} d_{34} d_{45} d_{15}$ reads

$$
\begin{aligned}
P_5 = \big[ & 2q^1_{1234567} - q^2_{1234567} + q^3_{1234567} - q^4_{1234567} + q^1_{1234657} - 2q^4_{1234657} + q^3_{1234675} + q^3_{1235467} \\
& - 2q^2_{1243567} - q^3_{1243567} - 2q^2_{1243657} + 4q^2_{1246357} - q^2_{1263547} + 2q^1_{1324657} - q^4_{1324657} \\
& + 2q^3_{1324675} - q^2_{1342567} + q^2_{1345267} - 2q^2_{1346257} + q^3_{1362475} - q^2_{1362547} - q^3_{1362574} + q^4_{1423657} \\
& - 2q^3_{1423675} + q^2_{1634257} + q^1_{1634527} - q^2_{1635247} - 2q^2_{1643257} + q^3_{2135467} + 2q^3_{2136457} + q^3_{2143567} \\
& - 2q^3_{2146357} + 2q^1_{2314657} - q^4_{2413657} - 2q^2_{1236457} - q^2_{1326457} - 2q^1_{1423756} \big] + \text{perm} .
\end{aligned}
\tag{77}
$$

The permutations in these equations are given by the elements of $S_7$ that leave each polarization structure fixed.

**Six Points.** At six points, there are four independent component functions, as can be seen in (16). The simplest of them reads

$$
f^{(2)}_{222} = 48 \left\langle \frac{u_1 u_3 u_5}{U_1} B_{123,456} + \frac{u_1^2 u_3 u_5 U_1}{2 u_2 u_6 U_3} F_1^{1234} F_1^{1256} \right\rangle_{222} ,
\tag{78}
$$

where $\langle \cdot \rangle_{222}$ means the average over permutations in the symmetry group $K_{222} = S_3 \ltimes (S_2 \times S_2 \times S_2)$ that stabilizes the propagator structure, *i.e.* the pairings $(1,2)$, $(3,4)$, $(5,6)$ of external points. The remaining six-point component functions are presented in Appendix D. We express the functions in terms of the six-point cross ratios (C.4).

**Seven Points.** At seven points, there are again four independent component functions (16), the simplest of them being

$$
\begin{aligned}
f^{(2)}_{223} = 48 \big\langle & 2u_{1234} u_{2456} B_{123,456} + u_{1256} u_{2643} B_{125,346} \\
& + u_{1243} u_{1256} u_{1562} F_1^{1234} F_1^{1256} + u_{1275} u_{3465} F_1^{1257} F_1^{3456} \big\rangle_{223} .
\end{aligned}
\tag{79}
$$

Here, we use the general cross ratios (C.1). As before, $\langle \cdot \rangle_{223}$ means averaging over the permutation group $K_{223} = (S_2 \ltimes (S_2 \times S_2)) \times S_3$ that stabilizes the propagator structure $d_{12}^2 d_{34}^2 d_{56} d_{67} d_{57}$ which multiplies the function $f_{223}$. The remaining three seven-point component functions are given in Appendix D.

**Any Multiplicity.** We present some (very preliminary and yet incomplete) guesses for the component functions $f^{(2)}_{2,n-2}$ and $f^{(2)}_n$ for any multiplicity $n$ in Appendix D.2.

## 4.4 Three-loop results

The ansätze for $f^{(3)}_{23}$ and $f^{(3)}_5$ contain 139 different conformal integrals.[22] Each integral appears with various permutations of the external (and internal) points. We can organize the answers as follows:

$$
f^{(3)}_{23} = 9 \left\langle \sum_{i=1}^{139} \sum_{\sigma \in \pi_i^{23}} c^\sigma_{23,i} I^{(3)}_{i,\sigma} \right\rangle_{23} , \qquad f^{(3)}_5 = 30 \left\langle \sum_{i=1}^{139} \sum_{\sigma \in \pi_i^5} c^\sigma_{5,i} I^{(3)}_{i,\sigma} \right\rangle_5 .
\tag{80}
$$

---

[22]Strictly speaking, we should not call the individual terms in three-loop ansatz conformal integrals, since we have not checked their convergence. Of course the full three-loop correlator should be free of divergences.

Here, $I^{(3)}_{i,\sigma}$ are three-loop conformal integrals

$$I^{(3)}_{i,\sigma} = I^{(3)}_i \left( x_{\sigma_1}, x_{\sigma_2}, x_{\sigma_3}, x_{\sigma_4}, x_{\sigma_5} \right), \tag{81}$$

the occurring permutations $\sigma$ of each integral $I_i$ are collected in the sets $\pi^{23}_i$ and $\pi^5_i$, and the coefficients $c^\sigma_{a,i}$ are rational functions of the five conformal cross ratios (C.2). As before, $\langle\cdot\rangle_{23}$ and $\langle\cdot\rangle_5$ denotes averaging over the permutation symmetry groups $S_2 \times S_3$ and $D_5$ respectively. The sets $\pi^{23}_i$ and $\pi^5_i$ contain between zero and eleven permutations with non-vanishing $c^\sigma_{a,i}$. The first 18 integrals $\{I^{(3)}_i \mid 1 \le i \le 18\}$ are products of one-loop and two-loop integrals,[23] the remaining $\{I^{(3)}_i \mid 19 \le i \le 139\}$ are genuine three-loop integrals. For example:

$$I_{65,(12345)} = \int dx_6 \, dx_7 \, dx_8 \, \frac{x^2_{12} x^2_{13} x^2_{15} x^2_{23} x^2_{47}}{x^2_{16} x^2_{17} x^2_{18} x^2_{27} x^2_{28} x^2_{37} x^2_{38} x^2_{46} x^2_{57} x^2_{67} x^2_{68}}, \tag{82}$$

$$c^{12345}_{23,65} = -\frac{37}{2} + \frac{9}{2u_2 u_5} + \frac{7u_1}{2u_2 u_5}, \qquad c^{12345}_{5,65} = \frac{2}{u_2 u_5} - \frac{2u_1}{u_2 u_5},$$

$$c^{13452}_{23,65} = \frac{13}{2}, \qquad c^{21345}_{5,65} = -2u_2 u_5,$$

$$c^{31245}_{23,65} = -\frac{15}{2}, \qquad c^{31245}_{5,65} = 2 + \frac{2}{u_1} - \frac{2u_2 u_5}{u_1},$$

$$c^{31452}_{23,65} = \frac{3}{2} + \frac{1}{2u_2} - \frac{u_1 u_3}{2u_2},$$

with all other $c^\sigma_{a,65} = 0$. Here, we use the five-point conformal cross ratios defined in (C.2). All integral expressions $I^{(3)}_i$ and rational coefficients $c^\sigma_{a,i}$ are provided in the attached file `results.m`. The numbers of various terms in the answers (80) are as follows:[24]

| number of: | $f^{(3)}_{23}$ | $f^{(3)}_5$ |
|---|---|---|
| occuring integrals $I^{(3)}_i$ (out of the 139) | 107 | 94 |
| non-zero $c_{i,\sigma}$ (in total) | 388 | 445 |
| terms in expanded expression inside $\langle\ldots\rangle_{\ldots}$ in (80) | 1320 | 1223 |
| terms in fully expanded expression in (80) | 13840 | 11725 |

(83)

The representation (80) is not unique: Some linear combinations of terms in the ansätze for $f^{(3)}_{23}$ and $f^{(3)}_5$ vanish due to Gram determinant relations (see Table 2).[25] Hence not all free coefficients in the ansätze are independent. Indeed, we find that matching against the twistor data leaves 103 parameters in $f^{(3)}_{23}$ unfixed. Similarly, 91 parameters in $f^{(3)}_5$ remain unfixed. We verified that this remaining freedom indeed amounts to adding Gram relations. We use this freedom to minimize the number of non-zero $c^\sigma_{a,i}$ in the answer, and to make their rational coefficients as simple as possible, by setting the remaining coefficients to particular values (effectively adding terms that sum to zero).

## 4.5 Planarity

Our results show that correlation functions of $\mathbf{20}'$ operators are free of higher-genus corrections at two loops up to seven points, and at three loops up to five points. At what loop order

---

[23]Besides the one- and two-loop integrals that appear in the two-loop ansätze (see Figure 1 and Figure 2), these products include one further two-loop integral, which features an external point that only appears in the numerator, namely: $\int dx_6 \, dx_7 \, x^2_{56}/(x^2_{16} x^2_{26} x^2_{36} x^2_{46} x^2_{67} x^2_{17} x^2_{27} x^2_{37})$.

[24]Of the 139 integrals in the ansätze, 28 do not occur in either answer.

[25]There might be further linear relations after integration, which we do not take into account here.

will the first higher-genus corrections show up? For higher-charge operators, it is clear that higher-genus terms will appear at lower loop orders. In fact, for sufficiently large charges, already the tree-level correlator will have higher-genus contributions. Consistently, also the one-loop corrections contain higher-genus terms [17]. These higher-genus terms at low loop orders do not arise from non-planar Feynman integrals, which do not exist at one and two loops. Rather, the higher-genus terms at larger charges originate in non-trivial color factors from the larger number of propagators that connect to each operator (whereas the color factors of $\mathbf{20}'$ operators are only delta functions).

We do not know about a rigorous argument for the absence of non-planar corrections at three loops at higher points. However, one can argue that for correlators of $\mathbf{20}'$ operators, the loop order at which non-planar terms start to appear will be independent of the number of inserted operators: Since their color structure is so simple, inserting more $\mathbf{20}'$ operators at a given loop order cannot increase the genus. In other words, any extra handles in the large $N_c$ expansion must be formed purely by loop corrections. Since this does not happen at four points up to three loops,[26] we find it reasonable to expect that the same will be true at higher points, *i.e.* that the perturbative order at which higher-genus terms appear will be independent of the number of inserted $\mathbf{20}'$ operators. This is consistent with the fact that we do not observe higher-genus terms in the three-loop five-point function, even though there exist non-planar three-point integrals.

It would be interesting to verify any of these arguments more rigorously.

## 4.6 Guide to the results file

All our results for the correlators of $\mathbf{20}'$ operators are collected in the attached MATHEMATICA file `results.m`. The file includes comments alongside every definition. In the file, the two-loop integrals (70) are defined as follows (see the definition of `intDef`):

$$
\begin{aligned}
&\texttt{F1[1,2,3,4]} = F_1^{1234}, && \texttt{YY[1,2,3]} = B_{123}, \\
&\texttt{F2[1,2,3,4]} = F_2^{1234}, && \texttt{LL[1,2,3,4,5]} = B_{1,23,45}, \\
& && \texttt{BB[1,2,3,4,5,6]} = B_{123,456}, \\
&\texttt{PP[1,2,5,3,4]} = \Pi_{1,25,34}, && \texttt{QQ[1,2,3,4,5]} = \Delta_{12,345}, \\
&\texttt{PB[1,2,3,4,5,6]} = \Pi_{1,23,456}, && \texttt{DP[1,4,2,3,5,6]} = \Delta_{14,23,56}, \\
&\texttt{PB7[1,2,3,4,5,6,7]} = \Pi_{123,4567}, && \texttt{DP7[1,2,3,4,5,6,7]} = \Delta_{1,234,567}, && (84)
\end{aligned}
$$

After loading the file with `<<"results.m"`, the results can be accessed through the following symbols:

`answer52A`: The two-loop five-point component functions $f_a^{(2)}$. Here, the component $a$ is specified by choosing $a = \texttt{A} \in \{23, 5\}$. The answers are written in terms of conformal integrals (84) and cross ratios (C.1). Each term in the expressions is canonicalized over the respective permutation group $K_a$, that is the answers are identical to the expressions inside $\langle \cdot \rangle$ in (71) and (72) (including the numerical prefactors).

`answer62A`: The same for six points, that is $\texttt{A} \in \{222, 24, 33, 6\}$. The answers are identical to the expressions inside $\langle \cdot \rangle$ in (78), (D.1), (D.2), and (D.3).

`answer72A`: The same for seven points, that is $\texttt{A} \in \{223, 25, 34, 7\}$. The answers are identical to the expressions inside $\langle \cdot \rangle$ in (79), (D.4), (D.5), and (D.6).

---

[26]For the four-point function, it was found that the potential genus-one term at three loops is proportional to a conformal Gram relation and thus vanishes [23].

`integrandN2A`: The fully expanded two-loop integrand of the component function $f_a^{(2)}$, where all integrands of conformal integrals as well as cross ratios are expanded in terms of squared distances $x_{ij}^2 = \texttt{x[i,j]}$, and the symmetrization (average) over the permutation group $K_a$ has been carried out. Here, $\texttt{N} \in \{5, 6, 7\}$, and A as above. The expressions are not explicitly symmetrized with respect to permutations of the integration points $\{x_{n+1}, x_{n+2}\}$. Calling

```
integrandN2A // Map[symmetrizeInt[N,2]]
```

generates a manifestly symmetric expression (*i.e.* the correct tree-level correlator component of N **20′** operators and two Lagrangian operators).

`I3def[6,7,8]`: A list of replacement rules that defines the 139 conformal integrals that contribute to the five-point three-loop function.

`cIAdef`: A list of replacement rules that defines the non-zero coefficients $c_{a,i}^\sigma$ that enter the three-loop component function $f_a^{(3)}$ in (80), where $a = \texttt{A} \in \{23, 5\}$. The coefficients are expressed in terms of the five-point cross ratios (C.2).

`answer53A`: The expressions inside the brackets $\langle \cdot \rangle$ in (80), including the numerical prefactors, where $\texttt{A} \in \{23, 5\}$. The coefficients are expressed in terms of cross ratios (C.2), the integrals $I_{i,\sigma}^{(3)}$ are left as abstract symbols.

`answer53Ax`: The same as `answer53A`, but with the integrals (or rather their integrands) as well as the cross ratios expanded in terms of squared distances $x_{ij}^2 = \texttt{x[i,j]}$.

`integrand53A`: The integrands of the component functions $f_a^{(3)}$ (80), $a = \texttt{A} \in \{23, 5\}$, completely expanded in terms of squared distances, and with the symmetrization (average) over the symmetry group $K_a$ carried out. Not symmetrized over permutations of the integration points $\{x_6, x_7, x_8\}$. As in the two-loop case,

```
integrand53A // Map[symmetrizeInt[5,3]]
```

achieves that symmetrization.

All expressions `answerN2A`, $\texttt{N} \in \{5, 6, 7\}$ are written in terms of the general cross ratios (C.1). To convert all cross ratios in the five- and six-point expressions to the basis cross ratios (C.2) and (C.4), one can use the following code:

```
{answer5223, answer525} /. u4Tox /. xToBGV5
{answer62222, answer6224, answer6233, answer626} /. u4Tox /. xToBGV6
```

# 5 OPE limit and other constraints

In Section 2 we have described in detail the construction of an ansatz for higher point ($n \geq 5$) correlation functions of twenty prime operators at two and three loops. The undetermined coefficients in the ansatz can be fixed, in principle, in two different ways. One is based on a twistor reformulation of $\mathcal{N} = 4$ SYM action, following the same strategy that was applied successfully in [34]. This was the approach of Section 3. The other is based on imposing OPE and supersymmetry constraints on the ansatz. This is the direct generalization of the four point bootstrap [23] and it is the approach we shall pursue in this section. The end result of this analysis is that we are able to fix the two-loop five-point correlator, and we are able to considerably reduce the number of undetermined coefficients of the six-point correlator at

two loops. One advantage of this approach is that it can be applied with small differences to correlators of operators with higher $R$-charges.

In a second part, we will do an OPE analysis of some of the correlation functions just obtained, and thus provide new OPE data at two loops. More concretely, we give the two-loop four-point function involving one Konishi operator in the $[0, 2, 0]$ representation and three $\mathbf{20'}$ operators.

## 5.1 Fixing the correlator at two loops

In the following we will study the OPE limit of correlation functions involving twenty prime operators and some Lagrangian insertions

$$\langle \mathcal{O}(x_1, y_1) \ldots \mathcal{O}(x_n, y_n) \mathcal{L}(x_{n+1}) \ldots \mathcal{L}(x_{n+k}) \rangle = \mathcal{G}_{n,k}(x_i, y_k). \tag{85}$$

As mentioned in the previous sections, these correlators control the loop corrections to the correlation functions of twenty prime operators. As an example, the two-loop correlator $\mathcal{G}_{n,0}$, is given by integrating the one-loop part of the correlator $\mathcal{G}_{n,1}$ over $x_{n+1}$. The advantage to do this is that it is easier to analyze the analytic structure of the correlator and interpret it in terms of OPE data. More concretely, the $\log u_1$ divergence (in the $x_{12}^2 \to 0$ limit) of the five-point correlator, where $u_1$ as in (C.2), is controlled by lower-loop information, and thus can be used to fix the undetermined coefficients.

The ansatz of $\mathcal{G}_{5,1}$, $\mathcal{G}_{5,2}$, and $\mathcal{G}_{6,1}$ up to one-loop order can be expressed in terms of a combination of one-loop four-point ladder integrals, as can be seen from the Appendix E.3. Moreover, these correlators should vanish once we impose the chiral algebra twist [51],[27] *i.e.* when the positions $x_1, \ldots, x_n$ are placed in a two-dimensional plane, and the polarizations $y_1, \ldots, y_n$ are set to

$$y_{ij}^2 = (v_i - v_j)(z_i - z_j), \tag{86}$$

with $z_i$ being a two-dimensional complex coordinate for the point $x_i$ (more explicitly $x_{ij}^2 = (z_i - z_j)(\bar{z}_i - \bar{z}_j)$), and $v_i$ a generic parameter. This is not the unique polarization for which the correlator vanishes, another one is the so-called Drukker-Plefka twist [53]

$$y_{ij}^2 = x_{ij}^2. \tag{87}$$

These last two constraints are powerful enough to completely fix the tree-level correlators $\mathcal{G}_{5,1}$ and $\mathcal{G}_{6,1}$, without imposing any other constraint from the OPE (*i.e.* it fixes the one-loop corrections to $\mathcal{G}_{5,0}$ and $\mathcal{G}_{6,0}$). This is not the case for the one-loop corrections to these correlators, as they are not completely fixed by these twists. Nonetheless, the number of undetermined coefficients in the ansatz spelled out in Section 2 is greatly reduced, as can be seen from these numbers:[28]

$$\mathcal{G}_{5,1}^{(1)} : \mathbf{64 + 66 = 130 \to 24}, \tag{88}$$

---

[27]The result of [51] implies that the loop corrections of $\mathcal{G}_{n,0}$ should vanish. It is possible to argue that up to two loops the integrand of $\mathcal{G}_{n,0}$ should also vanish in the chiral algebra twist limit, which is consistent with the twistor computation. This means that the Born level $\mathcal{G}_{n,k}$ for $k \leq 2$ vanishes in this twist limit. The three loop integrand for five point twenty prime operators does not vanish in the chiral algebra twist limit, however it does vanish once one integration is performed. For this reason, we expect that the one-loop correlators should vanish. The chiral algebra twist has been used to bootstrap higher point functions in [22, 52].

[28]We have used, in this section, a slightly improved ansatz compared to the one described in (12). This ansatz excludes polynomials that contain $(x_{ab}^2)^k$ for $k \geq \ell$ (where $a$ and $b$ are integration points). It is possible to show, using the results of Appendix E.3, that the terms excluded with this rule give products of one-loop four-point conformal integrals that are already included through other polynomials of the ansatz.

$$\mathcal{G}_{6,1}^{(1)} : \mathbf{235} + \mathbf{572} + \mathbf{173} + \mathbf{657} = \mathbf{1637} \to \mathbf{327}\,. \tag{89}$$

Another simple constraint to impose comes from studying the leading term in the Lorentzian OPE between two $\mathbf{20'}$ operators. The operators have $R$-charge, so it is better to choose wisely the polarizations of the external operators to avoid degeneracies in the OPE. One possible choice is setting the polarization to

$$Y_1 = \frac{1}{\sqrt{2}}(1, i, \alpha_1, i\alpha_1, 0, 0)\,, \qquad Y_2 = \frac{1}{\sqrt{2}}(1, i, \alpha_2, -i\alpha_2, 0, 0)\,, \tag{90}$$

taking one derivative in each $\alpha_1$ and $\alpha_2$, and subsequently setting both to zero. The corresponding operators then become

$$\mathcal{O}_1 \to \operatorname{tr}(ZX)\,, \quad \mathcal{O}_2 \to \operatorname{tr}(Z\bar{X})\,, \qquad \text{where} \qquad Z = \frac{\phi_1 + \phi_2}{\sqrt{2}}\,, \quad X = \frac{\phi_3 + \phi_4}{\sqrt{2}}\,. \tag{91}$$

The effect of this is that we are projecting into a channel where operators appearing at leading order in the OPE will have the form

$$\operatorname{Tr}(ZD^J Z) + \dots\,, \qquad J \geq 0\,, \tag{92}$$

where the dots represent different ways to distribute the derivatives. For simplicity, we will focus first on $\mathcal{G}_{5,1}$, and then generalize to higher points. The leading term in the OPE is just the BPS operator $\operatorname{Tr}(Z^2)$, and this reduces the number of unfixed coefficients to

$$\mathcal{G}_{5,1}^{(1)} : \mathbf{24} \to \mathbf{19}\,, \tag{93}$$

since the four-point function of twenty prime operators at two loops is known and should match the leading term in the OPE limit of the five-point function.

At subleading order in the OPE limit, the operators with $J \geq 2$ start to contribute. These operators are unprotected, *i.e.* their dimension depends non-trivially on the coupling constant, and one consequence is that they will give rise to $\log x_{12}^2$ terms in the OPE limit. This new structure will allow to fix more undetermined coefficients in the ansatz. The log terms come from the one-loop conformal integrals reviewed in Appendix E.3. This is one of the reasons why we decided to analyze the partially integrated correlators (85), since it allows to extract the $\log x_{12}^2$ divergences while dealing only with one-loop integrals. More importantly the coefficient of $\log x_{12}^2$ is completely determined by lower-loop data and the light-cone conformal blocks,[29] reducing the number of unfixed coefficients further to

$$\mathcal{G}_{5,1}^{(1)} : \mathbf{19} \to \mathbf{6}\,. \tag{95}$$

The remaining six constants can be fixed by looking at the singlet $R$-charge channel, *i.e.* $[0,0,0]$, in the (12) OPE. Here we have to be more cautious as there is degeneracy, *i.e.* more than one operator with the same spin and dimension contribute to the OPE. But luckily there is no degeneracy at leading order in this limit, and so we can use the lower-loop data from the

---

[29] Here we have used the light-cone conformal blocks obtained in [6] for higher-point functions. The formula reads

$$\langle \mathcal{O}(x_1)\dots\mathcal{O}(x_n)\rangle = \sum_k \frac{C_{12k}(x_{12}\cdot\partial_z)^J}{(x_{12}^2)^{\Delta_\phi - (\Delta_k - J)/2}} \int_0^1 [dt]\langle\mathcal{O}_k(x_2 + t x_{12}, z)\mathcal{O}(x_3)\dots\mathcal{O}(x_n)\rangle + \dots \tag{94}$$

where $[dt] = \Gamma(\Delta + J)dt\,(t(1-t))^{(\Delta+J)/2-1}/\Gamma^2((\Delta+J)/2)$, $\mathcal{O}_k(x,z)$ is a spin $J$ operator with polarization vector $z$ satisfying $z^2 = 0$ and the $\dots$ represent subleading terms in the light-cone limit $x_{12}^2 \to 0$. The lower-loop data in this case can be read off directly from the OPE of $\mathcal{G}_{5,1}^{(0)}$.

Konishi operator and also the stress tensor. We could have also used the correlator/amplitude duality to fix some of the above coefficients.

One issue that prevents us to go to higher loops or points is that the number of terms in the ansatz grows substantially. For this reason, it is important to look at the ansatz and check if some of the terms could be dropped based on some physical reasoning. This also has the advantage that it can be applied to correlators with higher $R$-charge. In the ansatz there are terms of the form

$$\frac{y_{12}^2 y_{15}^2 y_{23}^2 y_{34}^2 y_{45}^2}{x_{34}^2 x_{23}^2 x_{15}^2 \textcolor{red}{x_{12}^2 x_{45}^2}} \frac{\textcolor{red}{x_{12}^2 x_{45}^2} x_{12}^2 x_{45}^2}{x_{16}^2 x_{17}^2 x_{26}^2 x_{27}^2 x_{46}^2 x_{47}^2 x_{56}^2 x_{57}^2} \,, \tag{96}$$

with two double propagators in the numerator. We have highlighted in red terms that need to be generated by the interactions. One possible reason to eliminate these diagrams is that they are not present in the four-point function, and the interaction only involves four of the five points. We have also noticed that the result does not contain the following diagram:

$$\frac{y_{12}^2 y_{15}^2 y_{23}^2 y_{34}^2 y_{45}^2}{x_{12}^2 x_{15}^2 x_{23}^2 x_{34}^2 \textcolor{red}{x_{45}^2}} \frac{\textcolor{red}{x_{45}^2} x_{16}^2 x_{24}^2 x_{25}^2 x_{37}^2}{x_{17}^2 x_{26}^2 x_{27}^2 x_{36}^2 x_{46}^2 x_{47}^2 x_{56}^2 x_{57}^2 x_{67}^2} \,. \tag{97}$$

This term has five factors of $x_{ij}^2$ in the numerator, and one possible reason to eliminate such terms is that the interactions would not be able to generate so many terms in the numerator.[30] Notice that both terms (96) and (97) are not dropped by excluding the integrals in Figure 2, hence eliminating such terms reduces the ansatz further even if such integrals are excluded from the beginning.

The same strategy can be applied to the correlator $\mathcal{G}_{6,1}^{(1)}$, or in other words to the two-loop six-point function of twenty prime operators. This time the ansatz of (12) can produce diagrams with six propagators in the numerator, which at this loop order should not be possible from the interactions. Eliminating these diagrams (as well as the two double propagators mentioned in the $\mathcal{G}_{5,1}^{(1)}$ analysis), together with OPE constraints (leading log Lorentzian OPE and leading Euclidean OPE in the $[0, 2, 0]$ channel) reduces the number of undetermined coefficients (89) significantly:

$$\mathcal{G}_{6,1}^{(1)} : \mathbf{327 \rightarrow 27}. \tag{98}$$

Imposing the constraints coming from other OPE channels like the singlet, did not fix the ansatz completely in this case.[31] One use of this bootstrap exercise is to reduce both efficiently and substantially the number of undetermined coefficients that enter the approach of the previous section. This might be useful in the future at higher loops/points, since generating data with the twistor method becomes harder. Let us also point out that the bootstrap method can also be applied directly to correlation functions of operators with other $R$-charges.

## 5.2 OPE of the integrated correlator

In the previous subsection we have analyzed the OPE of partially integrated correlation functions (where a subset of the Lagrangian insertions is not integrated over). While this has proved useful to study part of the structure of the correlation function, in particular to fix it, it does not provide all the information contained in this observable. The goal of this subsection is to decompose the five-point function at two-loop level in terms of lower-point correlators.

---

[30] The only terms that can generate numerators in the action are the ones coming from the field strength and from interactions with fermions. Such terms however cannot generate five numerators at two-loop order.

[31] In principle, we could use integrability data [54,55] for the non-log part of the correlator to fix more coefficients of the ansatz.

This will give access to new two-loop four-point functions with one non-BPS operator. For simplicity, we will be working at leading and subleading orders. We are able to extract the two-loop OPE coefficients with two spinning operators and compare with a result that has been previously computed [56–58] (see [54] for an integrability-based computation).

We will take the Lorentzian OPE limit as explained in the last subsection, and subsequently take the two points to approach each other, $x_2 \to x_1$, keeping only the first three nontrivial orders. This method has been applied for five-point functions at weak and strong coupling [5, 6, 22]. One of the technical difficulties is that the correlation function is expressed in terms of conformal integrals that have not been computed before. To overcome this problem (at least partially), we will apply the method of asymptotic expansion [59–62] to these higher-point integrals. The expansions of the integrals together with the higher-point light-cone conformal blocks [5, 6, 22, 63] allows us to read off the following OPE data:[32]

$$\langle \mathrm{Tr}(ZX)(x_1)\mathrm{Tr}(Z\bar{X})(x_2)\mathcal{O}(x_3, y_3)\ldots\mathcal{O}(x_5, y_5)\rangle = x_{12}^{-2}\langle \mathrm{Tr}(Z^2)\mathcal{O}(x_3, y_3)\ldots\mathcal{O}(x_5, y_5)\rangle$$
$$+ (\text{descendant}) + \langle \mathrm{Tr}(D^2 Z^2)\mathcal{O}(x_3, y_3)\ldots\mathcal{O}(x_5, y_5)\rangle + \ldots, \quad (99)$$

where the ... represent subleading operators and the correlator involving the Konishi is given by

$$\langle \mathrm{Tr}(D^2 Z^2)\mathcal{O}(x_3, y_3)\mathcal{O}(x_4, y_4)\mathcal{O}(x_5, y_5)\rangle = \frac{\left(x_{34}^2/x_{41}^2\right)^{\gamma_\mathcal{K}/2}}{(x_{13}^2)^{2+\gamma_\mathcal{K}/2}(x_{45}^2)^2}\sum_{\ell=0}^{2}V_{1,34}^{2-\ell}V_{1,35}^{\ell}A^{\ell}(u_3, u_4), \quad (100)$$

where $V_{i,jk} = (z_i \cdot x_{ij}x_{ik}^2 - z_i \cdot x_{ik}x_{ij}^2)/x_{jk}$ is a typical tensor structure [64] (where $z_i^\mu$ with $z_i^2 = 0$ is a spin polarization vector) that usually appears in spinning correlators, and $\gamma_\mathcal{K} = 12g^2 - 48g^4 + \ldots$ is the anomalous dimension of the Konishi operator. In the limit $x_2 \to x_1$, the cross ratios $u_3$ and $u_5$ (C.2) become the usual four-point cross ratios. The coefficients, $A^\ell$, in this decomposition are finite functions of the cross ratios and have a perturbative expansion in the coupling $g$ (recall that this information is contained in the two-loop five-point function):

$$A^{\ell}(u_3, u_4) = \sum_{k=0}^{\infty} g^{2k}A_k^{\ell}(u_3, u_4), \quad (101)$$

$$A_2^0(u_3, u_4) = \mathcal{I}_1 a_{1,0} + \mathcal{I}_2 a_{2,0} + \mathcal{I}_3 a_{3,0} + b_{1,0}(\Phi^{(1)})^2 + b_{2,0}\Phi^{(1)} + \sum_{i=1}^{3}\sum_{j=0}^{1} c_{i,j,0}\partial_{u_{3+j}}\mathcal{I}_i + d_0,$$

$$A_2^1(u_3, u_4) = \mathcal{I}_1 a_{1,1} + \mathcal{I}_2 a_{2,1} + \mathcal{I}_3 a_{3,1} + b_{1,1}(\Phi^{(1)})^2 + b_{2,1}\Phi^{(1)} + \sum_{i=1}^{3}\sum_{j=0}^{1} c_{i,j,1}\partial_{u_{3+j}}\mathcal{I}_i + d_1,$$

$$A_2^2(u_3, u_4) = \mathcal{I}_1 a_{1,2} + \mathcal{I}_2 a_{2,2} + \mathcal{I}_3 a_{3,2} + b_{1,1}(\Phi^{(1)})^2 + b_{2,2}\Phi^{(1)} + \sum_{i=1}^{3}\sum_{j=0}^{1} c_{i,j,2}\partial_{u_{3+j}}\mathcal{I}_i + d_2,$$

where the integrals $\mathcal{I}_i$ are just the two loop ladders defined in (E.14), and $\Phi^{(1)} = F_1^{2354}/(x_{24}^2 x_{35}^2)$ is the one-loop box integral (70). Some of the coefficients $a_{i,j}$, $b_{i,k}$, $c_{i,j,k}$, and $d_i$ are given in Appendix E.2, and the full four-point function is given in the auxiliary file `CorrelatorInLimit.m`. We have omitted the terms with $k < 2$ because they can be read off from tree-level and one-loop five-point functions.

---

[32]Here we are being schematic. The goal of this equation is to show the overall structure and what four-point functions can be read off. We have suppressed the dependence on the OPE coefficients and some space-time prefactors multiplying the four-point functions. We also drop the dependence on the polarizations $y_3 \ldots y_5$ in (100) since it gives only an overall prefactor that does not depend on the coupling.

Our two-loop result for the correlator of a Konishi operator with three **20'** operators extends the earlier one-loop result [65]. Note that the one-loop result can also be obtained from integrability and the OPE [13].

# 6 Discussion

We have used numerically the twistor reformulation of $\mathcal{N} = 4$ SYM to completely fix the two-loop five-, six- and seven-point correlation functions and the three-loop five-point correlator of twenty prime operators with arbitrary polarizations. It is possible to generate numerical data for even higher-points and higher loops by isolating sets of polarizations and keeping a reasonable number of twistor diagrams. However the ansatz as described in Section 2 for those cases has still too many undetermined coefficients for a fitting.

It is also possible to follow the approach of this paper for correlators of higher-charge operators. One difficulty is the increasing size of the ansatz, which however could be mitigated by bootstrap methods similar to the ones we investigated.

We find that all two-loop as well as the five-point three-loop correlator of **20'** operators are free of non-planar corrections. We argued that this might remain true for any number of points, which however remains to be verified.

Our results for the two-loop five-, six-, and seven-point correlators of **20'** operators can be expressed in terms of a restricted set of conformal integrals, see (70) and Figure 1. All integrals that contribute have a propagator connecting the two integration points, and at most one numerator factor per integration point. Beyond seven points, there is only one further integral of this type:[33]

$$\Delta_{1234,5678} \equiv x_{23}^2 x_{67}^2 x_{48}^2 \int \frac{dx_a \, dx_b \, x_{1b}^2 x_{5a}^2}{x_{1a}^2 x_{2a}^2 x_{3a}^2 x_{4a}^2 x_{ab}^2 x_{5b}^2 x_{6b}^2 x_{7b}^2 x_{8b}^2} = \qquad . \quad (102)$$

If the pattern of contributing integrals that we observe for $n \leq 7$ continues to higher points, then all two-loop correlation functions of **20'** operators should be expressible in terms of the integrals in Figure 1 and the above eight-point integral. If true, this puts an extension of our results to eight points within reach, since it substantially reduces the numbers of undetermined coefficients in the ansatz (see Table 5). Of course it is also possible that the absence of other conformal integrals is a low-$n$ artifact and does not continue to higher points. It would be nice to understand more systematically which kinds of integrals can contribute at higher points (and at higher loops, where the data is much more limited).

With the fresh new data obtained in this paper, we have made some initial steps towards bootstrapping the integrand of correlation functions with five or more half-BPS operators. An obvious next step in this bootstrap game is to explore correlation functions of operators with different $R$-charge. This is essentially an uncharted territory and definitely deserves further analysis, specially because one might wonder if there are hidden structures as there are for four points [10, 15]. The twistor reformulation of $\mathcal{N} = 4$ SYM can also be very useful for this generalization.

We have also studied the first non-trivial order in the light cone OPE of a five point function and have obtained two loop four point function involving one Konishi operator in the

---

[33]This integral has five numerators, which is not in contradiction with the analysis of the previous section. Note that this is just a conformal integral, and we decided to add prefactors such that the weight in each point is zero.

[0, 2, 0] and three **20′** operators. It would be interesting to further develop this OPE analysis to subleading terms in the OPE, higher point functions and higher loops. This would be very important to probe the recently discovered dualities between three point functions and null polygon hexagonal Wilson loops [5,6]. The main obstacle to achieve this is the computation of five and six point conformal integrals (which in this paper we have only computed in a limit). We hope to make progress on this in the future.

The four point function of **20′**-operators was important to analyze many different physical limits, such as the Regge limit or event shapes [66–68]. It would be very interesting to use our recent data and extensions of it to study these physical observables with more points.

Another direction is the connection with integrability [11–13]. We have not managed to find a closed expression for the $n$-point correlation functions at two-loop in this work. However, it is possible to organize the results in terms of integrability contributions. At two-loop and for a particular set of mirror cuts, the only contributions are strings and loops of just one mirror particles. There are many relations between these objects involving several particles with the same objects with lower number of particles, such as decoupling and flipping. It is expected that all the necessary integrability contributions can be fixed. The set of integrals appearing for the correlation functions of the twenty primes operators forms also a basis of integrals for the correlators of other length-$k$ half-BPS operators. This follows because increasing the bridge lengths kill possible diagrams. Thus fitting the basis against the power series produced by integrability in a line (the integrals are unknown outside the line) should give the integrand for different correlators such as the dodecagon. The same strings and loops can also be used to produce non-planar data.

In addition, notice that the twistor method for computing correlation functions was the motivation for the proposal of the Correlahedron [28], which is a geometric object computing the integrands of correlation functions of the stress-tensor multiplet. One of the properties of this object is that it reduces to the "squared" Amplituhedron [69] when the light-like limit is taken.[34] The Correlahedron is a Grassmannian, and the external data are points in chiral Minkowski space. However, there are still some open questions about the proposal. An important one is concerning the volume form. The volume form for correlation functions can be more complicated, having different kinds of singularities. All the known results for correlation function integrands in the literature were shown to have an uplift to the Correlahedron language. This includes the four-point functions up to ten loops [23, 24], and the six-point tree-level correlator mentioned above. We hope that our new data can help to test the proposal further.

## Acknowledgments

We would like to thank Frank Coronado, Paul Heslop, Raul Pereira, Pedro Vieira for useful discussions. V.G. would like to thank Maria Nocchi for reading carefully part of Appendix E.

**Funding information** This work was supported by the Serrapilheira Institute (grant number Serra - R-2012-38185), and funded by the Deutsche Forschungsgemeinschaft (DFG, German Research Foundation) – 460391856. T.F. would like to thank the warm hospitality of the KITP Santa Barbara during the program Integrable22 where part of this work was done. This research was supported in part by the National Science Foundation under Grant No. NSF PHY-1748958. V.G. is supported by Simons Foundation grants #488637 (Simons collaboration on

---

[34]The proposed "squared" Amplituhedron is also a Grassmannian, but with fewer constraints [28] than the original Amplituhedron.

the non-perturbative bootstrap). Centro de Física do Porto is partially funded by Fundação para a Ciência e Tecnologia (FCT) under the grant UID04650-FCUP.

## A  Tree-level and one-loop $n$-point functions

This appendix is a review of the known results in the literature about tree-level and one-loop $n$-point functions of length-two half-BPS operators. At tree level and with general polarizations $Y_i$, one has

$$G_n\big|_0 = \left(d_{12}d_{23}\dots d_{n-1,n}d_{n1} + \text{non-cyclic permutations}\right) + \text{disconnected}, \qquad \text{(A.1)}$$

with propagators $d_{ij}$ as in (4).

At one-loop order, there are two ways of obtaining the results. The first method is the perturbative calculation of [21], and the second method uses integrability techniques [19]. The starting point of the integrability calculation is to consider all tree-level diagrams. The perturbative corrections are obtained by adding the so called mirror particle contributions. The relevant tree-level diagrams for integrability are the connected ones.[35] Considering the sphere, the cyclic graphs divide it into two faces, where each face forms a polygon with $2n$ edges for $n$ operators (each operator has a small size). At one-loop order for length-two half-BPS operators, the mirror particles in different faces do not interact, and the correlator is the product of the value of the two polygons. At two-loop order, this factorization breaks down and there are strings and loops of mirror particles connecting the two faces. From integrability, one has

$$\text{polygon}(1,\dots,2n) = \sum_{\substack{[i,i+1],[j,j+1]: \\ \text{non-consecutive edges}}} m(z_{ij}, \alpha_{ij}), \qquad \text{(A.2)}$$

where

$$m(z, \alpha) \equiv g^2 \frac{(z+\bar{z})-(\alpha+\bar{\alpha})}{2} F^{(1)}(z,\bar{z}), \qquad \text{(A.3)}$$

with the local cross ratios[36]

$$z_{ij}\bar{z}_{ij} = \frac{x_{i,j+1}^2 x_{i+1,j}^2}{x_{i,i+1}^2 x_{j+1,j}^2}, \qquad (1-z_{ij})(1-\bar{z}_{ij}) = \frac{x_{i,j}^2 x_{i+1,j+1}^2}{x_{i,i+1}^2 x_{j+1,j}^2},$$

$$\alpha_{ij}\bar{\alpha}_{ij} = \frac{y_{i,j+1}^2 y_{i+1,j}^2}{y_{i,i+1}^2 y_{j+1,j}^2}, \qquad (1-\alpha_{ij})(1-\bar{\alpha}_{ij}) = \frac{y_{i,j}^2 y_{i+1,j+1}^2}{y_{i,i+1}^2 y_{j+1,j}^2}, \qquad \text{(A.4)}$$

and

$$F^{(1)}(z,\bar{z}) = \frac{1}{z-\bar{z}}\left(2\operatorname{Li}_2(z) - 2\operatorname{Li}_2(\bar{z}) + \log(z\bar{z})\log\left(\frac{1-z}{1-\bar{z}}\right)\right), \qquad \text{(A.5)}$$

which is the box integral (70):

$$F^{(1)}\left(\frac{1}{1-z_{13}}, \frac{1}{1-\bar{z}_{13}}\right) = \frac{1}{\pi^2}F_1^{1243}, \qquad F^{(1)}(z_{13}, \bar{z}_{13}) = \frac{1}{\pi^2}F_1^{1432}. \qquad \text{(A.6)}$$

---

[35]The disconnected diagrams can in principle contribute to the integrability calculation because of the stratification procedure (which is a prescription for treating the boundary graphs of the moduli space) in hexagonalization. However in [19], it was argued that these contributions vanish at one-loop order. At the moment, it is not known if they contribute to higher-loop correlators.

[36]Notice that the expression for the polygon is valid for any 4d kinematics. All the local cross ratios $z_{ij}$ and $\bar{z}_{ij}$ depend only on the $n(n-3)/2$ cross ratios of the problem (which is also the number of terms in the sum in (A.2)).

The final result for the one-loop correlators is the sum of two of these polygon factors times the tree-level propagators of each tree-level graph. The calculation from integrability relies on the two-mirror-particle contribution obtained in [18] and the flip relations explored in [19] that enable one to iteratively obtain the contribution of "strings" of any number $n$ of interacting mirror particles. The result for the general polygon (A.2) also follows from an inductive argument. In order to compute the correlation functions of operators with length bigger than two, it is also possible to use the formula for the polygons. Similarly, each tree-level graph divides the surface into faces or polygons. However, the number of tree-level graphs grows substantially with the length $k$ of the operators involved. At the moment, as far as we know there is no closed formula for general $k$ even for the simplest case of five operators. The function (A.3) depends on the $R$-charge cross-ratios $\alpha_{ij}$, and therefore can change the original polarization structure of a given tree-level graph. In addition, the function $F^{(1)}(z, \bar{z})$ of (A.5) satisfies several properties, and many simplifications are expected when all graphs are summed. Note that it is possible to generate several correlators with different $k$'s using the expression for the polygons.

## B  Ansatz construction

As explained in Section 2, by mapping each factor $x_{ij}^2$ to an edge between vertices $i$ and $j$, we can identify each monomial in $x_{ij}^2$ with a multi-graph (*i.e.* a graph that admits "parallel" edges between the same vertices $i$ and $j$). Finding the most general polynomials $P_a^{(\ell)}$ hence amounts to listing all multi-graphs with $n$ external vertices with valency $\ell$, and $\ell$ internal valencies with valency $n + \ell - 5$, and taking a general linear combination of the corresponding monomials.

We split the construction of the ansatz for each polynomial $P_a$ into three steps. First, we construct all admissible *unlabeled* graphs with $n + \ell$ vertices.[37] Next, for each graph $g$, we construct a set of inequivalent labelings of the external vertices. Each such labeling $\sigma$ will correspond to one independent term in the ansatz that gets multiplied by an undetermined coefficient $c_{g,\sigma}$. In order to find the minimal set of inequivalent labelings, we make use of the permutation symmetry $K_a$ of the respective propagator factor $\prod_{ij} d_{ij}^{a_{ij}}$. Due to the total $S_n$ permutation symmetry of the correlator, also the polynomial $P_a$ must be invariant under the permutation group $K_a$. The set of inequivalent labelings therefore is $K_a \backslash S_n / H_g$, where $H_g$ is the automorphism group of the graph $g$.[38] Finally, we symmetrize each labeled graph over the residual symmetry group $K_a \times S_\ell$, where $S_\ell$ permutes the integration vertices. Putting everything together, we arrive at

$$P_a^{(\ell)} = \sum_{g \in \Gamma_{n,\ell}} \sum_{\sigma \in K_a \backslash S_n / H_g} c_{g,\sigma} \sum_{\pi \in K_a \times S_\ell} g_{\pi \circ \sigma} \,. \tag{B.1}$$

Here, $\Gamma_{n,\ell}$ is the set of all unlabeled multi-graphs with $n$ vertices of valency $\ell$, and $\ell$ vertices of valency $n + \ell - 5$.[39] For each graph $g \in \Gamma_{n,\ell}$, we sum over the labelings (permutations) $\sigma \in K_a \backslash S_n / H_g$ of the $n$ external points, where $H_g$ is the automorphism group of $g$, and $K_a$ is the symmetry group of the respective propagator factor $\prod_{ij} d_{ij}^{a_{ij}}$. Each such labeling produces

---

[37]For $n = 5$, all vertices have the same valency $\ell$, hence we need to explicitly distinguish different partitionings of the vertex set into external and internal vertices.

[38]To be precise, we do not care about $S_\ell$ relabelings of the integration points, since we will symmetrize the ansatz over $S_\ell$ permutations in any case. Hence the relevant group is $H_g = \mathrm{Aut}(g)/S_\ell|_n$, where $\mathrm{Aut}(g)$ is the automorphism group of $g$, and $|_n$ means restriction to the points $\{1, \dots, n\}$. For $n \neq 5$, this step is trivial, since internal and external vertices have different valencies and thus $\mathrm{Aut}(g) \subset S_n \times S_\ell$.

[39]In practice, the graph vertices are always labeled. However, the initial labeling of $g \in \Gamma_{n,\ell}$ is arbitrary and not relevant.

one independent term, *i.e.* comes with one independent coefficient $c_{g,\sigma}$. At the end, each independent term is symmetrized over permutations $K_a \times S_\ell$. To find all the graphs as well as their inequivalent labelings $\sigma \in K_a \backslash S_n / H_g$, we use SAGEMATH [49], in particular its interface to GAP [70].

We find the following numbers of different multi-graphs for various $n$ and $\ell$, where we distinguish internal from external vertices, but otherwise treat all vertices as identical (unlabeled):

| $|\Gamma_{5,2}|$ | $|\Gamma_{6,2}|$ | $|\Gamma_{7,2}|$ | $|\Gamma_{8,2}|$ | $|\Gamma_{9,2}|$ | $|\Gamma_{10,2}|$ | $|\Gamma_{11,2}|$ | $|\Gamma_{5,3}|$ | $|\Gamma_{6,3}|$ |
|---|---|---|---|---|---|---|---|---|
| 15 | 41 | 85 | 178 | 327 | 607 | 1051 | 429 | 4105 |

(B.2)

The residual permutation symmetry groups are as follows:

$$K_{23} = S_2 \times S_3, \qquad K_{222} = S_3 \ltimes (S_2 \times S_2 \times S_2), \qquad K_{223} = (S_2 \ltimes (S_2 \times S_2)) \times S_3,$$
$$K_5 = D_5, \qquad K_{24} = S_2 \times D_4, \qquad K_{25} = S_2 \times D_5,$$
$$K_{33} = S_2 \ltimes (S_3 \times S_3), \qquad K_{34} = S_3 \times D_4,$$
$$K_6 = D_6, \qquad K_7 = D_7, \qquad (B.3)$$

where $D_k$ is the dihedral group on $k$ elements. According to the above procedure, the total number $\aleph_a^{(\ell)}$ of independent terms (undetermined coefficients) in the ansatz for $P_a^{(\ell)}$ is

$$\aleph_a^{(\ell)} = \sum_{g \in \Gamma_{n,\ell}} |K_a \backslash S_n / H_g|. \qquad (B.4)$$

These are the numbers shown in Table 1.

## C  Kinematics

We define the general conformally invariant cross ratios:

$$u_{ijkl} = \frac{x_{ij}^2 x_{kl}^2}{x_{ik}^2 x_{jl}^2}, \qquad v_{ijkl} = \frac{x_{il}^2 x_{jk}^2}{x_{ik}^2 x_{jl}^2}. \qquad (C.1)$$

A collection of $n$ points in four dimensions has $4n - 15$ conformally invariant degrees of freedom, and therefore as many independent cross ratios.

**Five Points.** At $n = 5$, there are five independent cross ratios. A convenient choice is

$$u_i = u_{i,i+1,i+2,i+4}, \qquad 1 \le i \le 5, \qquad (C.2)$$

where the point labels are understood modulo 5. These are the same cross ratios used in [6]. One can express any conformally invariant combination of distances $x_{ij}$ in terms of the $u_i$ by comparing expressions in a fixed conformal frame. For example, one can set $x_5 = \infty$ and $x_{12}^2 = 1$. The relations (C.2) then imply

$$x_{13}^2 = \frac{1}{u_1}, \quad x_{14}^2 = \frac{1}{u_1 u_4}, \quad x_{23}^2 = \frac{u_2 u_5}{u_1}, \quad x_{24}^2 = \frac{u_5}{u_1 u_4}, \quad x_{34}^2 = \frac{u_3 u_5}{u_1 u_4}. \qquad (C.3)$$

**Six Points.** At six points, there are nine independent cross ratios. As a basis, we choose

$$u_i = u_{i,i+1,i+2,i+4}, \quad 1 \le i \le 6 \qquad \text{and} \qquad U_i = u_{i,i+2,i+3,i+5}, \quad 1 \le i \le 3, \qquad (C.4)$$

where the point labels are understood modulo 6. These are the same cross ratios used in [5] (see Figure 5 there). Again, one can go to a conformal frame where $x_6 = \infty$ and $x_{12}^2 = 1$, which fixes all remaining distances $x_{ij}^2$, $1 \le i, j \le 5$ in terms of the $u_i$ and $U_i$.

**Seven Points.** At seven points, one can pick a basis of 14 multiplicatively independent cross ratios. They will not be completely functionally independent, because seven points in four dimensions have only 13 conformally invariant degrees of freedom. The 14 multiplicatively independent cross ratios reduce to 13 degrees of freedom via a conformal Gram relation.[40] Nonetheless, any ratio of $x_{ij}^2$ can be uniquely written as a ratio of 14 multiplicatively independent cross ratios. One "nice" basis of 14 cross ratios appears to be:

$$\{u_{12j7} \mid 3 \le j \le 6\} \cup \{u_{1i7j} \mid 2 \le i < j \le 6\}. \tag{C.5}$$

Another potentially useful basis is (this is closer to the six-point set (C.4)):

$$u_i = u_{i,i+1,i+2,i+4}, \quad 1 \le i \le 7 \qquad \text{and} \qquad U_i = u_{i,i+2,i+3,i+6}, \quad 1 \le i \le 7. \tag{C.6}$$

Again, one can go to a conformal frame where $x_7 = \infty$ and $x_{12}^2 = 1$, which fixes all remaining distances $x_{ij}^2$, $1 \le i, j \le 6$ in terms of either of the two sets.

## D Explicit correlator expressions

### D.1 Correlator components

**Six Points.** Besides (78), the remaining six-point two-loop component functions are quoted in the following. The expressions are also included in the attached file `results.m`.

$$
\begin{aligned}
f_{24}^{(2)} = 4\Big\langle &-8B_{123,456}u_{1234}u_{2456} + B_{134,256}(2u_{1246} + 6u_{1243}u_{2365} - 2u_{1245}u_{2563}) \\
&+ 8\Pi_{3,45,126}u_{1263} - 8\Pi_{3,54,126}u_{1263} + \Pi_{1,34,256}(-8u_{1246} + 8u_{1245}u_{1652}) \\
&+ \Delta_{12,34,65}(-2u_{1245} + 4u_{1542}) + \Delta_{12,34,56}(2u_{1246} - 4u_{1642}) \\
&+ F_1^{1236}F_1^{1245}(-4u_{1243}u_{1462} + 4u_{1253}u_{1462} \\
&\qquad\qquad + u_{1234}u_{1263}u_{1356} + 3u_{1236}u_{1254}u_{1362} - u_{1246}u_{1253}u_{1462}) \\
&+ F_1^{1234}F_1^{3456}(-4u_{1243} + 4u_{1243}u_{3564} - 4u_{1243}u_{3456}u_{3564}) \\
&- 8B_{3,12,45}u_{1234} - 8B_{3,12,46}u_{1234} + 16B_{3,14,26}u_{1234} + B_{1,23,45}(-8u_{1354} + 8u_{1453}) \\
&+ 8\Pi_{3,45,12} - 8\Pi_{3,14,25}u_{1245} + \Pi_{3,12,45}(4 - 8u_{1425}) + 4\Delta_{35124} \\
&+ F_1^{1234}F_1^{1236}(4u_{1243}u_{1263} + 8u_{1263}u_{1342} - 4u_{1246}u_{1362}) + 24F_2^{1234}u_{1243} \\
&- 4F_2^{3142}u_{1243} + F_2^{1324}(4 + 2u_{1243}) + F_2^{1325}(4 - 2u_{1253}) - 4B_{123}\Big\rangle_{24}\,, \tag{D.1}
\end{aligned}
$$

$$
\begin{aligned}
f_{33}^{(2)} = 36\Big\langle &B_{124,356}(4u_{1245} + 2u_{1643} - 2u_{1243}u_{2356} - 2u_{1645}u_{2356} - 2u_{1642}u_{2653}) \\
&+ \Pi_{1,42,356}(4u_{1356} + 4u_{1623} - 4u_{1653} - 4u_{1325}u_{1653}) \\
&+ \Pi_{1,24,356}(-4u_{1356} - 4u_{1643} + 4u_{1653} + 4u_{1345}u_{1653}) \\
&+ \Delta_{14,25,63}(-2 - 2u_{1354} + 2u_{1453}) + \Delta_{14,25,36}(-2u_{1456} + 4u_{1654}) \\
&+ F_1^{1246}F_1^{1345}(2u_{1264} + 2u_{1452} - 2u_{1453} + 2u_{1254}u_{1364} + 2u_{1364}u_{1452} \\
&\qquad\qquad - 4u_{1356}u_{1463} - u_{1364}u_{1436}u_{1452} + u_{1264}u_{1426}u_{1453} + u_{1256}u_{1423}u_{1462}) \\
&- 8B_{1,23,45}u_{1354} + B_{1,24,35}(4 + 4u_{1354} - 4u_{1423} + 4u_{1453}) \\
&+ B_{1,24,56}(-4u_{1425} + 4u_{1524}) + 4\Pi_{1,23,45} + \Pi_{1,24,35}(-4 + 4u_{2543}) \\
&- 2F_2^{1425}u_{1254} + F_2^{1243}(-4 + 2u_{1432}) + F_2^{1245}(-2 + 6u_{1254} + 2u_{1452})\Big\rangle_{33}\,, \tag{D.2}
\end{aligned}
$$

---

[40]The seven-point conformal Gram relation takes the form $\det_{i,j}(x_{ij}^2) = 0$, see the discussion below (20).

$$\begin{aligned}
f_6^{(2)} = 3\Big\langle & B_{123,456}(-8 + 4u_{1436} + 16u_{2456} - 4u_{1432}u_{2456} - 4u_{1536}u_{2456}) \\
& + B_{124,356}(-2 - 4u_{1245} - 2u_{1346} + 2u_{1643} - 8u_{1243}u_{2356} \\
& \qquad\qquad + 4u_{1645}u_{2356} + 8u_{2653} + 8u_{1246}u_{2653} - 4u_{1345}u_{2653} - 8u_{1642}u_{2653}) \\
& + \Pi_{1,23,456}(16 - 8u_{1436} - 8u_{1456} - 16u_{1654} + 8u_{1435}u_{1654}) \\
& + \Pi_{1,25,346}(-4 + 8u_{1346} - 4u_{1356} - 4u_{1643} + 8u_{1653} - 4u_{1345}u_{1653}) \\
& + \Pi_{1,32,456}(-16 + 8u_{1426} + 8u_{1456} + 16u_{1654} - 8u_{1425}u_{1654}) \\
& + \Pi_{1,34,256}(-8 - 8u_{1256} + 4u_{1642} + 4u_{1652} - 4u_{1245}u_{1652}) \\
& + \Pi_{1,36,245}(4 + 4u_{1245} - 8u_{1265} - 8u_{1542} + 4u_{1562} + 4u_{1246}u_{1562}) \\
& + \Pi_{1,43,256}(8 + 8u_{1256} - 4u_{1632} - 4u_{1652} + 4u_{1235}u_{1652}) \\
& + \Delta_{12,36,45}(-4 + 4u_{1265} - 4u_{1562}) + \Delta_{12,36,54}(4 - 4u_{1264} + 4u_{1462}) \\
& + \Delta_{14,23,56}(-4 + 4u_{1436} - 4u_{1634}) + \Delta_{14,23,65}(-4u_{1435} + 8u_{1534}) \\
& + \Delta_{14,25,36}(-2 + 2u_{1456} - 2u_{1654}) + \Delta_{14,25,63}(4u_{1354} - 2u_{1453}) \\
& + F_1^{1234}F_1^{1456}(4 + 4u_{1263} + 2u_{1465} - 4u_{1564} \\
& \qquad\qquad - 8u_{1264}u_{2435} + 2u_{1462}u_{2435} + 4u_{1364}u_{2534} - 2u_{1463}u_{2534}) \\
& + F_1^{1236}F_1^{1245}(-4 - 6u_{1254} + 2u_{1263} - 4u_{1264} - 4u_{1362} + 4u_{1452} \\
& \qquad\qquad - 4u_{1264}u_{1352} + 6u_{1254}u_{1362} + 4u_{1256}u_{1362} - 2u_{1263}u_{1452} + 4u_{1256}u_{1462} \\
& \qquad\qquad + 4u_{1352}u_{1462} - 2u_{1234}u_{1263}u_{1356} - 6u_{1236}u_{1254}u_{1362} + 2u_{1246}u_{1253}u_{1462}) \\
& + F_1^{1245}F_1^{1346}(-4 - 4u_{1254} + 6u_{1452} - 4u_{1453} - 2u_{1463} + 2u_{1356}u_{1463} \\
& \qquad\qquad + 2u_{1256}u_{1462} + u_{1364}u_{1436}u_{1452} + u_{1264}u_{1426}u_{1453} - u_{1256}u_{1423}u_{1462}) \\
& + B_{1,23,45}(20 - 16u_{1324} - 12u_{1354} + 8u_{1423} - 12u_{1453} + 8u_{1325}u_{1453}) \\
& + B_{1,23,46}(16 + 16u_{1364} + 8u_{1423} - 8u_{1463} - 8u_{1326}u_{1463}) \\
& + B_{1,23,56}(-8u_{1523} - 4u_{1326}u_{1563}) \\
& + B_{1,24,36}(-12 + 8u_{1324} - 4u_{1364} - 4u_{1423} + 8u_{1463} - 4u_{1326}u_{1463}) \\
& + B_{1,25,34}(-4 + 4u_{1325} + 16u_{1345} + 4u_{1523} - 8u_{1324}u_{1543}) \\
& + B_{1,25,36}(-4 + 4u_{1365} + 4u_{1563} - 4u_{1326}u_{1563}) \\
& + B_{1,26,34}(8u_{1326} - 8u_{1346} + 4u_{1623} - 20u_{1643} + 4u_{1324}u_{1643}) \\
& + \Pi_{1,23,45}(8 + 8u_{2435}) + \Pi_{1,23,56}(12 + 4u_{2536} - 4u_{2635}) \\
& + \Pi_{1,24,35}(-8 - 8u_{2345} + 8u_{2543}) + \Pi_{1,25,34}(-8 - 8u_{2354} + 8u_{2453}) \\
& + \Pi_{1,25,36}(-4 - 12u_{2356} + 4u_{2653}) + \Pi_{1,26,35}(-8 - 8u_{2365} + 8u_{2563}) \\
& + \Pi_{1,34,25}(-4 + 4u_{2354} + 4u_{2453}) - 4\Delta_{13,456} \\
& + F_1^{1234}F_1^{1236}(-8 - 8u_{1243} + 8u_{1342} - 8u_{1362} + 4u_{1246}u_{1362} + 4u_{1342}u_{1362}) \\
& + F_2^{1234}(-16 + 16u_{1342}) + F_2^{1243}(-2 + 6u_{1234} + 2u_{1432}) \\
& + F_2^{1245}(-4 + 12u_{1254} + 4u_{1452}) + F_2^{1246}(-12 + 6u_{1462}) \\
& + F_2^{1254}(-4 + 4u_{1245} + 4u_{1542}) + F_2^{1263}(4 - 4u_{1236} - 20u_{1632}) \\
& + F_2^{1264}(-4 - 4u_{1246} + 4u_{1642}) - 8F_2^{1265}u_{1256} - 16F_2^{1364}u_{1643} + 4B_{123}\Big\rangle_6 .
\end{aligned} \tag{D.3}$$

As before, $\langle\cdot\rangle_{24}$, $\langle\cdot\rangle_{33}$, and $\langle\cdot\rangle_6$ means averaging over the respective permutation symmetry group $K_a$, see (B.3). The results are expressed in terms of the conformal integrals (70) and general cross ratios (C.1). They can easily be converted to the independent basis (C.4) of six-point cross ratios by expanding in $x_{ij}^2$ and setting $x_6 = \infty$ and $x_{12}^2 = 1$ (see Section 4.6).

**Seven Points.** At seven points, the two-loop component functions besides (79) are as follows. Again, all expressions are included in the attached file `results.m`.

$$
\begin{aligned}
f_{25}^{(2)} = 10\Big\langle & F_1^{1237}F_1^{3456}(-4u_{1273}-4u_{1273}u_{3465}+4u_{1273}u_{3564}) \\
& -4B_{123,456}u_{1234}u_{2456}+B_{134,256}(2u_{1246}+6u_{1243}u_{2365}-2u_{1245}u_{2563}) \\
& -4B_{123,457}u_{1237}u_{2754}+\Pi_{1,34,256}(-4u_{1246}+4u_{1245}u_{1652}) \\
& +\Pi_{1,34,267}(-4u_{1247}+4u_{1246}u_{1762})-4\Pi_{3,56,124}u_{1243}+4\Pi_{3,45,127}u_{1273} \\
& -4\Pi_{3,54,127}u_{1273}+4\Pi_{3,56,127}u_{1273}+\Delta_{12,34,65}(-u_{1245}+2u_{1542}) \\
& +\Delta_{12,34,56}(2u_{1246}-4u_{1642})+\Delta_{12,34,76}(-u_{1246}+2u_{1642}) \\
& +F_1^{1237}F_1^{1245}(-2u_{1243}u_{1472}+4u_{1253}u_{1472}-2u_{1257}u_{1472} \\
& \qquad\qquad +u_{1234}u_{1273}u_{1357}+3u_{1237}u_{1254}u_{1372}-u_{1247}u_{1253}u_{1472}) \\
& +F_1^{1245}F_1^{3456}(-2u_{1254}+2u_{1253}u_{2346}-2u_{1256}u_{2643}) \\
& -4B_{3,12,45}u_{1234}-4B_{3,12,47}u_{1234}+8B_{3,14,27}u_{1234}+B_{1,23,45}(-4u_{1354}+4u_{1453}) \\
& +4\Pi_{3,45,12}-4\Pi_{3,14,25}u_{1245}+\Pi_{3,12,45}(2-4u_{1425})+2\Delta_{35,124} \\
& +F_1^{1234}F_1^{1237}(2u_{1243}u_{1273}+4u_{1273}u_{1342}-2u_{1247}u_{1372})+12F_2^{1234}u_{1243} \\
& -2F_2^{3142}u_{1243}+F_2^{1324}(2+u_{1243})+F_2^{1325}(2-u_{1253})-2B_{123}\Big\rangle_{25}\,, \qquad (\text{D.4})
\end{aligned}
$$

$$
\begin{aligned}
f_{34}^{(2)} = 24\Big\langle & \Pi_{145,2367}(4-2u_{2365}-4u_{2764}+2u_{2364}u_{3457}) \\
& +\Pi_{415,2367}(-4+4u_{1375}+4u_{1672}+4u_{2365}-4u_{1372}u_{2365}-4u_{1675}u_{2365}) \\
& +\Delta_{1,245,367}(-2u_{1765}+2u_{1546}u_{1765}) \\
& +\Delta_{1,245,637}(-4+4u_{1547}+4u_{1735}-4u_{1534}u_{1745}) \\
& +\Delta_{1,425,637}(2-4u_{1527}+2u_{1523}u_{1735}) \\
& +\Delta_{1,425,736}(-2+4u_{1526}-2u_{1523}u_{1635}) \\
& +F_1^{1267}F_1^{1345}(-2+2u_{1672}+2u_{2463}+3u_{1276}u_{1354} \\
& \qquad\qquad +u_{1256}u_{1374}-4u_{1374}u_{1652}+2u_{1674}u_{2364}-2u_{1673}u_{2463} \\
& \qquad\qquad -2u_{1453}u_{1675}u_{2365}+4u_{1354}u_{1675}u_{2465}-u_{1275}u_{1354}u_{2564}) \\
& +B_{124,356}(4u_{1643}+4u_{2356}-4u_{1243}u_{2356}-4u_{1546}u_{2356}+4u_{1246}u_{2653}-4u_{1642}u_{2653}) \\
& +B_{145,267}(2-u_{1257}+2u_{1752}+3u_{1254}u_{2476}-2u_{2674}+u_{1256}u_{2674}-2u_{1652}u_{2674}) \\
& -4B_{124,567}u_{1245}u_{2567}+\Pi_{1,24,356}(4u_{1346}-4u_{1356}-4u_{1643}+4u_{1653}) \\
& +\Pi_{1,42,356}(-4u_{1326}+4u_{1356}+4u_{1623}-4u_{1653})+\Pi_{1,45,267}(-4+4u_{1762}) \\
& +\Pi_{4,15,237}(4u_{2374}-4u_{2473}+4u_{2475}-4u_{2453}u_{2574}) \\
& +\Pi_{4,51,237}(-4u_{1742}-4u_{2374}+4u_{1342}u_{2374}+4u_{2473})+4\Pi_{4,56,127}u_{1274} \\
& -4\Pi_{4,65,127}u_{1274}+\Delta_{14,25,37}(-2u_{1457}+4u_{1754}) \\
& +\Delta_{14,25,73}(-4-4u_{1354}+4u_{1453})+\Delta_{14,52,73}(4u_{1324}-2u_{1423}) \\
& +F_1^{1245}F_1^{4567}(-2u_{1254}+2u_{1254}u_{4675}-2u_{1254}u_{4567}u_{4675}) \\
& +F_1^{1247}F_1^{1345}(+4u_{1274}+4u_{1452}-4u_{1453}+4u_{1254}u_{1374} \\
& \qquad\qquad -4u_{1374}u_{1432}+4u_{1374}u_{1452}-4u_{1357}u_{1473} \\
& \qquad\qquad -2u_{1374}u_{1437}u_{1452}+2u_{1274}u_{1427}u_{1453}+2u_{1257}u_{1423}u_{1472}) \\
& -8B_{1,23,45}u_{1354}+B_{1,24,35}(6u_{1354}-2u_{1423}+4u_{1453}) \\
& +B_{1,24,36}(-2u_{1364}-2u_{1423}+4u_{1326}u_{1463})+B_{1,24,56}(-4u_{1425}+4u_{1524}) \\
& +B_{1,45,67}(-2u_{1675}+2u_{1547}u_{1675})+B_{4,12,35}(4u_{2435}-4u_{2534}) \\
& -4B_{4,12,56}u_{1245}-4B_{4,12,57}u_{1245}+B_{4,15,27}(4u_{1245}-4u_{2457}+4u_{2754}+4u_{1742}u_{2754})
\end{aligned}
$$

$$+ 4\Pi_{1,23,45} + \Pi_{1,24,35}(-4 + 4u_{2543}) + \Pi_{4,15,26}(-4 + 4u_{1652}) + 4\Pi_{4,56,12}$$
$$+ F_2^{1243}(-4 + 2u_{1432}) + F_2^{1245}(-4 + 12u_{1254} + 4u_{1452})$$
$$- 2F_2^{1425}u_{1254} + F_2^{1456}(-4 + 2u_{1564}) - 2F_2^{4152}u_{1254}\big\rangle_{34}, \tag{D.5}$$

$$
\begin{aligned}
f_7^{(2)} = 7\Big\langle & \Pi_{123,4567}(4 + 4u_{2476} - 4u_{2573} - 8u_{2674} - 4u_{3654} \\
& \qquad - 4u_{2576}u_{3456} + 4u_{2675}u_{3456} - 4u_{2475}u_{3654} + 8u_{2574}u_{3654} + 4u_{2673}u_{3654}) \\
& + \Pi_{125,3467}(-2 + 2u_{2376} + 2u_{2475} - 2u_{3465} + 4u_{3564} \\
& \qquad + 2u_{2374}u_{3465} + 2u_{2675}u_{3465} - 4u_{2375}u_{3564} - 2u_{2476}u_{3564} - 2u_{2674}u_{3564}) \\
& + \Pi_{127,3456}(4u_{2365} - 4u_{2563} - 4u_{2364}u_{3457} + 4u_{2463}u_{3457} - 4u_{2465}u_{3754} + 4u_{2564}u_{3754}) \\
& + \Pi_{145,2367}(-2u_{2365} - 2u_{2764} + 2u_{2364}u_{3457} + 2u_{2765}u_{3457} - 2u_{2465}u_{3754} + 2u_{2564}u_{3754}) \\
& + \Delta_{1,234,567}(4 - 4u_{1437} - 4u_{1764} + 4u_{1436}u_{1764}) \\
& + \Delta_{1,234,657}(-4 + 4u_{1437} + 4u_{1754} - 4u_{1435}u_{1754}) + \Delta_{1,234,756}(4 - 2u_{1436} - 2u_{1654}) \\
& + \Delta_{1,256,347}(2 - 2u_{1657} - 2u_{1746} + 2u_{1645}u_{1756}) \\
& + \Delta_{1,256,437}(-2 + 2u_{1657} + 2u_{1736} - 2u_{1635}u_{1756}) + \Delta_{1,256,734}(2 - u_{1436} - u_{1654}) \\
& + \Delta_{1,324,567}(-4 + 4u_{1427} + 4u_{1764} - 4u_{1426}u_{1764}) + \Delta_{1,324,657}(4 - 2u_{1427} - 2u_{1754}) \\
& + \Delta_{1,347,526}(-2 + 2u_{1627} + 2u_{1746} - 2u_{1624}u_{1746}) + \Delta_{1,347,625}(2 - u_{1527} - u_{1745}) \\
& + \Delta_{1,423,567}(4 - 2u_{1327} - 2u_{1763}) + \Delta_{1,437,526}(2 - u_{1627} - u_{1736}) \\
& + F_1^{1234}F_1^{1567}(2u_{1243} + 2u_{1576} + 4u_{1273}u_{1546} - 4u_{1372}u_{1546} + 4u_{1243}u_{1576} \\
& \qquad - 4u_{1342}u_{1576} - 4u_{1273}u_{1645} + 2u_{1372}u_{1645} - 4u_{1243}u_{1675} \\
& \qquad + 2u_{1342}u_{1675} + 2u_{1245}u_{2536} + 2u_{1572}u_{2536} - 2u_{1246}u_{2635} - 2u_{1573}u_{2635} \\
& \qquad + 4u_{1274}u_{1645}u_{2435} - 4u_{1274}u_{1546}u_{2436} - 2u_{1374}u_{1645}u_{2534} + 4u_{1374}u_{1546}u_{2634}) \\
& + F_1^{1256}F_1^{1347}(-3u_{1265} + u_{1374} + 2u_{1275}u_{1364} - 6u_{1265}u_{1374} - 2u_{1275}u_{1463} \\
& \qquad + 6u_{1265}u_{1473} - 2u_{1374}u_{1562} + u_{1473}u_{1562} - 2u_{1364}u_{1572} + u_{1463}u_{1572} \\
& \qquad + u_{1263}u_{2354} + u_{1372}u_{2354} - u_{1264}u_{2453} - u_{1375}u_{2453} \\
& \qquad - u_{1463}u_{1576}u_{2356} + 2u_{1364}u_{1576}u_{2456} + 2u_{1276}u_{1463}u_{2653} - 2u_{1276}u_{1364}u_{2654}) \\
& + B_{123,456}(-2 - 2u_{1436} + 2u_{1634} + 4u_{2654} + 2u_{1234}u_{2456} \\
& \qquad - 2u_{1635}u_{2456} + 4u_{1435}u_{2654} - 2u_{1534}u_{2654} - 2u_{1632}u_{2654}) \\
& + B_{124,356}(-1 - 2u_{1245} - u_{1346} + u_{1643} + 4u_{2653} - 4u_{1243}u_{2356} \\
& \qquad + 2u_{1645}u_{2356} + 4u_{1246}u_{2653} - 2u_{1345}u_{2653} - 4u_{1642}u_{2653}) \\
& + B_{124,567}(-8 + 4u_{1246} + 4u_{1547} + 4u_{1642} + 8u_{2765} + 4u_{1245}u_{2567} \\
& \qquad - 4u_{1247}u_{2765} - 4u_{1546}u_{2765} - 4u_{1742}u_{2765}) \\
& + B_{125,346}(2 - 2u_{1452} - 4u_{2346} + 2u_{2643} + 4u_{1352}u_{2346} \\
& \qquad + 2u_{1654}u_{2346} - 6u_{1256}u_{2643} - 2u_{1354}u_{2643} - 2u_{1652}u_{2643}) \\
& + B_{125,347}(-u_{1357} - u_{2347} - u_{1754}u_{2347} - u_{1257}u_{2743} \\
& \qquad + 2u_{1253}u_{2347} + 2u_{1457}u_{2347} + 2u_{1354}u_{2743} - 2u_{1453}u_{2743}) \\
& + \Pi_{1,23,457}(8 - 4u_{1437} - 4u_{1457} - 8u_{1754} + 4u_{1435}u_{1754}) \\
& + \Pi_{1,23,567}(4 - 4u_{1567} - 4u_{1765}) + \Pi_{1,24,567}(4 - 4u_{1547} - 4u_{1765} + 4u_{1546}u_{1765}) \\
& + \Pi_{1,25,347}(2u_{1347} - 2u_{1357} - 2u_{1743} + 2u_{1753}) \\
& + \Pi_{1,26,347}(-2 + 2u_{1347} + 2u_{1763} - 2u_{1346}u_{1763}) \\
& + \Pi_{1,26,457}(-2 + 4u_{1457} - 2u_{1467} - 2u_{1754} + 4u_{1764} - 2u_{1456}u_{1764}) \\
& + \Pi_{1,32,457}(-8 + 4u_{1427} + 4u_{1457} + 8u_{1754} - 4u_{1425}u_{1754})
\end{aligned}
$$

$$+ \Pi_{1,32,567}(-4 + 4u_{1567} + 4u_{1765}) + \Pi_{1,34,256}(-2 - 6u_{1256} + 2u_{1652})$$

$$+ \Pi_{1,34,267}(-4 - 4u_{1267} + 2u_{1742} + 2u_{1762} - 2u_{1246}u_{1762})$$

$$+ \Pi_{1,34,567}(4 - 4u_{1567} - 4u_{1765})$$

$$+ \Pi_{1,37,245}(2 + 2u_{1245} - 4u_{1275} - 4u_{1542} + 2u_{1572} + 2u_{1247}u_{1572})$$

$$+ \Pi_{1,37,256}(2 - 2u_{1276} - 2u_{1652} + 2u_{1257}u_{1672})$$

$$+ \Pi_{1,42,567}(-4 + 4u_{1527} + 4u_{1765} - 4u_{1526}u_{1765})$$

$$+ \Pi_{1,43,256}(2 + 6u_{1256} - 2u_{1652})$$

$$+ \Pi_{1,43,267}(4 + 4u_{1267} - 2u_{1732} - 2u_{1762} + 2u_{1236}u_{1762})$$

$$+ \Pi_{1,43,567}(-4 + 4u_{1567} + 4u_{1765})$$

$$+ \Pi_{1,45,237}(4 - 2u_{1237} - 2u_{1257} + 4u_{1732} + 2u_{1235}u_{1752})$$

$$+ \Pi_{1,45,267}(-4 - 4u_{1267} + 2u_{1752} + 2u_{1762} - 2u_{1256}u_{1762})$$

$$+ \Pi_{1,47,256}(2u_{1256} - 2u_{1276} - 2u_{1652} + 2u_{1672})$$

$$+ \Delta_{12,37,45}(-2 + 2u_{1275} - 2u_{1572}) + \Delta_{12,37,54}(2 - 2u_{1274} + 2u_{1472})$$

$$+ \Delta_{12,37,56}(-2 + 2u_{1276} - 2u_{1672}) + \Delta_{12,37,65}(2 - 2u_{1275} + 2u_{1572})$$

$$+ \Delta_{14,23,57}(-4 + 4u_{1437} - 4u_{1734}) + \Delta_{14,23,75}(4 - 4u_{1435} + 4u_{1534})$$

$$+ \Delta_{14,25,37}(-1 + u_{1457} - u_{1754}) + \Delta_{14,25,73}(2 + 2u_{1354} - 2u_{1453})$$

$$+ \Delta_{14,52,73}(-1 - u_{1324} + u_{1423})$$

$$+ F_1^{1234}F_1^{1457}(2 + 4u_{1273} - 4u_{1372} + 2u_{1475} - 2u_{1574} - 4u_{1274}u_{2435}$$
$$- 2u_{1375}u_{2435} + 2u_{1472}u_{2435} + 2u_{1275}u_{2534} + 4u_{1374}u_{2534} - 2u_{1473}u_{2534})$$

$$+ F_1^{1237}F_1^{1245}(-4 - 2u_{1253} - 6u_{1254} + 2u_{1273} - 2u_{1274} - 4u_{1372}$$
$$+ 4u_{1452} - 2u_{1274}u_{1352} + 6u_{1254}u_{1372} + 2u_{1257}u_{1372} + 2u_{1243}u_{1452}$$
$$- 2u_{1273}u_{1452} + 2u_{1243}u_{1472} - 2u_{1253}u_{1472} + 2u_{1257}u_{1472}$$
$$+ 4u_{1352}u_{1472} - 2u_{1234}u_{1273}u_{1357} - 6u_{1237}u_{1254}u_{1372} + 2u_{1247}u_{1253}u_{1472})$$

$$+ F_1^{1245}F_1^{1347}(-4 - 4u_{1254} + 6u_{1452} - 2u_{1453} - 2u_{1472} - 2u_{1473}$$
$$+ u_{1354}u_{1432} + u_{1374}u_{1432} + u_{1257}u_{1472} + u_{1357}u_{1473}$$
$$+ u_{1374}u_{1437}u_{1452} + u_{1274}u_{1427}u_{1453} - u_{1257}u_{1423}u_{1472})$$

$$+ B_{1,23,45}(10 - 8u_{1324} - 6u_{1354} + 4u_{1423} - 6u_{1453} + 4u_{1325}u_{1453})$$

$$+ B_{1,23,47}(8 + 8u_{1374} + 4u_{1423} - 4u_{1473} - 4u_{1327}u_{1473})$$

$$+ B_{1,23,56}(6 - 4u_{1325} - 6u_{1365} - 6u_{1563} + 4u_{1326}u_{1563})$$

$$+ B_{1,23,57}(-4 + 4u_{1325} + 4u_{1573} - 4u_{1327}u_{1573}) + B_{1,23,67}(2 - 4u_{1326} - 4u_{1623})$$

$$+ B_{1,24,37}(-6 + 4u_{1324} - 2u_{1374} - 2u_{1423} + 4u_{1473} - 2u_{1327}u_{1473})$$

$$+ B_{1,24,56}(4 - 4u_{1425} + 4u_{1524}) + B_{1,24,57}(2 + 4u_{1524} - 2u_{1427}u_{1574})$$

$$+ B_{1,25,34}(2u_{1325} + 6u_{1345} - 2u_{1324}u_{1543})$$

$$+ B_{1,25,37}(-2 + 2u_{1375} + 2u_{1523} - 2u_{1327}u_{1573}) + B_{1,25,47}(-3 + 2u_{1574} - u_{1427}u_{1574})$$

$$+ B_{1,26,34}(-2 + 2u_{1346} + 2u_{1623} - 2u_{1324}u_{1643}) + B_{1,26,37}(-1 + 2u_{1673} - u_{1327}u_{1673})$$

$$+ B_{1,26,45}(-2 + 2u_{1426} + 8u_{1456} + 2u_{1624} - 4u_{1425}u_{1654})$$

$$+ B_{1,27,34}(4u_{1327} - 4u_{1347} + 2u_{1723} - 10u_{1743} + 2u_{1324}u_{1743})$$

$$+ B_{1,27,45}(2 - 2u_{1457} - 6u_{1754}) + B_{1,34,56}(3 - 6u_{1435} + 3u_{1436}u_{1564})$$

$$+ \Pi_{1,23,45}(4 + 4u_{2435}) + \Pi_{1,23,56}(6 + 2u_{2536} - 2u_{2635})$$

$$+ \Pi_{1,23,67}(6 + 2u_{2637} - 2u_{2736}) + \Pi_{1,24,35}(-4 - 4u_{2345} + 4u_{2543})$$

$$+ \Pi_{1,25,34}(-4 - 4u_{2354} + 4u_{2453}) + \Pi_{1,25,36}(-2 - 6u_{2356} + 2u_{2653})$$

$$+ \Pi_{1,26,35}(-4 - 4u_{2365} + 4u_{2563}) + \Pi_{1,26,37}(-2 - 6u_{2367} + 2u_{2763})$$
$$+ \Pi_{1,27,36}(-4 - 4u_{2376} + 4u_{2673}) + \Pi_{1,34,25}(-2 + 2u_{2354} + 2u_{2453}) - 2\Delta_{14,567}$$
$$+ F_1^{1234}F_1^{1237}(-4 - 4u_{1243} + 4u_{1342} - 4u_{1372} + 2u_{1247}u_{1372} + 2u_{1342}u_{1372})$$
$$+ F_2^{1234}(-8 + 8u_{1342}) + F_2^{1243}(-1 + 3u_{1234} + u_{1432})$$
$$+ F_2^{1245}(-4 + 12u_{1254} + 4u_{1452}) + F_2^{1247}(-6 - 6u_{1274} + 6u_{1472})$$
$$+ F_2^{1254}(-1 + 3u_{1245} + u_{1542}) + F_2^{1265}(-2 + 2u_{1256} + 2u_{1652})$$
$$+ F_2^{1273}(2 - 2u_{1237} - 10u_{1732}) + F_2^{1275}(-2 - 2u_{1257} + 2u_{1752})$$
$$- 4F_2^{1276}u_{1267} - 16F_2^{1374}u_{1743} + 2B_{123}\Big\rangle_7 . \tag{D.6}$$

Again, $\langle \cdot \rangle_{25}$, $\langle \cdot \rangle_{34}$, and $\langle \cdot \rangle_7$ means averaging over the respective permutation symmetry group $K_a$, see (B.3). The results are expressed in terms of the conformal integrals (70) and general cross ratios (C.1). To convert to a multiplicatively independent set of 14 cross ratios, one can expand all $u_{ijkl}$ in terms of $x_{ij}^2$, and then pick a conformal frame, for example by setting $x_7 = \infty$ and $x_{12}^2 = 1$, and solve for the remaining $x_{ij}^2$ in terms of a basis of 14 cross ratios, for example (C.5) or (C.6).

## D.2 Preliminary $n$-point guesses

Based on our explicit two-loop results for $n = 5, 6, 7$, we have tried to guess general $n$-point expressions for two classes of component functions, namely $f_{2,n-2}^{(2)}$ and $f_n^{(2)}$. These guesses are very preliminary and yet incomplete. To fully determine these $n$-point functions will require more input. Yet, at least a subset of terms is matched nicely by our guesses. The (preliminary and incomplete) expression for the component functions $f_{2,n-2}^{(2)}$ is:

$$f_{2,n-2}^{\text{guess}\,(2)} = 4(n-2)\times \tag{D.7}$$
$$\Big\langle -B_{123} + F_2^{1324} + F_2^{1325} + 6F_2^{1234}u_{1243} + \tfrac{1}{2}F_2^{1324}u_{1243} - F_2^{3142}u_{1243} - \tfrac{1}{2}F_2^{1325}u_{1253}$$
$$- 2B_{3,12,45}u_{1234} - 2B_{3,12,4n}u_{1234} + 4B_{3,14,2n}u_{1234} + B_{1,23,45}(-2u_{1354} + 2\delta_{n,5} + 2\Theta_{n\geq 6}u_{1453})$$
$$+ \Pi_{3,12,45} + 2\Pi_{3,45,12} - 2\Pi_{3,14,25}u_{1245} - 2\Pi_{3,12,45}u_{1425} + \Delta_{35,124}$$
$$+ F_1^{1234}F_1^{123n}(u_{1243}u_{12n3} + 2u_{12n3}u_{1342} - u_{124n}u_{13n2})$$
$$+ \frac{\Theta_{n\geq 6}}{4}\Big[-8B_{123,456}u_{1234}u_{2456} + 8\Pi_{3,45,12n}u_{12n3} - 8\Pi_{3,54,12n}u_{12n3}$$
$$- 8\Pi_{1,34,256}(u_{1246} - u_{1245}u_{1652}) - 2\Delta_{12,34,65}(u_{1245} - 2u_{1542})$$
$$- 4F_1^{12,n-3,n-2}F_1^{3456}u_{12,n-2,n-3} - 4F_1^{123n}F_1^{1245}u_{1243}u_{14n2}$$
$$+ c_n\Big(F_1^{123n}F_1^{1245}(u_{1234}u_{12n3}u_{135n} + 3u_{123n}u_{1254}u_{13n2} + 4u_{1253}u_{14n2} - u_{124n}u_{1253}u_{14n2})$$
$$+ 2B_{134,256}(u_{1246} + 3u_{1243}u_{2365} - u_{1245}u_{2563}) + 2\Delta_{12,34,56}(u_{1246} - 2u_{1642})\Big)\Big]\Big\rangle_{2,n-2},$$

where $\langle \cdot \rangle_{2,n-2}$ means averaging over the permutation group $K_{2,n-2} = S_2 \times D_{n-2}$ of the external labels $(1, \ldots, n)$, and

$$c_n = \begin{cases} 1, & n = 6, \\ 2, & n = 7. \end{cases} \tag{D.8}$$

With this expression, we find

$$f_{23}^{(2)} = f_{23}^{\text{guess}\,(2)},$$
$$f_{24}^{(2)} = f_{24}^{\text{guess}\,(2)} + 16\Big\langle F_1^{1234}F_1^{3456}u_{1243}u_{3564}(1 - u_{3456})\Big\rangle_{24},$$

$$f_{25}^{(2)} = f_{25}^{\text{guess}(2)} + 20\Big\langle -2B_{123,457}u_{1237}u_{2754} - 2\Pi_{3,56,124}u_{1243} + 2\Pi_{3,56,127}u_{1273}$$
$$+ \Pi_{1,34,267}(-2u_{1247} + 2u_{1246}u_{1762}) + \Delta_{12,34,76}(-\tfrac{1}{2}u_{1246} + u_{1642})$$
$$- F_1^{1237}F_1^{1245}u_{1257}u_{1472} + F_1^{1245}F_1^{3456}(u_{1253}u_{2346} - u_{1256}u_{2643})$$
$$- 2F_1^{1237}F_1^{3456}u_{1273}(1 + u_{3465} - u_{3564})\Big\rangle_{25}. \tag{D.9}$$

Similarly, our preliminary guess for the component $f_n^{(2)}$ is (this still involves a lot of guesswork):

$$f_n^{\text{guess}(2)} = 2n\times$$
$$\Big\langle B_{123} - \Delta_{1nnnn} + \Pi_{12345}(2 + 2u_{2435}) + \Pi_{13425}(-1 + u_{2354} + u_{2453})$$
$$+ \sum_{k=5}^n \big(\Pi_{12k3k}(-2 - 2u_{23kk} + 2u_{2kk3}) + \Pi_{12k3k}(-2 - 2u_{23kk} + 2u_{2kk3})\big)$$
$$+ \sum_{k=6}^n \big(\Pi_{12k3k}(1 - u_{23kk} - u_{2kk3}) + \Pi_{123kk}(3 + u_{2k3k} - u_{2k3k})\big)$$
$$+ B_{12345}(-4u_{1324} + 2u_{1423} + \delta_{n5}(-4u_{1354} + 2u_{1453} + u_{1325}u_{1453}))$$
$$+ B_{1243n}(-2 - 2u_{13n4} + 3u_{14n3} - u_{132n}u_{14n3})$$
$$+ B_{1234n}(4 + 4u_{13n4} + 2u_{1423} - 2u_{14n3} - 2u_{132n}u_{14n3}) + B_{12n34}(2u_{132n} - 2u_{1n43})$$
$$+ \sum_{k=5}^{n-1} B_{123kk}(-3u_{1kk3} - 3u_{13kk} + 2u_{132k}u_{1kk3}) + B_{123nn}(-2u_{1n23}) + B_{12n3n}u_{1nn3}$$
$$+ F_2^{1234}(-4 + 4u_{1342}) + F_2^{12nn}(-2u_{12nn}) + F_2^{12n3}(1 - u_{123n} - 5u_{1n32})$$
$$+ F_2^{12nn}(-1 + u_{12nn} + u_{1nn2}) + F_2^{12nn}(-1 - u_{12nn} + u_{1nn2})$$
$$+ F_1^{1234}F_1^{123n}(-2 - 2u_{1243} + 2u_{1342} + u_{13n2}(-2 + u_{124n} + u_{1342}))$$
$$+ d_1^c\big(F_2^{13n4}(-4u_{1n43}) + F_2^{1245}(-1 + 3u_{1254} + u_{1452})\big)$$
$$+ \Theta_{n\geq 6}\Big[F_1^{123n}F_1^{1245}(-u_{12n4}(1 + u_{1352}) + u_{125n}(u_{13n2} + u_{14n2}))$$
$$+ F_1^{123n}F_1^{1245}d_2^c\tfrac{1}{2}(-2 - 2u_{13n2} - 3u_{1254}(1 + (-1 + u_{123n})u_{13n2})$$
$$+ 2u_{1452} - u_{12n3}(-1 + u_{1234}u_{135n} + u_{1452}) + u_{124n}u_{1253}u_{14n2} + 2u_{1352}u_{14n2})$$
$$+ F_1^{1245}F_1^{134n}(-u_{1453} + \tfrac{1}{2}u_{125n}u_{14n2} + \tfrac{1}{2}u_{135n}u_{14n3})$$
$$+ F_1^{1245}F_1^{134n}d_2^c(-1 - u_{1254} + \tfrac{3}{2}u_{1452} + \tfrac{1}{4}u_{13n4}u_{143n}u_{1452}$$
$$+ \tfrac{1}{4}u_{12n4}u_{142n}u_{1453} - \tfrac{1}{4}u_{125n}u_{1423}u_{14n2} - \tfrac{1}{2}u_{14n3})$$
$$+ F_1^{1234}F_1^{145n}(1 - u_{15n4} - 2u_{12n4}u_{2435})$$
$$+ F_1^{1234}F_1^{145n}d_2^c(u_{12n3} + \tfrac{1}{2}u_{14n5} + \tfrac{1}{2}u_{14n2}u_{2435} + u_{13n4}u_{2534} - \tfrac{1}{2}u_{14n3}u_{2534})$$
$$+ 5B_{12345} + B_{1243n}(-1 + 2u_{1324} + u_{13n4} - u_{1423} - u_{14n3})$$
$$+ B_{1253n}(-1 + u_{13n5} - u_{132n}u_{15n3}) + B_{12n34}(-2u_{134n} + u_{1n23} - 3u_{1n43} + u_{1324}u_{1n43})$$
$$+ B_{12nnn}(-1 + u_{1n2n} + 4u_{1nnn} + u_{1n2n} - 2u_{1n2n}u_{1nnn}) + d_2^c B_{1235n}(-u_{132n}u_{15n3})$$
$$+ \sum_{k=4}^{n-2} F_2^{12kk}(-\tfrac{1}{2} + \tfrac{3}{2}u_{12kk} + \tfrac{1}{2}u_{1kk2}) + F_2^{124n}(-3 + \tfrac{3}{2}d_2^c u_{14n2} - 3d_3^c u_{12n4})$$
$$+ B_{124356}(-\tfrac{1}{2} - u_{1245} - \tfrac{1}{2}u_{1346} + \tfrac{1}{2}u_{1643} - 2u_{1243}u_{2356} + u_{1645}u_{2356} + 2u_{2653}$$
$$+ 2u_{1246}u_{2653} - u_{1345}u_{2653} - 2u_{1642}u_{2653})$$
$$+ \sum_{k=6}^n \sum_{j=3}^{k-3} \big(\Pi_{1jj2kk}(-1 - 3u_{12kk} + u_{1kk2}) + \Pi_{1jj2kk}(+1 + 3u_{12kk} - u_{1kk2})\big)$$
$$+ \sum_{k=4}^{n-2} \Big[\sum_{j=2}^{k-2} (\Pi_{1jjkkn} - \Pi_{1jjkkn})(+2 - 2u_{1kk+n} - 2u_{1nk+k})$$

$$
\begin{aligned}
&+ \Pi_{12\underaccent{\ddot{}}{k}kkn}(+2 - 2u_{1k\underaccent{\ddot{}}{k}n} - 2u_{1n\dot{k}k} + 2u_{1k\underaccent{\ddot{}}{k}k}u_{1n\dot{k}k}) \\
&+ \Pi_{1\underaccent{\dot{}}{k}2kkn}(-2 + 2u_{1k2n} + 2u_{1n\dot{k}k} - 2u_{1k2\dot{k}}u_{1n\dot{k}k}) \\
&+ \Pi_{12\dot{k}kkn}(u_{1\dot{k}kn} - u_{1\dot{k}kn} - u_{1n\dot{k}k} + u_{1n\dot{k}\dot{k}}) \\
&+ \Pi_{1\dot{k}n2kk}(u_{12k\dot{k}} - u_{12n\dot{k}} - u_{1\dot{k}k2} + u_{1\dot{k}n2}) \\
&+ \Pi_{13n2k\dot{k}}(1 - u_{12n\dot{k}} - u_{1\dot{k}k2} + u_{12kn}u_{1\dot{k}n2}) \\
&+ \Pi_{12\underaccent{\dot{}}{n}kkn}(-1 + u_{1\dot{k}kn} + u_{1n\dot{n}\dot{k}} - u_{1\dot{k}k\underaccent{\dot{}}{n}}u_{1n\dot{n}\dot{k}}) \\
&+ \Pi_{1k\dot{k}2\underaccent{\dot{}}{n}n}(-1 + 1u_{12\underaccent{\dot{}}{n}n} + u_{1nk2} - u_{12k\underaccent{\dot{}}{n}}u_{1n\underaccent{\dot{}}{n}2}) \\
&+ \Pi_{1k\dot{k}2\underaccent{\dot{}}{n}n}(+1 - 1u_{12\underaccent{\dot{}}{n}n} - u_{1nk2} + u_{12k\underaccent{\dot{}}{n}}u_{1n\underaccent{\dot{}}{n}2}) \Big] \\
&+ \sum_{k=5}^{n-1}\Big(\Delta_{123n\dot{k}\dot{k}}(-1 + u_{12nk} - u_{1kn2}) + \Delta_{123nk\dot{k}}(1 - u_{12n\dot{k}} + u_{1\dot{k}n2})\Big) \\
&+ \Delta_{14253n}\tfrac{1}{2}(-1 + u_{145n} - u_{1n54}) + \Delta_{14235n}d_2^{c}(-1 + u_{143n} - u_{1n34}) \\
&+ \Delta_{1423n5}(2d_3^{c} - d_2^{c}u_{1435} + 2u_{1534}) + \Delta_{1425n3}\tfrac{1}{2}(2d_3^{c} - d_2^{c}u_{1453} + 2u_{1354})\Big] \Big\rangle_{\text{dihedral}},
\end{aligned}
\tag{D.10}
$$

where $\langle \cdot \rangle_n$ means averaging over the dihedral group $K_n \equiv D_n$ of external points, and we use the shorthand notation

$$
\underaccent{\ddot{}}{n} = n - 3, \quad \underaccent{\dot{}}{n} = n - 2, \quad \dot{n} = n - 1, \quad \ddot{n} = n + 1 \quad \text{etc.}
\tag{D.11}
$$

as well as

$$
d_1^{c} = \begin{cases} 0, & n = 5, \\ 1, & n = 6, \\ 2, & n = 7, \end{cases} \qquad
d_2^{c} = \begin{cases} 1, & n = 6, \\ 2, & n = 7, \end{cases} \qquad
d_3^{c} = \begin{cases} 0, & n \leq 6, \\ 1, & n = 7. \end{cases}
\tag{D.12}
$$

With this expression, we find

$$
\begin{aligned}
f_5^{(2)} &= f_5^{\text{guess}\,(2)}, \\
f_6^{(2)} &= f_6^{\text{guess}\,(2)} + 12\big\langle B_{123,456}\big(-2 + u_{1436} + u_{2456}(4 - u_{1432} - u_{1536})\big)\big\rangle_6, \\
f_7^{(2)} &= f_7^{\text{guess}\,(2)} + 14\big\langle -\tfrac{1}{2}\Delta_{14,52,73}(1 + u_{1324} - u_{1423}) + (12\ \text{five-point terms}\ B_{...}) \\
&\qquad + (4\ \text{six-point terms}\ B_{...}) + (3\ \text{six-point terms}\ F_1^{..}F_1^{...}) \\
&\qquad + (12\ \text{seven-point terms}\ \Delta_{...}) + (4\ \text{seven-point terms}\ \Pi_{...}) \\
&\qquad + (2\ \text{seven-point terms}\ F_1^{..}F_1^{...})\big\rangle_7.
\end{aligned}
\tag{D.13}
$$

# E  Integrals

## E.1  Asymptotic expansions

The goal of this appendix is to review and explain the main idea behind the method of asymptotic expansions that allows us to obtain the integrals in the correlators as a series expansion in one cross ratio. This will be a simple extension of the analysis that has been done for four-point conformal integrals [60–62].

A generic two-loop finite conformal integral with at most six external points has the following form[41]

$$I = \int \frac{[d^d x_\ell]}{x_{78}^2 \prod_{i=1}^6 (x_{i7}^2)^{a_i} (x_{i8}^2)^{b_i}}, \tag{E.1}$$

with $1 + \sum_i a_i = d$, $1 + \sum_i b_i = d$ and $[d^d x_\ell] \equiv d^d x_7 d^d x_8$. Without loss of generality, it is possible to use conformal symmetry to send one point to infinity, say $x_6$ (if the integral only has five points, then we send the point $x_5$ to infinity), and one point to zero, say $x_1$. The method of asymptotic expansions can be used to obtain a series expansion of an integral when one variable is small, say $x_2^2 \to 0$. Within this method, we are instructed to divide each integration region into two parts, one where the integration variable is of the size of $x_2^2$ and another where it is much bigger.[42] In each region, it is possible to simplify the integrand using the general expansion

$$\frac{1}{(x_{ij}^2)^c} = \sum_{n=0}^\infty \binom{-c}{n} \frac{(x_j^2 - 2x_i \cdot x_j)^n}{(x_i^2)^{c+n}}, \quad x_i \gg x_j, \qquad \binom{a}{b} \equiv \frac{\Gamma(a+1)}{\Gamma(1+a-b)\Gamma(1+b)}. \tag{E.2}$$

At two loops, there are four different regions:

$$\begin{aligned}
\text{region 1}: & \quad x_7, x_8 \approx x_2, \\
\text{region 2}: & \quad x_7 \approx x_2, \quad x_2 \ll x_8, \\
\text{region 3}: & \quad x_8 \approx x_2, \quad x_2 \ll x_7, \\
\text{region 4}: & \quad x_2 \ll x_7, x_8.
\end{aligned} \tag{E.3}$$

We will denote the integral (E.1) with the domain of integration restricted to the respective region by $I_k$, $k = 1, \ldots, 4$, and can expand:

$$\begin{aligned}
I_1 &= \sum_{n_1, \ldots n_6 = 0}^\infty \prod_{i=1}^3 \binom{-a_{i+2}}{n_i} \binom{-b_{i+2}}{n_{i+3}} \int \frac{[d^d x_\ell]}{x_{78}^2 \prod_{i=1}^2 (x_{i7}^2)^{a_i} (x_{i8}^2)^{b_i}} \prod_{j=3}^5 \frac{\prod_{i=0}^1 (x_{7+i}^2 - 2x_j \cdot x_{7+i})^{n_{j-2+3i}}}{(x_j^2)^{a_j + b_j + n_{j-2} + n_{j+1}}}, \\
I_2 &= \sum_{n_i = 0}^\infty \binom{-b_2}{n_4} \prod_{i=1}^3 \binom{-a_{i+2}}{n_i} \int \frac{[d^d x_\ell](2x_7 \cdot x_8 - x_7^2)^{n_5}(2x_2 \cdot x_8 - x_2^2)^{n_4}}{(x_8^2)^{1+b_1+b_2+n_4+n_5} \prod_{i=1}^2 (x_{i7}^2)^{a_i}} \prod_{j=3}^5 \frac{(x_7^2 - 2x_j \cdot x_7)^{n_{j-2}}}{(x_{8j}^2)^{b_j}(x_j^2)^{a_j + n_{j-2}}}, \\
I_4 &= \sum_{n_1, \ldots n_2 = 0}^\infty \binom{-a_2}{n_1} \binom{-b_2}{n_2} \int \frac{[d^d x_\ell](x_2^2 - 2x_2 \cdot x_7)^{n_1}(x_2^2 - 2x_2 \cdot x_8)^{n_2}}{x_{78}^2 (x_7^2)^{a_1+a_2+n_1}(x_8^2)^{b_1+b_2+n_2} \prod_{i=3}^5 (x_{i7}^2)^{a_i} (x_{i8}^2)^{b_i}},
\end{aligned} \tag{E.4}$$

where $I_3$ is obtained from $I_2$ by $a_i \leftrightarrow b_i$. The regions 1 and 4 are expressed in terms of two-loop integrals, while the regions 2 and 3 are given by products of two one-loop integrals. But in all of the regions the integrals have at most 4 external points, and thus are simpler. It is simple to extract the leading (when they go to zero) dependence on $x_2^2$:

$$I_1 \approx (x_2^2)^{d-(1+a_1+a_2+b_1+b_2)}, \quad I_2 \approx (x_2^2)^{\frac{d}{2}-a_1-a_2}, \quad I_4 \approx (x_2^2)^0. \tag{E.5}$$

The factors in the numerators of the integrals $I_k$ can be written as a combination of integrals with open indices contracted with some external vectors. As an example, take a two-loop

---

[41]Instead of the factor of $x_{78}^2$ in the denominator, there could be a factor $x_{78}^{2n}$, $n \geq 1$ in the numerator (see *e.g.* Figure 2), but such integrals do not occur in the correlators discussed in this work, and hence we do not consider them here. Higher powers of $x_{78}^2$ in the denominator do not appear in our integrals since they are divergent.

[42]In the asymptotic expansion method, we are instructed to integrate over all space and shift the dimension from 4 to $d = 4 - 2\epsilon$ to regulate possible divergences in each region that should cancel when all contributions are combined, as we shall see.

integral depending on two vectors $x_3$, $x_4$ with 2 open indices:[43]

$$\int \frac{[d^d x_\ell] N^{\mu_1 \mu_2}}{D(x_3, x_4)} = x_3^{\mu_1} x_3^{\mu_2} T_0 + x_3^{\mu_1} x_4^{\mu_2} T_1 + x_3^{\mu_2} x_4^{\mu_1} T_2 + x_4^{\mu_1} x_4^{\mu_2} T_3 + \delta^{\mu_1 \mu_2} T_4 \,, \qquad \text{(E.6)}$$

where $D$ is some denominator, and $T_i$ are scalar integrals that can be obtained by contracting this equation with $x_3^{\mu_1} x_3^{\mu_2}, \ldots, x_4^{\mu_1} x_4^{\mu_2}, \delta^{\mu_1 \mu_2}$ and inverting this system of equations.

**Example: Five-Point Double Pentaladder**

One of the integrals that appear in the two-loop five-point function is the 5-pt double pentaladder integral

$$I = \frac{\Delta_{23,145}}{x_{14}^2 x_{15}^2 c x_{45}^2} = \int \frac{[d^4 x_\ell] x_{27}^2 x_{38}^2}{x_{17}^2 x_{18}^2 x_{28}^2 x_{37}^2 x_{47}^2 x_{48}^2 x_{57}^2 x_{58}^2 x_{78}^2} \,, \qquad \text{(E.7)}$$

that is obtained from (E.1) by setting

$$a_1 = b_1 = b_2 = a_3 = a_4 = b_4 = a_5 = b_5 = 1, a_2 = b_3 = -1 \,.$$

Recall that we can send one point to infinity, say $x_5$. Let us analyze all four regions for this integral

$$I_1 = \sum_{n_1, n_2, n_3 = 0}^{\infty} \int \frac{[d^d x_\ell] x_{27}^2 x_{38}^2 (2x_3 \cdot x_7 - x_7^2)^{n_1} (2x_4 \cdot x_7 - x_7^2)^{n_2} (2x_4 \cdot x_8 - x_8^2)^{n_3}}{x_7^2 x_8^2 x_{28}^2 (x_3^2)^{1+n_1} (x_4^2)^{2+n_2+n_3} x_{78}^2} \,,$$

$$I_2 = \sum_{n_i = 0}^{\infty} \int \frac{[d^d x_\ell] x_{27}^2 x_{38}^2 (2x_2 \cdot x_8 - x_2^2)^{n_1} (2x_7 \cdot x_3 - x_7^2)^{n_2} (2x_7 \cdot x_4 - x_7^2)^{n_3} (2x_7 \cdot x_8 - x_7^2)^{n_4}}{x_7^2 (x_8^2)^{2+n_1+n_4} (x_3^2)^{1+n_2} (x_4^2)^{1+n_3} x_{48}^2} \,,$$

$$I_3 = \sum_{n_i = 0}^{\infty} \frac{1}{(x_4^2)^{1+n_1}} \int \frac{[d^d x_\ell] x_{27}^2 x_{38}^2 (2x_8 \cdot x_4 - x_8^2)^{n_1} (2x_8 \cdot x_7 - x_8^2)^{n_2}}{x_8^2 x_{28}^2 (x_7^2)^{2+n_2} x_{37}^2 x_{47}^2} \,,$$

$$I_4 = \sum_{n_1 = 0}^{\infty} \int \frac{[d^d x_\ell] x_{27}^2 x_{38}^2 (2x_8 \cdot x_2 - x_2^2)^{n_1}}{x_7^2 (x_8^2)^{2+n_1} x_{37}^2 x_{47}^2 x_{48}^2 x_{78}^2} \,. \qquad \text{(E.8)}$$

Some comments are in order here: $I_2$ does not contribute since the integral in $x_7$ integrates to zero (this is a scaleless integral [59]); the leading power of $I_1$ is $(x_2^2)^{1-2\epsilon}$ and thus this region is subleading compared to $I_3$ and $I_4$; the terms $x_8^2$ in the numerator of $I_3$ and $x_2^2$ in the numerator of $I_4$ can be dropped to leading order in $x_2^2 \to 0$. In the following we will focus on the last two regions[44]

$$I_3 = \frac{1}{(x_3^2)^{5-d}} \sum_{n_i = 0}^{\infty} \frac{2^{n_1+n_2} \Gamma(\epsilon) \Gamma(1-\epsilon)}{(x_4^2)^{1+n_1} (x_2^2)^\epsilon} \left[ \frac{x_3^2 \Gamma(1-\epsilon+n_1+n_2)}{\Gamma(2-2\epsilon+n_1+n_2)} - \frac{2x_3 \cdot x_2 \Gamma(2-\epsilon+n_1+n_2)}{\Gamma(3-2\epsilon+n_1+n_2)} \right]$$

$$\times (x_2 \cdot x_4)^{n_1} \int \frac{d^d x_7 (x_7^2 - 2x_2 \cdot x_7)(x_2 \cdot x_7)^{n_2}}{(x_7^2)^{2+n_2} x_{37}^2 x_{47}^2} \,,$$

---

[43]The generalization to more external vectors and more indices is straightforward. This formula follows just by the SO($d$) symmetry of the system.

[44]We have used the one loop integral formula with numerators

$$\int \frac{d^d x_0 \, x_0^{\mu_1} \ldots x_0^{\mu_J}}{(x_0^2)^\alpha (x_{02}^2)^\beta} = \frac{\Gamma(\alpha + \beta - \frac{d}{2}) \Gamma(\frac{d}{2} - \alpha + J) \Gamma(\frac{d}{2} - \beta)}{\Gamma(\alpha) \Gamma(\beta) \Gamma(d - \alpha - \beta + J)} \frac{x_2^{\mu_1} \ldots x_2^{\mu_J}}{(x_2^2)^{\alpha+\beta-\frac{d}{2}}} + \ldots \,, \qquad \text{(E.9)}$$

where the $\ldots$ represent subleading terms in $x_2^2$.

$$I_4 = \frac{1}{(x_3^2)^{5-d}} \sum_{n_1=0}^{\infty} 2^{n_1} \int \frac{d^4x_7 \, d^4x_8 \, (x_7^2 - 2x_2 \cdot x_7)x_{38}^2 (x_8 \cdot x_2)^{n_1}}{x_7^2 (x_8^2)^{2+n_1} x_{37}^2 x_{47}^2 x_{48}^2 x_{78}^2} , \tag{E.10}$$

where we have scaled all points by $x_i^\mu \to |x_3| x_i^\mu$. In practice we have to truncate upper limit of the sum, which translates into evaluating the integral as expansion around $x_2 \to 0$. The integrals in each region might have $\epsilon$ divergences which should cancel, for each order in $x_2$, when all regions are combined. The leading term in $x_2 \to 0$ comes from truncating both sums to the $n_i = 0$ and furthermore neglecting the $x_2 \cdot x_3$ and $x_2 \cdot x_7$ in the numerators of both regions

$$I_3 \to \frac{1}{(x_3^2)^{1+2\epsilon}} \frac{\Gamma(\epsilon)\Gamma^2(1-\epsilon)}{(x_4^2)(x_2^2)^\epsilon} \frac{1}{\Gamma(2-2\epsilon)} \int \frac{d^d x_7}{x_7^2 x_{37}^2 x_{47}^2} , \tag{E.11}$$

$$I_4 \to \frac{1}{(x_3^2)^{1+2\epsilon}} \int \frac{[d^d x_\ell] x_{38}^2}{(x_8^2)^2 x_{37}^2 x_{47}^2 x_{48}^2 x_{78}^2} . \tag{E.12}$$

Notice that each integral is divergent but when we plug the values of the integrals and combine them, the divergences disappear,[45] giving rise to

$$I = \frac{1}{8} \Big( \tilde{\Delta} \big( 4u_3 \partial_{u_3} \tilde{I}_2 - 4\partial_{u_3} \tilde{I}_3 - u_4 \partial_{u_4} \tilde{I}_1 + 4u_4 \partial_{u_4} \tilde{I}_2 \big) + 2u_3 u_4 \partial_{u_3} \tilde{I}_1 + 4u_3^2 \partial_{u_3} \tilde{I}_2 \tag{E.13}$$

$$-4(u_4+1)u_3 \partial_{u_3} \tilde{I}_2 - +4(1-u_3-u_4)\partial_{u_3} \tilde{I}_3 + u_3 u_4 \partial_{u_4} \tilde{I}_1 + (u_4-1)u_4^2 \partial_{u_4} \tilde{I}_1 + 4u_3 u_4 \partial_{u_4} \tilde{I}_2$$

$$+4(1-u_4)u_4 \partial_{u_4} \tilde{I}_2 - 8u_4 \partial_{u_4} \tilde{I}_3 + 8u_4 \Phi^{(1)}(2 - \log(u_1 u_4)) + 48\zeta_3 + (u_3 - u_4 + 1 + \tilde{\Delta})u_4 \mathcal{I}_1 \Big) ,$$

where $\tilde{\Delta} = \sqrt{(u_4 - u_3 + 1)^2 - 4u_4}$, we have used the conformal frame with $x_5 \to \infty$, $x_{14}^2 = 1$, and $\tilde{I}_j = \tilde{\Delta} \mathcal{I}_j$ with

$$\mathcal{I}_1 = \int \frac{d^4x_7 \, d^4x_8}{x_{17}^2 x_{18}^2 x_{37}^2 x_{38}^2 x_{47}^2 x_{78}^2} , \quad \mathcal{I}_2 = \int \frac{d^4x_7 \, d^4x_8}{x_{18}^2 x_{37}^2 x_{38}^2 x_{47}^2 x_{78}^2} , \quad \mathcal{I}_3 = \int \frac{d^4x_7 \, d^4x_8}{x_{17}^2 x_{18}^2 x_{37}^2 x_{48}^2 x_{78}^2} . \tag{E.14}$$

## E.2 Coefficients of the Konishi Four-Point Function

In the main text, we have written down the expression for the two-loop four-point correlator involving three **20′** operators and one Konishi operator in terms of a linear combination of conformal integrals. The coefficients in this linear combination are given by

$$a_{1,0} = \frac{c(z-1)(\bar{z}-1)}{15z^2\bar{z}^3}(24z^7(\bar{z}-1)^2\bar{z}^2(2\bar{z}-1) - z^4(\bar{z}-1)(\bar{z}(\bar{z}(\bar{z}(2\bar{z}(\bar{z}(48\bar{z}^2 - 369\bar{z} + 169)$$

$$+1087) - 7) - 1926) + 123) + 180) + z^6\bar{z}(\bar{z}(\bar{z}(161 - 2\bar{z}(3\bar{z}(248\bar{z} - 523) + 1028)) + 74)$$

$$+51) - z^5(\bar{z}(\bar{z}(\bar{z}(2\bar{z}(\bar{z}(6\bar{z}(16\bar{z} - 277) + 3067) - 2126) + 647) + 552) - 417) + 108)$$

$$+z^3(\bar{z}(\bar{z}(\bar{z}(\bar{z}(\bar{z}(5019 - 2\bar{z}(\bar{z}(24\bar{z} + 665) - 458)) - 10252) + 5208) - 1268) + 483) + 72)$$

$$+z^2(\bar{z}(\bar{z}(\bar{z}(\bar{z}(\bar{z}(\bar{z}(\bar{z}(276\bar{z} + 335) - 1223) + 1919) + 2370) - 4940) + 4983) - 1776) - 144)$$

$$-z\bar{z}(\bar{z}(\bar{z}(\bar{z}(\bar{z}(4\bar{z}(\bar{z}(33\bar{z} - 53) + 138) + 1473) - 2068) + 75) + 2280) - 1152)$$

$$+3\bar{z}^2(\bar{z}(\bar{z}(\bar{z}(\bar{z}(17\bar{z} + 93) - 77) - 81) + 176) - 48)) , \tag{E.15}$$

$$a_{1,1} = \frac{2c}{15z^2\bar{z}^3}(z-1)(\bar{z}-1)(48z^7(\bar{z}-1)^2\bar{z}^3 - 2z^6(\bar{z}-1)\bar{z}(\bar{z}(\bar{z}(744\bar{z}^2 - 657\bar{z} + 29) + 19)$$

$$-3) - z^5(\bar{z}(\bar{z}(2\bar{z}(\bar{z}(\bar{z}(96\bar{z}^2 - 966\bar{z} + 1459) - 462) + 15) + 313) - 333) + 36)$$

---

[45]We have used integration by parts identities (IBPS) to reduce each three point integral to a sum of master integrals that we have computed with the MAPLE package HYPERINT [71].

$$+ z^4(\bar{z}(\bar{z}(2\bar{z}(\bar{z}(\bar{z}(\bar{z}(-48\bar{z}^2 + 321\bar{z} + 98) - 998) + 1145) + 1041) - 1627) - 159) + 108)$$

$$+ 2z^2\bar{z}(\bar{z}(\bar{z}(\bar{z}(\bar{z}(\bar{z}(6\bar{z}^2 + 82\bar{z} - 3) + 1116) + 470) - 1047) + 876) - 180)$$

$$+ 2z^3(\bar{z}(\bar{z}(\bar{z}(\bar{z}(\bar{z}(\bar{z}(5\bar{z}(12\bar{z} - 79) - 114) + 1067) - 3251) + 1631) - 618) + 276) - 36)$$

$$+ z\bar{z}^2(\bar{z}(\bar{z}(900 - \bar{z}(\bar{z}(4\bar{z}(9\bar{z} - 8) + 469) + 1351)) + 24) - 360)$$

$$+ 3\bar{z}^3(\bar{z}(\bar{z}(\bar{z}(8\bar{z} + 81) - 37) + 52) - 24)),$$

$$a_{1,2} = \frac{c}{15z^2\bar{z}^2}(z-1)(\bar{z}-1)(-2z^5(\bar{z}(\bar{z}(\bar{z}(\bar{z}(96\bar{z}^2 - 630\bar{z} + 517) + 73) + 200) + 137) - 183)$$

$$+ 24z^7(\bar{z}-1)^2\bar{z}(2\bar{z}+1) - 2z^6(\bar{z}-1)(\bar{z}(\bar{z}(3\bar{z}(248\bar{z} - 91) - 55) + 23) - 15)$$

$$+ z^4(\bar{z}(\bar{z}(\bar{z}(2\bar{z}(3\bar{z}(51 - 16\bar{z}) + 284) - 883) + 1557) + 2635) - 1062) - 552)$$

$$+ z^3(\bar{z}(\bar{z}(\bar{z}(2\bar{z}(\bar{z}(72\bar{z} - 107) - 109) + 1047) - 5514) + 2385) - 456) + 576)$$

$$- z^2\bar{z}(\bar{z}(\bar{z}(\bar{z}(\bar{z}(6\bar{z}(6\bar{z} + 5) + 145) - 2872) + 243) + 720) + 288) + z\bar{z}^2(\bar{z}(264 - \bar{z}(\bar{z}(12\bar{z}$$

$$- 25) + 496) + 687)) + 576) + 3(\bar{z}-1)\bar{z}^4(5\bar{z} + 88)),$$

$$b_{1,0} = -\frac{4c}{15z\bar{z}}(z-1)(\bar{z}-1)(z^7\bar{z}(6\bar{z}(\bar{z}(\bar{z}(3\bar{z} - 10) + 12) - 6) + 1) + z^6(3\bar{z}(\bar{z}(9 - \bar{z}(\bar{z}(4\bar{z}(5\bar{z}$$

$$- 9) + 15) + 3)) + 1) + 1) + 3z^5(\bar{z}(\bar{z}(\bar{z}(\bar{z}(2\bar{z}(3\bar{z}(\bar{z} + 6) - 17) - 57) + 42) - 5) + 9) - 12)$$

$$- z^4(\bar{z}(\bar{z}(\bar{z}(\bar{z}(3\bar{z}(5\bar{z}(4\bar{z} + 3) + 57) - 916) + 804) - 126) + 9) - 72)$$

$$+ z^3(\bar{z}(\bar{z}(\bar{z}(3\bar{z}(3\bar{z}(\bar{z}(8\bar{z} - 1) + 14) - 268) + 916) - 171) - 45) - 60)$$

$$- 3z^2(\bar{z}(\bar{z}(\bar{z}(\bar{z}(\bar{z}(3\bar{z}(4\bar{z} - 3) + 5) - 42) + 57) + 34) - 36) - 6)$$

$$+ z\bar{z}(\bar{z}(\bar{z}(\bar{z}(\bar{z}(\bar{z}(\bar{z} + 3) + 27) - 9) - 45) + 108) - 60) + \bar{z}^2(\bar{z}(\bar{z}((\bar{z} - 36)\bar{z} + 72) - 60) + 18)),$$

$$b_{1,1} = -\frac{4c}{15z\bar{z}}(z-1)(\bar{z}-1)(36z^7(\bar{z}-1)^3\bar{z}^2 + z^6(2 - 3(1 - 2\bar{z})^2\bar{z}(5\bar{z}^2(2\bar{z} - 1) - 1))$$

$$+ z^3(\bar{z}^2(\bar{z}(3\bar{z}(9\bar{z}(4\bar{z}^2 + \bar{z} + 8) - 268) + 608) + 57) - 12) + 3z^5(\bar{z}(\bar{z}(\bar{z}(\bar{z}(4\bar{z}(3\bar{z}(\bar{z} + 5)$$

$$+ 1) - 133) + 72) - 5) + 22) - 12) - 3z^4(\bar{z}(\bar{z}(\bar{z}(\bar{z}(\bar{z}(6\bar{z}(6\bar{z} + 5) + 133) - 408) + 268) - 12)$$

$$+ 15) - 12) - 3z^2\bar{z}(\bar{z}(\bar{z}(\bar{z}(\bar{z}(4\bar{z}(3\bar{z} + 1) + 5) - 12) - 19) + 72) - 12) + 3z\bar{z}^2((\bar{z}(\bar{z} + 22)$$

$$- 15)\bar{z}^2 + 12) + 2\bar{z}^3(\bar{z}((\bar{z} - 18)\bar{z} + 18) - 6)),$$

$$b_{1,2} = \frac{-2c}{15z\bar{z}}(z-1)(\bar{z}-1)(z^7(\bar{z}-1)(12\bar{z}^3(3\bar{z} - 5) - 1) - 3z^2\bar{z}^2(\bar{z}(\bar{z}(3(\bar{z} - 2)\bar{z} + 41) - 36)$$

$$+ 48) + z^6(1 - \bar{z}(3\bar{z}(2\bar{z}(\bar{z}(4\bar{z}(5\bar{z} - 6) + 5) - 2) + 3) + 8)) + 3z^5\bar{z}(\bar{z}(\bar{z}(4\bar{z}(\bar{z}(3\bar{z}(\bar{z} + 4) + 7)$$

$$- 29) + 19) + 6) + 23) - z^4(\bar{z}(\bar{z}(\bar{z}(2\bar{z}(3\bar{z}(\bar{z}(16\bar{z} + 5) + 58) - 433) + 347) + 123) + 60)$$

$$+ 12) + z^3(\bar{z}(\bar{z}(\bar{z}(3\bar{z}(4\bar{z}(5\bar{z} + 1) + 19) - 347) + 250) + 108) + 60) - z\bar{z}^3((\bar{z} - 1)\bar{z}(\bar{z}(\bar{z} + 9)$$

$$- 60) - 60) + \bar{z}^7 + \bar{z}^6 - 12\bar{z}^4),$$

where $u_4 = z\bar{z}$, $u_3 u_5 = (1-z)(1-\bar{z})$, and $c = N_c^2(N_c^2 - 1)/(z-\bar{z})^4$. The other coefficients can be extracted from the ancillary MATHEMATICA file `CorrelatorInLimit.m`.

The procedure to obtain these coefficients is the following: Start by choosing the polarizations of points $x_1$ and $x_2$ according to (90), then insert the expressions of the conformal integrals, expanded as explained in the previous subsection,[46] take the limit $x_2 \to x_1$ and keep terms only to sub-subleading order (since the integrals were computed up to sub-subleading order in $x_2 \to x_1$, one is able to extract information about two primaries), use the light-cone conformal blocks (or alternatively the conformal Casimir equations), subtract the contribution of the twenty prime operator, and finally read off the coefficients by translating powers of $u_5 - 1$ and $u_2 - 1$ to the corresponding structures of a spinning four-point function. The resulting coefficients are listed in the file.

---

[46]We provide these expanded conformal integrals in the ancillary file `ListFivePtIntegrals.wl`.

### E.3 Simple One-Loop Integrals

In the main text, we have expressed the one-loop correlator involving half-BPS operators and the Lagrangian in terms of the following one-loop conformal integrals:

$$\int \frac{d^4 x_0 \, x_{06}^2}{x_{10}^2 x_{20}^2 x_{30}^2 x_{40}^2 x_{50}^2} = t_{12345,6} \,, \tag{E.16}$$

$$\int \frac{d^4 x_0 \, (x_{07}^2)^2}{x_{10}^2 x_{20}^2 x_{30}^2 x_{40}^2 x_{50}^2 x_{60}^2} = t_{123456,7} \,. \tag{E.17}$$

The first can be expressed in terms of linear combinations of one-loop box integrals while the second can be expressed in terms of the first, once we notice that $x_{07}^2 = \sum_{i=1}^{6} a_i x_{i0}^2$

$$t_{12345,6} = \frac{1}{\det x_{ij}^2} \sum_{i=1}^{5} a_i I_i \,, \tag{E.18}$$

where $I_i$ are one-loop box integrals where the position $i$ is absent and $a_i$ are polynomials in $x_{kl}^2$ with degree

$$\text{degree 1 in 6 and } i \text{ and 2 for the other points} \tag{E.19}$$

for $a_i$. These coefficients are highly constrained by symmetry (permutation of the points) and the fact that the integral should reduce to the usual ladder in the limit $x_6 \to x_i$ with $x_i$ being any of the external points in the denominator of the integral. For example, we have

$$a_5 = (x_{56}^2 \det x_{ij}^2) - 2 \sum_{i=1}^{4} x_{i5}^2 x_{i6}^2 \det{}_i' x_{lk}^2 + \sum_{i \neq j=1}^{5} x_{i5}^2 x_{j6}^2 V_{ij} \,, \tag{E.20}$$

where $V_{12} = x_{34}^2 (x_{14}^2 x_{23}^2 + x_{13}^2 x_{24}^2 - x_{12}^2 x_{34}^2)$ and the prime in the determinant means that point $x_i$ has been removed.

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
