# Peer review of "Higher-Point Integrands in N=4 super Yang-Mills Theory"

_SciPost Physics, doi:SciPost Phys. 15, 059 (2023)_

## Round 2 · Referee Report · Anonymous (Referee 1) · 2023-5-5

Report

I am happy with the way the authors addressed my comments so the paper can be published as it is.

---

## Round 2 · Referee Report · Anonymous (Referee 2) · 2023-5-9

Strengths

as before

Weaknesses

as before

Report

Let us not over-edit, I am happy with the extra paragraph on parity and also the comment on [64].

Yet the bulk references [11-14], [15-18] remain biased: including [15] on non-planar tilings means definite progress, but it postpones the contribution of the competing group by a year. Nobody in the community will deny that 1611.05577 is much more complete than 1611.05436 which does not contain comments on gluing and thus remains restricted to tree level.

But the two groups of references are introduced by the text "Correlation functions of single-trace half-BPS operators are also most amenable to the integrability-based hexagonalization approach [11-14], especially at higher points and higher genus [15-19]. 1611.05436 is about the idea of hexagon tilings - at tree - but for 4-pt functions, which are shown to work out on a number of examples. And the paper came (slightly) earlier. Where is it?

We do not want to open Pandorra's box. I do have a clear sense that this is not the only instance of "selective citing" in the manuscript.

From my point of view this requires a further amendment, but I leave the question to the editor.

Requested changes

See report.

  • validity: -
  • significance: -
  • originality: -
  • clarity: -
  • formatting: -
  • grammar: -

Author:  Thiago Fleury  on 2023-05-11  [id 3664]

(in reply to Report 2 on 2023-05-09)

Dear referee,
In the version 3 that was just resubmitted, the reference arxiv:1611.05436 was added to the Introduction.
We hope that our manuscript can now be accepted for publication.
Thanks.
Thiago

---

## Editorial Decision

published